# INDUCING UNCERTAINTY ON OPEN-WEIGHT MODELS FOR TEST-TIME PRIVACY IN IMAGE RECOGNITION

## ABSTRACT

A key concern for AI safety remains understudied in the machine learning (ML) literature: how can we ensure users of ML models do not leverage predictions on incorrect personal data to harm others? This is particularly pertinent given the rise of open-weight models, where simply masking model outputs does not suffice to prevent adversaries from recovering harmful predictions. To address this threat, which we call *test-time privacy*, we induce maximal uncertainty on protected instances while preserving accuracy on all other instances. Our proposed algorithm uses a Pareto optimal objective that explicitly balances test-time privacy against utility. We also provide a certifiable approximation algorithm which achieves $(\varepsilon, \delta)$ guarantees without convexity assumptions. We then prove a tight bound that characterizes the privacy-utility tradeoff that our algorithms incur. Empirically, our method obtains at least $> 3\times$ stronger uncertainty than pretraining with marginal drops in accuracy on various image recognition benchmarks. Altogether, this framework provides a tool to guarantee additional protection to end users.

## 1 INTRODUCTION

Data privacy is increasingly important for large-scale machine learning (ML), where models are often trained on sensitive user instances (European Parliament, 2016). Furthermore, open-weight image recognition models, where users have access to the model parameters and architecture, have proliferated (TorchVision, 2016; Google, 2023; Microsoft, 2024).

Yet, there has been little work done to address privacy threats to ML models due to incorrect personal data, especially data which are public such as images posted to public forums. Concretely, suppose a model provider trains an open-weight medical imaging model $f$ which classifies skin images as harmless ailments like "Benign Keratosis" or serious diseases like "Melanoma" (Sun et al., 2016). Next, a health insurance company scrapes images from public forums to build risk profiles. Then, this health insurance company downloads the open-weight model $f$ to automatically screen images for potential health liabilities. In particular, a person $p$ posts a photo of a harmless birthmark to a public health forum to ask a question. During the upload, a compression error causes the image file to become corrupted, severely distorting the birthmark. This results in an image $x_p$. When the health insurance company feeds $x_p$, scraped from the public forum, into $f$, it confidently classifies $x_p$ as "Melanoma". This erroneous classification is then automatically added to person $p$'s risk profile, resulting in person $p$ being unfairly denied coverage.[1] We call this threat model *test-time privacy* (TTP), and make this concrete in Fig. 1.[2] This privacy threat model is inspired by definitions of privacy which correspond to protecting a user from unfair interference or intrusion (Merriam-Webster, 2022). This differs from settings in privacy which mainly protect sensitive information.

We discuss why existing solutions, like unlearning (Sekhari et al., 2021) or differential privacy (Dwork et al., 2006) do not suffice to solve this problem in Sec. 2. Furthermore, naive solutions like masking model outputs do not work for open-weight image recognition models, since the model parameters

---

[1] Recently, there has been significant progress in building effective image recognition models for skin disease, making this problem pertinent (Yang et al., 2018); (Liu et al., 2025).

[2] We provide additional test-time privacy examples beyond medical classifiers in Appx. E.

Figure 1: An adversary, like a health insurance company (🏢), can query a pretrained model $f$ (🧠) and use its outputs to make harmful decisions. However, after running our algorithm, the new model $f_U$ (🧠) provides maximal uncertainty, protecting against such an adversary.

and architecture are available to the model user. An adversary could simply remove such a mask. To make this clear, we comprehensively detail our threat model as a security game in Appx. A and provide various motivating attacks in Appx. B. Therefore, we ask the following research question:

*Can we ensure test-time privacy against adversaries with access to an open-weight model?*

To do so, we argue it suffices to have uniform model outputs over the protected instances. That way, a data controller can only guess at the prediction. Thus, we revisit inducing maximal uncertainty over a dataset (Pereyra et al., 2017). Furthermore, we want to obtain high performance on all other instances as well. In particular, we answer our research question affirmatively, providing:

- A method to finetune a pretrained model with a Pareto optimal objective, rendering the model maximally uncertain over protected instances while preserving accuracy on others.
- Several principled $(\varepsilon, \delta)$-certified full-batch and sequential algorithms which approximate the Pareto objective, derived without assumptions of convexity.
- A theoretical analysis of the privacy-utility tradeoff that our algorithms incur, establishing a tight, non-vacuous bound.
- Empirical studies on image recognition models like ResNet50 (He et al., 2016) trained on datasets like CIFAR100 (Krizhevsky et al., 2009), observing that our algorithms maintain high uniformity on protected instances while guaranteeing excellent utility on the rest.

Following the literature on privacy, we focus on protecting a subset of the training data. However, as detailed in Sec. 4 and Appx. A, our setup and algorithms can also work for corrupted test instances. The code for our experiments is available for reproducibility at `https://tinyurl.com/testtimeprivacy`.

## 2 RELATED WORK

**Data Privacy**: Data leakage is a persistent danger for large information systems (Al-Rubaie and Chang, 2019). In the context of ML, data privacy is ubiquitous (Fredrikson et al., 2015); (Song et al., 2017); (Yeom et al., 2018). Approaches to privacy-preserving ML include differential privacy (DP) (Dwork et al., 2006); (Balle and Wang, 2018); (Amin et al., 2024), homomorphic encryption (Brakerski et al., 2014); (Lee et al., 2022); (Aono et al., 2017), and model obfuscation (Zhou et al., 2023). Notably, these methods protect against various privacy violations, like reconstruction attacks (Dinur and Nissim, 2003) due to failures of anonymization (Li et al., 2012). However, existing methods do not prevent confidently correct classification, and thus fail to protect against the attacks we consider in our setting. For example, if $x_p$ is a corrupted medical image record, an adversary may not be able to use a DP model $f$ to recover the record $x_p$ exactly, but they can still produce a confident prediction of e.g. "Melanoma" to harm person $p$.

A dominant viewpoint in the privacy community is that a model $f$ working as expected does not constitute a privacy violation, e.g. correctly predicting "Melanoma" for the corrupted medical image $x_p$, as it has learned something underlying about nature (Mcsherry, 2016); (Bun et al., 2021). Furthermore, the privacy leakage occurred when $x_p$ became public (Kamath, 2020). This view misses the point: ML models are often trained and applied on freely available data. For example, training data could be scraped from the web or social media platforms. Subsets of this data can be obsolete, corrupted, or confidential. With such data as input, model $f$ presents a clear and present danger for AI safety which differential privacy falls short of addressing, as the above example showed. In parallel, as humans, we learn to not act upon certain kinds of knowledge. For example, when we read confidential documents or learn that previously obtained knowledge is incorrect, we are not allowed to share or act on this knowledge.

**Unlearning**: A related subfield is machine unlearning, which is inspired by the right to be forgotten (RTBF), mandating that ML model providers delete user data upon request (European Parliament, 2016). In practice, model providers must remove user data and its effects from trained models and algorithms. Unlearning methods usually do so by approximating (and evaluating performance against) the model retrained from scratch without the protected user data (Sekhari et al., 2021); (Bourtoule et al., 2021); (Kurmanji et al., 2023).

However, while unlearning helps model providers comply with the RTBF, it cannot protect against attacks within our threat model. Specifically, recent unlearning research has established that data in the support of the training distribution will likely still be confidently predicted with the same prediction as before, even after using state-of-the-art algorithms (or even after applying exact retrain-from-scratch algorithms) to unlearn them (Zhao et al., 2024). That is, denoting the pretrained model as $f$ and the unlearned model as $f_u$, for typical training instances, it holds that $f = f_u$.

To make clear why unlearning does not solve our problem, recall the example of a model $f$ trained on skin images to predict disease. This time, to remove the corrupted medical image $x_p$ from $f$, person $p$ invokes the RTBF. Thus, the data controller for $f$ unlearns $x_p$, yielding $f_u$. But, even after unlearning, any medical insurance company can still access the publicly available $x_p$ and obtain $f_u(x_p)$. But, $f_u(x_p) = f(x_p)$, and thus the medical insurance company *incorrectly* labels person $p$ as high risk for medical coverage. Thus, the unfair and dangerous scenario for person $p$ remains. This holds similarly for unlearning methods which deal with corrupted or obsolete data, as they still do not aim to reduce confidence in the final prediction (Schoepf et al., 2025).

*Differences from Unlearning*: Importantly, what we propose is **not an unlearning algorithm**, which would need to be aligned with the goals of unlearning (and indistinguishability from retrain-from-scratch). Instead, we aim to address an entirely new threat scenario—test time privacy—that unlearning cannot solve, which we detail in Appx. A. For example, indistinguishability from retrain is inconsequential in our threat model. Furthermore, we also consider corrupted test examples, unlike unlearning which focuses only on the training dataset. Finally, what may constitute a privacy violation in unlearning, e.g. revealing that an instance is in the forget set via a membership inference attack (Shokri et al., 2017) does *not* constitute a violation in our threat model. This holds similarly for other threat models; for example, reconstruction attacks which lead to recovery of $x_p$ (Dinur and Nissim, 2003) or adversarial attacks which lead to misclassification of $x_p$ (Goodfellow et al., 2015) are not violations in our threat model, as explained in Appx. A.

Still, the privacy guarantees that we provide in the threat model of test-time privacy are complementary to the guarantees that unlearning can provide. However, related work has focused heavily on unlearning–we fill this gap by presenting a framework for test-time privacy.

**Additional Related Works**: Due to space constraints, we provide an additional related works section in Appx. C. Critically, we describe how differentially privacy methods like private aggregation of teacher ensembles (PATE) (Papernot et al., 2018) or label differential privacy (LabelDP) (Ghazi et al., 2021) differ from our setting, how label model inversion attacks (Zhu et al., 2019) relate to our threat model, and also why misclassification-based methods for unlearning (Cha et al., 2024) are suboptimal for addressing our threat model.

## 3  TEST-TIME PRIVACY THREAT MODEL

**Notation:** Let $\mathcal{X} \subset \mathbb{R}^d$ be a sample space and let $\mathcal{Y} \subset \mathbb{R}^o$ be a label space. Denote $\mathcal{Z} = \mathcal{X} \times \mathcal{Y}$ as the space of feature-label pairs. Let $\mathcal{Z}^n$ be the $n$-fold Cartesian product of $\mathcal{Z}$ such that a dataset $\mathcal{D} \subset \mathcal{Z}^n$ is a collection of $n$ feature-label pairs. Then, the $i$th instance is denoted as $\mathcal{D}^{(i)}$ with its feature in $\mathcal{X}$ being $\mathcal{D}^{(i,\mathcal{X})}$ and label being $\mathcal{D}^{(i,\mathcal{Y})}$. Following the unlearning literature, we subset $\mathcal{D}$ as a "forget set" $\mathcal{D}_f$, containing instances to protect, and a retain set $\mathcal{D}_r = \mathcal{D} \setminus \mathcal{D}_f$. Then, suppose we have a (randomized) learning algorithm $\mathcal{A} : \mathcal{Z}^n \to \mathcal{W}$, where $\mathcal{W} \subset \mathbb{R}^z$ is a parameter space. Let the set of hypotheses parameterized with respect to this parameter space be $\mathcal{H}_\mathcal{W}$. Let $f_{\boldsymbol{w}} \in \mathcal{H}_\mathcal{W}$ be the hypothesis parameterized by $\boldsymbol{w} \in \mathcal{W}$, defined as $f_{\boldsymbol{w}} : \mathcal{X} \to \Delta_{|\mathcal{Y}|}$, where $\Delta_{|\mathcal{Y}|}$ is the probability simplex $\{p_1, ..., p_\gamma : p_i \geq 0, \sum_{i=1}^{|\mathcal{Y}|} p_i = 1\}$. When $\boldsymbol{A}$ is a matrix, $||\boldsymbol{A}||_2$ is the 2 operator norm. When $\boldsymbol{v}$ is a vector, $||\boldsymbol{v}||_2$ is the $\ell_2$ norm. Furthermore, let $\lambda_{\min}(\boldsymbol{A})$ denote the minimum eigenvalue of $\boldsymbol{A}$. If we have an objective $f_A + f_B$, we denote its gradient evaluated at $\boldsymbol{w}$ as $\nabla_{\boldsymbol{w},A,B}$ and Hessian as $\boldsymbol{H}_{\boldsymbol{w},A,B}$. Finally, when for sets $\mathcal{S}, \mathcal{F} \subset \mathcal{N}$ and $\mathcal{R}$ and mechanisms $\mathcal{M}, \mathcal{M}' : \mathcal{N} \to \mathcal{R}$, we have $\Pr(\mathcal{M}(\mathcal{S}) \in \mathcal{R}) \leq e^\varepsilon \Pr(\mathcal{M}'(\mathcal{F}) \in \mathcal{R}) + \delta$ and $\Pr(\mathcal{M}'(\mathcal{S}) \in \mathcal{R}) \leq e^\varepsilon \Pr(\mathcal{M}(\mathcal{F}) \in \mathcal{R}) + \delta$, we will denote $\mathcal{M}(\mathcal{S}) \approx_{\varepsilon,\delta,\mathcal{R}} \mathcal{M}'(\mathcal{F})$. We provide a symbol table in Appx. N.

Following recent work (Baig and Pietrzak, 2025), we define our threat model as an informal *test-time privacy (TTP) game*. This section summarizes the full model detailed in Appx. A.

Firstly, the game involves three actors: a **data corrupter** (*c*) who creates a "forget set" $\mathcal{D}_f$ of corrupted instances; a benign **model provider** ($\tau$) who releases a model; and a **TTP adversary** ($\nu$) who aims to use the model to get a confident, harmful prediction $m$ on instances $\boldsymbol{x} \in \mathcal{D}_f$. Next, the provider $\tau$ trains a model $f$, is notified of the corrupted set $\mathcal{D}_f$, and then runs an algorithm $\mathcal{G}$ to release a protected model $\hat{f}$. We make two key assumptions:

**Assumption 3.1** (Open-Weight Access). *The adversary $\nu$ has full access to $\hat{f}$'s architecture and weights.*

**Assumption 3.2** ($\tau$-Limited Knowledge). *The provider $\tau$ knows $\mathcal{D}_f$ but does not know the specific harmful label $m$ the adversary seeks, nor do they know the adversary $\nu$.*

The adversary $\nu$ wins if they can leverage open-weight access to $\hat{f}$ to *confidently* recover the sensitive prediction $m$. The provider $\tau$ wins if $\hat{f}$ leaves $\nu$ uncertain about $m$. Given this, together with Asm. 3.2, the provider's optimal defense is to have $\hat{f}$ output maximal uncertainty (i.e. a uniform distribution) on $\mathcal{D}_f$ while maintaining accuracy on all other data. Critically, Asm. 3.1 renders simple defenses like output masking useless, as the adversary $\nu$ could just remove the mask. Therefore, a successful defense must perturb the model weights in a non-invertible manner, which motivates our approach in Sec. 4.

Finally, our formulation is general. While we may define $\mathcal{D}_f$ in terms of training data, this is done without loss of generality; the forget set can consist of any corrupted instances, whether from the training or test set. We provide simple TTP attacks to further motivate this threat model in Appx. B.

## 4  APPROACHES AND ALGORITHMS

In order to prevent test-time privacy violations, it suffices to have the model output a uniform distribution over the forget set, rendering the model maximally uncertain. Then, an adversary can only guess at the original sensitive prediction.[3] We also would like to preserve retain set accuracy; to that end, we present an algorithm that finetunes the pretrained model with a Pareto optimal objective. To make this algorithm concrete, we define a *uniform learner*, which we prove to exist in many common hypothesis classes. Then, we use this concept in order to construct a Pareto objective.

### 4.1  THE FINETUNING PARETO LEARNER

For a dataset $\mathcal{D} \subset \mathcal{Z}^n$, we denote the pretrained model as $\mathcal{A}(\mathcal{D})$. Then, to make $\mathcal{A}(\mathcal{D})$ uniform over the forget set, we introduce our core concept of a *uniform learner*:

---

[3]Please see Appx. A for a clear characterization of why this is optimal within our threat model, and why we can consider $\mathcal{D}_f \subset \mathcal{D}$, where $\mathcal{D}$ is a training dataset, without loss of generality.

**Definition 4.1** (Uniform learner). *Suppose we have a (randomized) learning algorithm $\mathcal{K} : \mathcal{Z}^n \to \mathcal{W}$ that, given $\mathcal{D} \subset \mathcal{Z}^n$, yields the parameter $\mathcal{K}(D) = \boldsymbol{w}_{\mathcal{D}}$. We say $\mathcal{K}$ is a uniform learner if $\forall \mathcal{D} \in \mathcal{Z}^n$, $\boldsymbol{w}_{\mathcal{D}}$ parametrizes $f_{\boldsymbol{w}_{\mathcal{D}}} \in \mathcal{H}_{\mathcal{W}}$ and satisfies:*

$$|| f_{\boldsymbol{w}_{\mathcal{D}}} - \underbrace{(\frac{1}{|\mathcal{Y}|}, ..., \frac{1}{|\mathcal{Y}|})}_{|\mathcal{Y}| \; times} ||_{\infty, \mathcal{X}} = 0. \tag{1}$$

That is, $\mathcal{K}$ is a uniform learner if its parameterized outputs yield the uniform distribution $U[0, |\mathcal{Y}|]$ for all inputs across all datasets. We define this as a learning algorithm for full generality to handle e.g. neural networks with nonlinear transformations in their last layer. Furthermore, $\mathcal{K}$ exists in many common hypothesis spaces, proved in Appx. H.2:

**Proposition 4.2.** *Suppose we have a hypothesis space $\mathcal{H}_{\mathcal{W}}$ consisting of functions where the ultimate layer is an affine transformation and the outputs are passed through a softmax. Let $\mathcal{K}$ be a uniform learner. Then, $f_{\mathcal{K}(\mathcal{D})} \in \mathcal{H}_{\mathcal{W}} \; \forall \mathcal{D} \subset \mathcal{Z}^n$.*

*Proof Sketch:* Setting the weights in the ultimate affine layer to 0 yields uniform outputs.

Most classifiers built and deployed in ML recently, including multilayer perceptrons (MLP), residual networks (ResNets) (He et al., 2016), and transformers (Vaswani et al., 2017) satisfy the premise of Prop. 4.2, making it widely applicable.

Next, we assume $\mathcal{A}$ and $\mathcal{K}$ are obtained through empirical risk minimization (ERM) to a local or global minima. That is, let $\mathcal{A}(\mathcal{D}) = \text{argmin}_{\boldsymbol{w} \in \mathcal{W}} \mathcal{L}_{\mathcal{A}}(\boldsymbol{w}, \mathcal{D})$ and $\mathcal{K}(\mathcal{D}) = \text{argmin}_{\boldsymbol{w} \in \mathcal{W}} \mathcal{L}_{\mathcal{K}}(\boldsymbol{w}, \mathcal{D})$, where $\mathcal{L}_{\mathcal{A}}$ penalizes incorrect classification and $\mathcal{L}_{\mathcal{K}}$ penalizes a lack of uniformity in model outputs. One choice of $\mathcal{L}_{\mathcal{K}}$ is the KL divergence (Kullback and Leibler, 1951) between the softmax outputs and the uniform distribution. This loss has been previously used to penalize highly confident classifier predictions (Pereyra et al., 2017); we thus adapt this loss to our setting. Furthermore, by Prop. 4.2, this loss can be completely minimized over $\mathcal{W}$.

Critically, we seek the optimal tradeoff between uniformity over the forget set and utility over the retain set. That is, we should produce a learner that is Pareto optimal with respect to $\mathcal{L}_{\mathcal{K}}$ and $\mathcal{L}_{\mathcal{A}}$. This learner can be characterized as follows:

**Proposition 4.3.** *Let $\theta \in (0, 1)$. Fix $\mathcal{D} \subset \mathcal{Z}^n$ and consider the forget set $\mathcal{D}_f \subset \mathcal{D}$ and the retain set $\mathcal{D}_r = \mathcal{D} \setminus \mathcal{D}_f$. Then, if $\mathcal{M}_{\theta}(\mathcal{D}) = \text{argmin}_{\boldsymbol{w} \in \mathcal{W}} \theta \mathcal{L}_{\mathcal{K}}(\boldsymbol{w}, \mathcal{D}_f) + (1 - \theta) \mathcal{L}_{\mathcal{A}}(\boldsymbol{w}, \mathcal{D}_r)$ is a global minimizer, it is globally Pareto optimal with respect to $\mathcal{L}_{\mathcal{K}}(\boldsymbol{w}, \mathcal{D}_f)$ and $\mathcal{L}_{\mathcal{A}}(\boldsymbol{w}, \mathcal{D}_r)$. Similarly, if $\mathcal{M}_{\theta}(\mathcal{D})$ is a local minimizer, it is locally Pareto optimal.*

*Proof Sketch:* This holds by contradiction. If the solution to the minimized objective was not globally (locally) Pareto optimal, since $\theta \in (0, 1)$, it could not be the global (local) minimizer. See Appx. H.3 for a full proof. Definitions of Pareto optimality are included in Appx. H.3 as well.

As shown in Prop. 4.3, $\mathcal{M}_{\theta}$ yields a parameter that, given $\theta \in (0, 1)$, presents a Pareto optimal tradeoff between uniformity over the forget set and utility over the retain set. One can adjust $\theta$ to vary over many Pareto optimal solutions, yielding different tradeoffs between uniformity and utility. This yields Alg. 1, in which we finetune a pretrained model by using it as initialization for $\mathcal{M}_{\theta}(\mathcal{D})$.

### 4.2 The Certified Pareto Learner

While the aforementioned algorithm in 4.1 provably guarantees an optimal tradeoff, so long as its objective is minimized, we would also like to make it *certified*, obtaining a certificate that a third party can inspect to verify test-time privacy. Thus, to design a certified approximation algorithm, we take inspiration from certified unlearning (Zhang et al., 2024), which aims to add a small amount of structured noise such that the pretrained model becomes indistinguishable from the retrained model. In our setting, we would like to make the pretrained model indistinguishable from the solution to the Pareto objective. To do so, we define a new notion of $(\varepsilon, \delta)$-indistinguishability and use this definition to design a certifiable algorithm, with results in Sec. 6.

Firstly, to motivate our definition, recall the definition of differential privacy (Dwork et al., 2006):

**Definition 4.4** (($\varepsilon, \delta$)-differential privacy). *Suppose we have privacy budgets $\varepsilon \in (0, 1)$ and $\delta > 0$. A randomized algorithm $\mathcal{M} : \mathcal{Z}^n \to \mathcal{W}$ satisfies $(\epsilon, \delta)$-differential privacy if $\forall \mathcal{T} \subset \mathcal{W}$ and $\forall \mathcal{D}, \mathcal{D}' \in \mathcal{Z}^n$ s.t. $||\mathcal{D} - \mathcal{D}'||_1 \leq 1$:*

---

**Algorithm 1** $\mathcal{M}_\theta$ Finetuning

---

**Require:** Dataset $\mathcal{D}$; forget set $\mathcal{D}_f$; pretrained model $\boldsymbol{w}^* = \mathcal{A}(\mathcal{D})$; uniformity-utility tradeoff coefficient $\theta$; $e$ epochs.
 Use $\boldsymbol{w}^*$ as initialization for the Pareto learner $\mathcal{M}_\theta(\mathcal{D})$.
 Optimize the Pareto learner $\mathcal{M}_\theta(\mathcal{D})$ for $e$ epochs with e.g. SGD to yield $\boldsymbol{w}^-$.
 **return** $\boldsymbol{w}^-$.

---

$$\mathcal{M}(\mathcal{D}) \approx_{\varepsilon,\delta,\mathcal{T}} \mathcal{M}(\mathcal{D}'). \tag{2}$$

This guarantees that the algorithm $\mathcal{M}$ applied on a dataset is statistically indistinguishable from the same algorithm applied on all datasets different by one instance. One can leverage this definition to formalize certified unlearning (Sekhari et al., 2021):

**Definition 4.5** (($\varepsilon, \delta$)-certified unlearning). *Suppose we have privacy budgets $\varepsilon \in (0,1)$ and $\delta > 0$. Consider $\mathcal{D} \subset \mathcal{Z}^n$ and let $\mathcal{D}_f \subset \mathcal{D}$ be the forget set to be unlearned, and $\mathcal{D}_r = \mathcal{D} \setminus \mathcal{D}_f$ be the retain set. $\mathcal{U} : \mathcal{Z}^n \times \mathcal{Z}^n \times \mathcal{W} \to \mathcal{W}$ is an ($\varepsilon, \delta$)-certified unlearning algorithm if $\forall \mathcal{T} \subset \mathcal{W}$, we have:*

$$\mathcal{U}(\mathcal{D}, \mathcal{D}_f, \mathcal{A}(\mathcal{D})) \approx_{\varepsilon,\delta,\mathcal{T}} \mathcal{A}(\mathcal{D}_r). \tag{3}$$

This formalizes making $\mathcal{A}(\mathcal{D})$ indistinguishable from $\mathcal{A}(\mathcal{D}_r)$. In light of Def. 4.5, we seek to make $\mathcal{A}(\mathcal{D})$ indistinguishable from $\mathcal{M}_\theta(\mathcal{D})$. Thus, we provide the following new definition:

**Definition 4.6** (($\varepsilon, \delta, \theta$)-certified Pareto learner). *Suppose we have privacy budgets $\varepsilon \in (0,1)$ and $\delta > 0$ with $\mathcal{D} \subset \mathcal{Z}^n$. Let $\mathcal{D}_f \subset \mathcal{D}$ be the forget set and let $\mathcal{D}_r = \mathcal{D} \setminus \mathcal{D}_f$ be the retain set. Suppose we have $\theta \in (0,1)$ and $\mathcal{M}_\theta(\mathcal{D}) = argmin_{\boldsymbol{w} \in \mathcal{W}} \theta \mathcal{L}_\mathcal{K}(\boldsymbol{w}, \mathcal{D}_f) + (1-\theta)\mathcal{L}_\mathcal{A}(\boldsymbol{w}, \mathcal{D}_r)$, where $\mathcal{K}$ is a uniform learner and $\mathcal{A}$ is a learning algorithm both obtained through ERM. An algorithm $\mathcal{G} : \mathcal{Z}^n \times \mathcal{Z}^n \times \mathcal{W} \to \mathcal{W}$ is a ($\varepsilon, \delta, \theta$)-certified Pareto learner if $\forall \mathcal{T} \subset \mathcal{W}$:*

$$\mathcal{G}(\mathcal{D}, \mathcal{D}_f, \mathcal{A}(\mathcal{D})) \approx_{\varepsilon,\delta,\mathcal{T}} \mathcal{M}_\theta(\mathcal{D}). \tag{4}$$

**Discussion:** Qualitatively, the conditions in Def. 4.6 mean that the model obtained by algorithm $\mathcal{G}$ is statistically indistinguishable from a model that is Pareto optimal between utility over the retain set and uniformity over the forget set. Here, we consider the classical setting of $\varepsilon \in (0,1)$.[4] Finally, note that satisfying Def. 4.5 and Def. 4.6 together is not possible for forget sets which overlap; thus, a model provider should adopt whichever approach corresponds to their threat model.

One way we can design an algorithm which satisfies Def. 4.6 is by taking a Newton step towards the Pareto model and applying structured Gaussian noise; this yields Alg. 2, which is certifiable as proved in Appx. G. Using local convex approximation (Nocedal and Wright, 1999), in which we add a regularization term to the objective of the Pareto learner, we design Alg. 2 without any assumptions of convexity on the component loss functions.

In addition, Alg. 2 requires inverting a Hessian, which is computationally infeasible for practical neural networks e.g. ResNets, even after employing conjugate gradient methods (Nocedal and Wright, 1999) and Hessian vector product techniques (Pearlmutter, 1994). To resolve this issue, we also propose a derived Alg. 3 in Appx. G, which computes an efficient estimator for the inverse Hessian (Agarwal et al., 2016). Furthermore, this algorithm does not assume convergence to a local minima for $\mathcal{A}(\mathcal{D})$, handling e.g. early stopping. An online version is presented in Appx. I as Alg. 4. While Alg. 2, 3 and 4 have more hyperparameters than Alg. 1, they offer a certificate which can be used to verify use of our method by a third party; we present ways to reduce hyperparameters in Appx. J.

## 5 Theoretical Analysis

In what follows, we aim to analyze various properties of Alg. 1 and Alg. 2 to understand how to appropriately choose $\theta$ and the privacy-utility tradeoffs these algorithms incur. To clarify the notation

---

[4]Notably, Balle and Wang (2018) provide a way to achieve ($\varepsilon, \delta$)-indistinguishability for $\varepsilon > 1$, and their technique can be adapted without loss of generality to our setting.

---

**Algorithm 2** $(\epsilon, \delta, \theta)$-Certified Uniformity with Exact Inverse Hessian

---

**Require:** Dataset $\mathcal{D}$; forget set $\mathcal{D}_f$; pretrained model $\boldsymbol{w}^* = \mathcal{A}(\mathcal{D})$; privacy budgets $\varepsilon$ and $\delta$; uniformity-utility tradeoff coefficient $\theta$; local convex coefficient $\lambda$; norm upper bound $C$.

$\tilde{\boldsymbol{w}} \leftarrow \boldsymbol{w}^* - (\boldsymbol{H}_{\boldsymbol{w}^*, \mathcal{K}, \mathcal{A}} + \lambda \boldsymbol{I})^{-1} \nabla_{\boldsymbol{w}^*, \mathcal{K}, \mathcal{A}}$.  // Derived in Appx. G.

Compute $\Delta$ as the bound in Eq. (G.20).

$\sigma = \frac{\Delta}{\epsilon} \sqrt{2 \ln(1.25/\delta)}$.

$\boldsymbol{w}^- \leftarrow \tilde{\boldsymbol{w}} + Y$ where $Y \sim \mathcal{N}(\boldsymbol{0}, \sigma^2 \boldsymbol{I})$.

**return** $\boldsymbol{w}^-$.

---

used in this section, we include a symbol table in Appx. N. Firstly, we seek to understand how we can choose $\theta$ to guarantee uniformity over the forget set. To do so, we provide a constraint to be satisfied to ensure uniformity. Then, we provide an appropriate lower bound on $\theta$ to ensure the constraint is satisfied. In doing so, one obtains a bound on the privacy of our algorithm. We next want to obtain a bound on the utility of our algorithm. To that end, we upper bound the difference between the retain loss of the locally optimal learned model $\mathcal{A}(\mathcal{D}_r)$ and the locally optimal solution to the Pareto objective $\mathcal{M}_\theta(\mathcal{D})$. We obtain a tight, non-vacuous bound with respect to $\theta$ and characterize it asymptotically. Incorporating the two bounds provides a concrete characterization of the privacy-utility tradeoffs that occur when our algorithms are used.

In particular, across all algorithms, we make the pretrained model indistinguishable from:

$$\mathcal{M}_\theta(\mathcal{D}) = \arg \min_{||\boldsymbol{w}||_2 \leq C, \boldsymbol{w} \in \mathcal{W}} \theta \mathcal{L}_\mathcal{K}(\boldsymbol{w}, \mathcal{D}_f) + (1 - \theta) \mathcal{L}_\mathcal{A}(\boldsymbol{w}, \mathcal{D}_r) + \frac{\lambda}{2} ||\boldsymbol{w}||_2^2 \quad (5)$$

$$= \arg \min_{||\boldsymbol{w}||_2 \leq C, \boldsymbol{w} \in \mathcal{W}} \theta \sum_{i=1}^{|\mathcal{D}_f|} \ell_\mathcal{K}(\boldsymbol{w}, \mathcal{D}_f^{(i)}) + (1 - \theta) \sum_{j=1}^{|\mathcal{D}_r|} \ell_\mathcal{A}(\boldsymbol{w}, \mathcal{D}_r^{(j)}) + \frac{\lambda}{2} ||\boldsymbol{w}||_2^2, \quad (6)$$

where $\ell_\mathcal{K}$ and $\ell_\mathcal{A}$ are component loss functions corresponding to individual data instances in the forget and retain sets, respectively. Note that $\lambda$ is present either as weight decay in the Pareto learning in Alg. 1 or as part of the local convex approximation in Alg. 2. Furthermore, note that the objective is constrained by $||\boldsymbol{w}||_2 \leq C$; we use this as a part of our local convex approximation when deriving Alg. 2; it is however unecessary for Alg. 1. Similarly, unlearning methods assume this either implicitly or explicitly (Zhang et al., 2024). One can use projected gradient descent (Nocedal and Wright, 1999) during pretraining to satisfy this constraint.

Note that Alg. 1 has $\lambda \approx 0$. For Alg. 2, by Lemma H.9, our models $f_{\boldsymbol{w}^-}$ and $f_{\mathcal{M}_\theta(\mathcal{D})}$ have approximately the same outputs over $\mathcal{D}_f$, where $\boldsymbol{w}^-$ are the weights after applying one of our TTP algorithms. Hence, for any of our algorithms, to ensure indistinguishability from uniformity over $\mathcal{D}_f$, it suffices to ensure that $\mathcal{M}_\theta$ satisfies the following constraint:

$$||f_{\mathcal{M}_\theta(\mathcal{D})}(\mathcal{D}_f) - U[0, |\mathcal{Y}|]||_\infty \leq \gamma. \quad (7)$$

We then have the following bound on Eq. (7) with respect to $\theta$, the proof of which is in Appx. H.10:

**Proposition 5.1.** *Let $\mathcal{M}_\theta(\mathcal{D})$ be the global solution to the Pareto objective. Choose, as surrogate losses, $\ell_\mathcal{K}(\boldsymbol{w}, \mathcal{D}_f^{(i)}) = D_{KL}(f_{\boldsymbol{w}}(\mathcal{D}_f^{(i)})||U[0, |\mathcal{Y}|])$, the KL divergence between the model outputs over the forget set and the uniform distribution, and $\ell_\mathcal{A}(\boldsymbol{w}, \mathcal{D}_r^{(j)}) = \mathbb{H}_{CE}(\mathcal{D}_r^{(j, \mathcal{Y})}, f_{\boldsymbol{w}}(\mathcal{D}_r^{(j, \mathcal{X})}))$, the cross entropy between model predictions and labels over the retain set. Then, $||f_{\mathcal{M}_\theta(\mathcal{D})}(\mathcal{D}_f) - U[0, |\mathcal{Y}|]||_\infty \leq \sqrt{2(\frac{1-\theta}{\theta}|\mathcal{D}_r| \ln |\mathcal{Y}|)}$.*

*Proof Sketch*: By using Prop. 4.2 and the fact that $\mathcal{M}_\theta(\mathcal{D})$ is a global minimizer, we can yield a bound on $\mathcal{L}_\mathcal{K}$. Then, standard inequalities yield our result.

Then, we can choose $\theta$ as follows to guarantee Eq. (7), the proof of which is in Appx. H.11:

**Corollary 5.2.** *For a $\gamma > 0$, choosing $\theta \geq \frac{2|\mathcal{D}_r| \ln |\mathcal{Y}|}{\gamma^2 + 2|\mathcal{D}_r| \ln |\mathcal{Y}|}$ guarantees that Eq. (7) holds.*

**Discussion**: Note that Cor. 5.2 is well-defined in that $\theta \in (0, 1)$ for any choice of $|\mathcal{D}_r|, |\mathcal{Y}|$ and $\varepsilon$. Furthermore, Cor. 5.2 restricts $\mathcal{M}_\theta(\mathcal{D})$ to a subset of Pareto optimal solutions, but this does not

render it no longer Pareto optimal; thus, our formulation as in Appx. G still holds in its entirety. Importantly, this is a sufficient *but not necessary* condition to satisfy Eq. (7).

Similarly, by Lemma H.9, we can study the affect of $\theta$ in $\mathcal{M}_\theta(\mathcal{D})$ on the (empirical) retain error on $\mathcal{D}_r$, after our algorithms are applied. To provide this bound, we require two key assumptions:

**Assumption 5.3.** *The gradients of $\ell_\mathcal{K}$ and $\ell_\mathcal{A}$ are Lipschitz in $\boldsymbol{w}$ with constants $\frac{P_\mathcal{K}}{|\mathcal{D}_f|}$ and $\frac{P_\mathcal{A}}{|\mathcal{D}_r|}$.*

**Assumption 5.4.** *The Hessians of $\ell_\mathcal{K}$ and $\ell_\mathcal{A}$ are Lipschitz in $\boldsymbol{w}$ with constants $\frac{F_\mathcal{K}}{|\mathcal{D}_f|}$ and $\frac{F_\mathcal{A}}{|\mathcal{D}_r|}$.*

**Discussion:** Note that these assumptions are only used to prove Thm. 5.5 and in Appx. G; we do not require them to prove all previously mentioned theorems. These assumptions, similar to those studied by Zhang et al. (2024), are less restrictive than those typically studied in certified unlearning Sekhari et al. (2021); importantly, we do not assume (strong) convexity of the losses.

We then present a tight, non-vacuous bound on the retain error after applying any of our algorithms:

**Theorem 5.5.** *Suppose Assumptions 5.3 and 5.4 hold, and let $P_\mathcal{K}, P_\mathcal{K}, F_\mathcal{K}, F_\mathcal{A}$ be as defined in Assumptions 5.3 and 5.4. Let $\alpha^* := \mathcal{L}_\mathcal{A}(\mathcal{A}(\mathcal{D}_r), \mathcal{D}_r)$ be the locally optimal (empirical) retain loss, achieved by $\mathcal{M}_\theta(\mathcal{D})$ when $\theta = 0$. Let $\alpha(\theta) := \mathcal{L}_\mathcal{A}(\mathcal{M}_\theta(\mathcal{D}), \mathcal{D}_r)$ be the locally optimal retain loss obtained by $\mathcal{M}_\theta(\mathcal{D})$ when $\theta \in (0, 1)$. Suppose all weights used throughout are bounded by $||\boldsymbol{w}||_2 \leq C$. Additionally, denote by $F := \theta M_\mathcal{K} + (1 - \theta)F_\mathcal{A}$ and $P := \theta P_\mathcal{K} + (1 - \theta)P_\mathcal{A}$. Consider regularization coefficient $\lambda \geq L + 2\theta CF + \sqrt{2\theta CF(P + 2\theta CF + 8P_\mathcal{K})}$. Then, we have the following bound:*

$$|\alpha^* - \alpha(\theta)| \leq \mathcal{O}(\lambda C^2\theta + C^2\theta^2). \tag{8}$$

*Proof Sketch*: We subtract the first order conditions, by definition of $\alpha^*$ and $\alpha(\theta)$, to get an expression with respect to the gradients; plugging in an equivalent path integral expression and applying Lemma H.2 yields our desired result, with a full proof (including the full bound) in Appx. H.12.

**Discussion**: Three key hyperparameters should be kept small to ensure high retain accuracy: the $\ell_2$ regularization coefficient $\lambda$, the max model weight magnitude $C$, and the Pareto frontier hyperparameter $\theta$. In particular, large regularization coefficients take the model off the Pareto frontier. However, smaller or sparser weights are preferred, since the bound grows quadratically in $C$.

In addition, that when $\theta = 0$, the bound simplifies to 0, indicating that it is tight near 0. We demonstrate that it is tight near 1 in Appx. L. Furthermore, in the case of Alg. 1, since $\lambda \approx 0$, we do not need the condition on $\lambda$ and obtain a clearer characterization. Furthermore, we can obtain a more concise bound with simpler techniques, but such a bound is vacuous and does not incorporate information about $\theta$; we elaborate on this in Appx. H.12.

## 6 EMPIRICAL ANALYSIS

Below, we provide empirical results for Alg. 1 and Alg. 2. We firstly discuss our experimental setup, baselines, and define our uniformity metric. Next, we provide our core results across Alg. 1, Alg. 2, and our baselines for several architectures and benchmarks. We also comment on the Pareto frontier of Alg. 1 and Alg. 2, providing additional insight into the structure of our problem.

**Setup and Baselines**. Our primary results on Alg. 1 are for ResNet50 (He et al., 2016) trained on SVHN, CIFAR10, and CIFAR100. We also provide results for logistic regression on MNIST to evaluate Alg. 2. We then include additional experiments with more complicated datasets and models, such as ViT (Dosovitskiy et al., 2021) and TinyImageNet (Le and Yang, 2015), in Tab. L.3. We compare results with the pretrained model and the model retrained without the forget set, which constitutes exact unlearning (Bourtoule et al., 2021). We also compare our methods to LabelDP (Ghazi et al., 2021) and a synthetic baseline that assigns random labels to instances neighboring the forget set. Across methods, we compare retain accuracy, test accuracy, and forget uniformity. We provide more details and the rationale for our baselines in Appx. K.

**Providing a Uniformity Metric**: We require a metric to compare uniformity over the forget set in an interpretable manner. Thus, we define the "confidence distance" as $\max\{0, f(\boldsymbol{x})_t - \frac{1}{|\mathcal{Y}|}\}$ for $\boldsymbol{x} \in \mathcal{D}_f$, where $f(\boldsymbol{x})_t$ is the max confidence score. In our experiments, we use this as the primary

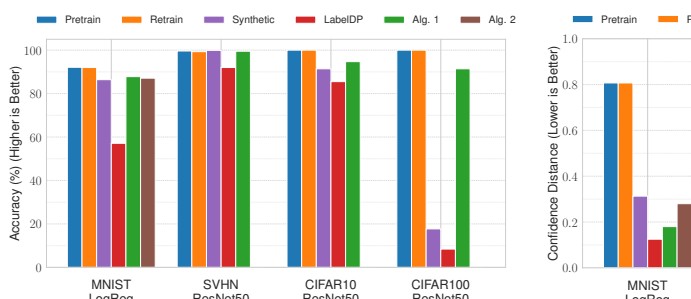

(a) Accuracy on retain set for baselines as well as Alg. 1 and Alg. 2 with $\theta = 0.75$.

(b) Confidence distance on forget set for baselines as well as Alg. 1 and Alg. 2 with $\theta = 0.75$.

Figure 2: Across datasets, observe a significant drop in confidence distance, where lower is better, for both our algorithms. We also observe that both algorithms provide strong accuracy on the retain set. We observe similar behavior for the test set in Appx. L, while the baselines are inconsistent. Variance is negligible for all metrics.

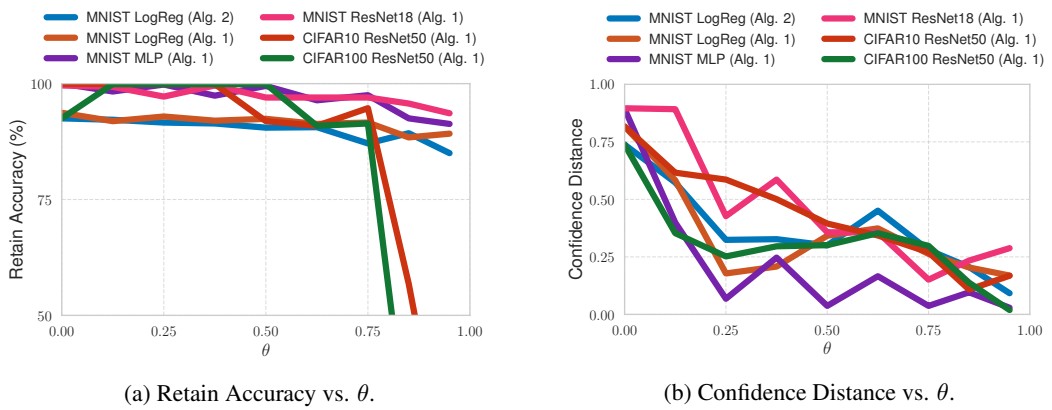

(a) Retain Accuracy vs. $\theta$.

(b) Confidence Distance vs. $\theta$.

Figure 3: From Fig. 3a, we observe that for simple datasets, the retain accuracy decreases smoothly. However, for larger datasets like CIFAR10 and CIFAR100 as one passes $\theta \approx 0.75$, retain accuracy drops significantly. This motivates our choice of $\theta = 0.75$ used throughout our experiments. In Fig. 3b we observe that the confidence distance decreases roughly linearly as $\theta$ increases.

metric for uniformity, reporting the average confidence distance over the forget set. We discuss why this is reasonable in Appx. D and compare it to alternative metrics in Appx. L.

**Overall Results:** The results for Alg. 1 are presented in Fig. 2, in which we were able to achieve a $> 3\times$ decrease in confidence distance with only a 0.01% and 0.04% decrease in retain and test accuracy, respectively, for a ResNet50 pretrained on SVHN. We obtain similar results for MNIST, CIFAR10, and CIFAR100: retain and test set accuracies remain high, while forget confidence distance is significantly reduced. Results for the test set are deferred to Appx. L. We additionally find that the synthetic baseline can induce uniformity well for SVHN, but can either fail to induce uncertainty entirely (CIFAR10) or induce uncertainty at great cost to retain and test accuracy (CIFAR100). We observe similar behavior for TinyImageNet in Appx. L.3. This holds similarly for LabelDP, which furthermore undesirably reduces the confidence distance on retain and test sets, while our method does not, as demonstrated in Tab. L.4. Furthermore, our observations coincide with Zhao et al. (2024), observing that unlearned models still produce confident predictions on deleted instances.

Furthermore, as illustrated by Fig. 2, we find that Alg. 2 also induces uniformity well, while marginally reducing retain and test accuracy. Thus, this algorithm produces a certificate through which test-time privacy can be verified while still obtaining a good privacy-utility tradeoff. For both algorithms, tables are included in Appx. L for completeness.

**Pareto Frontiers:** To better understand the structure of our problem, we explore the Pareto frontier in Fig. 3. We observe that for MNIST, CIFAR10, and CIFAR100, various $\theta$ can provide good retain

accuracy, albeit at the cost of uniformity. In general, we find that $\theta \approx 0.75$ offers a solid privacy-utility tradeoff. Thus, the $\varepsilon$ in Prop. 5.1 can be chosen fairly large while ensuring low confidence distance.

**Additional Experiments**: We conduct various additional experiments in in Appx. L and briefly comment about them here. Firstly, we obtain excellent performance for TinyImageNet and ViT in Appx. L.3. Secondly, as desired, we obtain obtain high confidence distances on the retain and test sets in Appx. L.4. Thirdly, we study the optimization dynamics of Alg. 1 in Appx. L.6, providing mathematical and empirical evidence for the necessity of early stopping in large models when using Alg. 1. Fourthly, we evaluate our method on several strong TTP attacks, demonstrating that we can still offer effective defense, especially when compared to pretraining or retraining, in Appx. L.7. Fifthly, in Appx. L.8, we find that we preserve strong accuracy and high confidence, as desired, on test instances which are nearest neighbors to the forget set instances. Thus, an adversary querying nearby instances outside of the forget set does not suffice to circumvent our algorithms. Sixthly, we find that we can induce uncertainty on forget instances which were not part of the original training dataset, while still preserving retain and test accuracies, in Appx. L.9. Seventhly, we provide ablations on the size of our forget set in Appx. L.10. Finally, we compare our confidence metric to an $\ell_2$ uniformity metric, finding that they highly correlate, in Appx. L.11.

## 7 DISCUSSION

We present *test-time privacy*, a threat model in which an adversary seeks to directly use a confident prediction for harm. This contrasts with existing work like PATE and LabelDP, which focus on protecting against model inversion and leakage of ground truth labels. To protect against a test-time privacy adversary, we present multiple algorithms to induce uniformity on a known corrupted subset while preserving utility on the rest of the data instances. This can be used to prevent adversaries from taking advantage of model outputs. Furthermore, we prove a privacy-utility tradeoff for our algorithms, providing a tight bound which is empirically verified. We hope our test-time privacy can further inspire the community to explore different threat models for sensitive data. Limitations and future directions are provided in Appx. F.

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

# Appendix

# Table of Contents

# A  FULL TEST-TIME PRIVACY THREAT MODEL

Following recent works on privacy and cybersecurity (Baig and Pietrzak, 2025), we begin by making our threat model concrete as an informal security game. Broadly, we consider a *test-time privacy (TTP) game* where a *TTP adversary* aims use an open-weight ML model $\hat{f}$ to produce a confident, harmful prediction $m$ for a specific set of corrupted inputs $\mathcal{D}_f$ drawn from a distribution $\mathcal{P}$.

**Actors and Assets**: The game begins with three key actors:

1. The data corrupter $c$, an entity that either maliciously or erroneously creates a "forget set" $\mathcal{D}_f$ of corrupted instances, e.g. a server which makes an error in compressing a medical image uploaded to an online forum.

2. The *model provider* $\tau$, a benign challenger that uses a learning algorithm $\mathcal{A}$ and releases a model $f$. For example, $f$ can classify skin disease from skin images. They then seek to ensure TTP by running algorithm $\mathcal{G}$ to obtain model $\hat{f}$.

3. The TTP adversary $\nu$ e.g. a potential medical insurance provider who has access to the architecture and parameters of $\hat{f}$ and aims to obtain harmful prediction $m$ on $\mathcal{D}_f$ to e.g. use as a warrant to reject insurance applicants.

We operate under two core assumptions:

**Assumption A.1** (Open-Weight Access). *The adversary $\nu$ has complete access to the model's architecture and weights.*

**Assumption A.2.** *$\tau$-Limited Knowledge The model provider $\tau$ is notified about the existence of a corrupted forget set $\mathcal{D}_f$, but does not know the specific harmful label $m$. Furthermore, they do not know the specific adversary $\nu$.*

**Remarks**: Specifically, Asm. A.1 renders naive defenses, like obtaining $\hat{f}$ by masking softmax outputs of $f$, useless, as an adversary can simply remove such a mask and recover prediction $m$. Furthermore, to make Asm. A.2 concrete, following the example provided in Sec. 1, the model provider $\tau$ does not know whether e.g. $\nu$ is a medical insurance company aiming to obtain a prediction of "Melanoma" to reject coverage or a defense attorney in a criminal case against a doctor aiming to obtain a prediction of "Benign" to clear a doctor of accusations of medical malpractice.

**Game**: The game is then played in the first round.

**Round 1**: The first round contains preliminary steps as follows:

- The corrupter $c$ corrupts the data and yields $\mathcal{D}_f$, which adversary $\nu$ gains access to e.g. through the public Internet.

- The model provider $\tau$ trains a model $f$ over instances from $\mathcal{P}$.

- The model provider $\tau$ is made aware that $\mathcal{D}_f$ contains corrupted instances, and seeks to protect them from a TTP adversary $\nu$.

**Round 2**: The second round contains the following steps:

1. The model provider $\tau$, who is aware of TTP, aims to provide a model $\hat{f}$ to replace $f$ such that $\hat{f}(x) \neq m$, where $m$ is the harmful prediction. However, they are *unaware of which prediction $m$ is*. $\tau$ thus runs an algorithm $\mathcal{G}$ with respect to $f$, $\mathcal{D}_f$, and a training dataset $\mathcal{D}$, which yields a new model $\hat{f}$.

2. The TTP adversary takes model $\hat{f}$ and attempts to obtain a confident prediction $m$ which serves as a warrant to endanger individuals e.g. to reject individuals from a health insurance provider because their image was classified as a high risk disease like melanoma.

**Win Conditions**: The TTP adversary $\nu$ wins if it can leverage the provided information about $\hat{f}$, including the model weights of $\hat{f}$, to *confidently* recover the sensitive prediction $m$ for any $x \in \mathcal{D}_f$, as they can then e.g. use this prediction as a warrant to reject people's insurance applications. The model provider $\tau$ wins if $\hat{f}$ leaves $\nu$ uncertain as to whether the prediction is $m$ or not.

Given this win condition, an algorithm $\mathcal{G}$ satisfies *test-time privacy* if the adversary can only guess at the model output for all instances in $\mathcal{D}_f$. Thus, it is optimal to induce maximal uncertainty over $\mathcal{D}_f$, since the model provider $\tau$ does not know beforehand which sensitive class results in a TTP violation. We make this concrete in Appx. C, where we discuss existing methods to misclassify and relabel training instances.

In particular, in the discriminative setting–which our work focuses on–it is optimal for model $f$ to output uniform softmax outputs over $\mathcal{D}_f$, while maintaining strong accuracy on all other instances. Furthermore, to defend against such a adversary with open-weight model access, one must perturb the model weights in a non-invertible manner, motivating our approaches detailed in Sec. 4.

**Importantly**, while in our formulation in Sec. 4 we define the forget set of corrupted instances in terms of the training dataset, we do so **without loss of generality**. As detailed previously, we assume that the forget set contains *all* corrupted instances, including instances outside of the training dataset that are known to be corrupted. Denoting the set of training forget set instances $\mathcal{D}_f^{\text{train}}$ and $\mathcal{D}_f^{\text{test}}$, we can thus let $\mathcal{D}_f = \mathcal{D}_f^{\text{train}} \cup \mathcal{D}_f^{\text{test}}$ and again consider $\mathcal{D} = \mathcal{D}_f \cup \mathcal{D}_r$; in this scenario, all formal definitions and statements throughout Sec. 4, Sec. 5, and elsewhere follow in the exact same manner. A concrete example of when instances outside of a training dataset can become relevant is credit score classification; one's credit score report can become corrupted, even if they are not in the training dataset, and one should be able to ask a credit bureau to remedy this to ensure that e.g. a loan officer does not incorrectly estimate their credit score.

Next, our work is motivated by *prevention* rather than *correction*. In particular, we assume that the adversary leverages the model $\hat{f}$ as a part of an automated system by which they make decisions about people, including those in the forget set $\mathcal{D}_f$. The goal of TTP is to prevent this automation from causing harm *before* the harm occurs. Thus thus eliminates the need for a costly, time consuming, and technically challenging appeals process.

Finally, in Appx. B, we present some simple TTP attacks on open-weight image classifiers to further motivate our threat model.

## B  DEFINING TEST-TIME PRIVACY ATTACKS

In what follows, in light of our threat model provided in Appx. A, we design some simple test-time privacy attacks to motivate our problem. We also include experiments on these attacks, and how Alg. 1 performs against them, in Appx. L.7.

Our first simple algorithm is to add a small amount of uniformly sampled Gaussian noise, presented in Alg. 5. We find that this is not very effective in increasing confidence distance, as demonstrated in Tab. L.8 and Tab. L.9. When it brings the confidence distances from low to moderate, the model is usually confidently wrong, as demonstrated in Tab. L.10.

One way to more optimally attack the TTP of a pretrained model is by finding instances in a $\phi$-ball around the forget set instances that maximize the prediction confidence. To design such an attack, suppose we have a pretrained classifier $f_{\boldsymbol{w}^*} : \mathcal{X} \to \Delta_{|\mathcal{Y}|}$. Here, $f_{\boldsymbol{w}}(\boldsymbol{x}) = \text{softmax}(\boldsymbol{z}(\boldsymbol{x}))$, for $\boldsymbol{x} \in \mathcal{X}$, where $\boldsymbol{z}$ is a vector of logits. For a forget set instance, we begin by adding a small amount of uniform noise to break symmetry and obtain a nonzero gradient. We then want to obtain the worst-case perturbation over the logits by solving the optimization problem:

$$\max_{\phi} \max_{j} \boldsymbol{z}_{\boldsymbol{w}^*}^{(j)}(\boldsymbol{x} + \phi), \tag{B.9}$$

$$\text{s.t. } ||\phi||_{\infty} \le \gamma. \tag{B.10}$$

Since the max function is not differentiable everywhere, we use LogSumExp to approximate it. Denote $\rho(f_{\boldsymbol{w}}(\boldsymbol{x} + \phi)) = \log \sum_{j=1}^{|\mathcal{Y}|} \exp\left(\boldsymbol{z}_{\boldsymbol{w}}^{j}(\boldsymbol{x} + \phi)\right)$. This yields the optimization problem:

$$\max_{\phi} \rho(f_{\boldsymbol{w}^*}(\boldsymbol{x} + \delta)), \tag{B.11}$$

$$\text{s.t. } ||\phi||_{\infty} \le \gamma. \tag{B.12}$$

Following the Fast Gradient Sign Method (FGSM), a simple attack used to generate adversarial examples (Goodfellow et al., 2015), we design an attack as Alg. 6. Intuitively, we take a single linear step towards maximizing the function. We design also design stronger attack based on Projected Gradient Descent (PGD) (Madry et al., 2018) as Alg. 7, taking 40 steps while incrementally maximizing the confidence function while projecting back to the ball around the original instance. Empirical results are in Appx. L.7.

## C  ADDITIONAL RELATED WORK

**Differential Privacy:**  Differential privacy has widely been studied in the ML community in order to ensure privacy-preservation (Chaudhuri and Monteleoni, 2008); (Chaudhuri et al., 2011); (Abadi et al., 2016); (Chua et al., 2024). There also exist methods to finetune pretrained models to satisfy differential privacy (Yu et al., 2022). Furthermore, there are also ways to aggregate label noise to preserve privacy (Papernot et al., 2018).

However, differential privacy is designed to address an entirely different threat model than ours. In particular, in the threat model of differential privacy, an adversary seeks to use model outputs to recover private information about data instance $x_p$ corresponding to person $p$ with e.g. a model inversion attack. A differentially private classifier generally results in confident, accurate predictions. This does not address our threat model, where an adversary may use confident model outputs to violate the privacy of person $p$ in a different manner, taking advantage of them directly to use as a warrant to cause harm to person $p$.

**Label Differential Privacy**: Similarly, our formulation differs from label differential privacy (LabelDP) (Ghazi et al., 2021), which seeks to protect an adversary from learning the true labels of the instances in the training data. Given an instance, even after computing $f(x_p)$, under LabelDP an adversary cannot be confident that $f(x_p) = y$. However, LabelDP is applied to the entire dataset; our threat model involves only a particular subset of the training data. Furthermore, we do not need to protect the user's ground truth label, necessarily. In our law enforcement example in Sec. 1, the agency does not care about the ground truth label. Instead, they want any confirmation such that they have a warrant to act adversarially towards person $p$; for this, a confident prediction by model $f$ suffices. Finally, LabelDP results in poor retain and test accuracy for larger datasets e.g. CIFAR100, as demonstrated in Fig. 2.

Furthermore, from the perspective of protecting the privacy of the labels themselves, rather than protecting against any confident prediction, Busa-Fekete et al. (2021) demonstrate that testing a model, trained with LabelDP, on the training dataset allows an adversary to recover the labels of the label-private data with high probability. Since our algorithms induce uniformity, an adversary cannot infer the correct forget set labels by testing the model on the training dataset; thus, we provide better privacy against this threat model than LabelDP as well. Wu et al. (2023) argue that, under this threat model where one seeks to protect the labels, any model that generalizes must leak the accurate labels when tested on the training data. However, as we demonstrate by inducing uniformity while maintaining high test accuracy, this only holds when the model is to be tested on the *entire* training data, not a *subset* of the training data (or other test instances which are known to be corrupted), as in our setting.

**Label Model Inversion Attacks**: Related to LabelDP are model inversion attacks to recover the ground truth labels, like gradient inversion (Zhang et al., 2022); (Zhu et al., 2019); (Zhao et al., 2020). Yet, these methods do not report the confidence values for the recovered labels. Thus, they do not constitute test-time privacy attacks within our threat model. Furthermore, by the same token as above, an adversary seeks to recover a confident prediction to use as a warrant, not necessarily the ground truth labels. Still, these methods could potentially be extended to test-time privacy attacks by reporting a confidence score for the recovered labels. We leave this to future work.

**Other Paradigms in Privacy**: Other paradigms in the privacy literature correspond to a notion of "test-time privacy" which differ from our threat model. For example, several works study defense against model inversion attacks as test-time privacy (Wang et al., 2019); (Xiao et al., 2020); (Sun et al., 2021); (Tran and Fioretto, 2023). However, this is a separate threat model from ours; the adversary already has access to the instance $x_p$ within our threat model.

**Misclassification & Relabeling in Machine Unlearning**: Recently, methods have emerged to finetune a model to misclassify rather than mimicking retraining from scratch (Cha et al., 2024). There are other similar relabeling methods in the debiasing literature which could be used for this purpose (Angelopoulos et al., 2025). However, these methods often achieve poor performance on the remaining training data and fail to provide protection against our threat model in all cases. In particular, a purposefully incorrect classification can also be used to endanger an individual. To make this concrete, following the medical insurance example in Sec. 1, suppose the model provider misclassified $\boldsymbol{x}_p$ as "Benign" instead of "Melanoma" to protect against the insurance provider. But, as it turns out, the model user is a medical facility instead of the insurance provider. Thus, doctors may miss out on a critical medical diagnosis for a person $p$, again resulting in harm. We adjusted our existing justification for this in Appendix C to make this more clear. Furthermore, in the binary classification case, if an adversary knows that $\boldsymbol{x}_p$ is in the forget set, they can recover the true $f(\boldsymbol{x}_p)$ by taking complements, if an unlearning method which seeks to induce misclassification is used. They can also use the information that learned representations are markedly different than other similar examples to understand the method used. Additionally, in the multiclass setting, an adversary can still take complements of this class, yielding a probability of recovering the true class which is significantly better than choosing uniformly at random. Instead, it is fairer and more robust to have an output that is maximally uncertain.

**Model Calibration and Confidence**: In our setting, we use the model softmax outputs to represent the adversary's confidence in the final prediction. However, some argue that this type of interpretation is incorrect, i.e. ML models are poorly calibrated (Guo et al., 2017). Still, this interpretation is common (Pearce et al., 2021), and thus a model user would likely rely on the softmax outputs as the confidence scores. We leave to inducing uncertainty over the calibrated outputs to future work.

## D  UNIFORMITY METRIC

The confidence distance quantifies the adversary's confidence in their final prediction, i.e. the difference between the argmax softmax score and the uniform softmax score. Importantly, our method aims to have the adversary lack confidence in their final prediction. Thus, our metric captures what we aim to measure and is interpretable, since it is minimized at 0.

Furthermore, confidence distance allows us to quantify how uncertain the model is without relying on accuracy, since a drop in forget set accuracy is not the goal of our formulation. Next, if the maximum confidence score is very close to the uniform distribution, the probability mass of the output distribution must be distributed over the other softmax outputs, clearly yielding that the higher our uniformity metric, the more confident our model is, and the lower our uniformity metric, the less confident our model is. Additionally, it takes the dataset into account; for example, in CIFAR10, one would expect a uniformity score of $\approx 0.2$ to be reasonable, as then the adversary can only be $\approx 30\%$ confident that they have a useful prediction. However, for CIFAR100, a uniformity score of $\approx 0.2$ is much better, as it implies that an adversary can only be $\approx 21\%$ confident that they have a useful prediction.

One objection to the use of this metric may be that it does not indicate uniformity if it is low. For example, on CIFAR10, one could have a confidence score of $0.2$, which yields that the max softmax output is $0.35$. There could be three other nonzero softmax outputs of $0.3, 0.3, 0.1, 0.05$; this clearly is not uniform. However, this ensures test-time privacy; a test-time privacy adversary now has little confidence in their prediction, even if they choose the first one, rendering their warrant for misuse of sensitive data useless.

We empirically compare our confidence distance metric to other similar metrics in Appx. L, finding that when our confidence metric is minimized, other metrics are minimized.

## E  TEST-TIME PRIVACY EXAMPLES

Here, we provide a set of examples of the TTP threat model:

**Health Insurance**: Suppose an open-weight medical imaging model $f$ is released, designed to perform multiclass classification of skin photos into categories like "Dysplastic Nevus", "Benign Keratosis", which are usually harmless, or serious classes like "Melanoma" (Sun et al., 2016). A

person $p$ posts a photo of a harmless birthmark on his arm to a public health forum to ask a question. During the upload, an e.g. server error or compression issue causes the image file to become corrupted, severely distorting the birthmark. This results in a photo $x_p$. Next, a health insurance startup decides to build risk profiles by scraping these public forums. They download the open-weight model $f$ to automatically screen images for potential health liabilities. When they feed $x_p$ into $f$, it confidently classifies $x_p$ as "Melanoma". This erroneous classification is then automatically added to person $p$'s risk profile, resulting person $p$ being unfairly denied coverage.

**Criminal Records**: Suppose a model $f$ is trained on criminal records to predict individual crime likelihood. Additionally, suppose the criminal record $x_p$ of a person $p$ is corrupted and publicly available. Then, $f(x_p)$ predicts that person $p$ is highly likely to commit crime. An adversarial law enforcement agency, or even a prospective employer, may ignore or be unaware of warnings about the data being corrupted, rendering a dangerous scenario for person $p$.[5] To make this clear, provide a figure similar to that of Fig. 1 at Fig. F.4.

**Mortgage Loans**: Suppose a model $f$ is trained on various items relevant to whether one receives a mortgage loan or not, like bank statements and past rent payments. Person $p$ has corrupted rent payment history $x_p$. Then, the bank runs model $f$ and obtains $f(x_p)$, which confidently says that $x_p$ is undeserving of a loan.

**Car Insurance**: Suppose a model $f$ is trained on one's history of car accidents. Person $p$ has corrupted car accident history $x_p$. Then, when applying for car insurance, the provider runs model $f$ and obtains $f(x_p)$, which confidently says that $x_p$ is undeserving of a loan.[6]

We provide an additional example in the generative setting as well:

**News Articles**: Consider a text-to-image generative model trained on a large dataset, including web data, which has web articles and associated images. A popular news site publishes an article about a businessperson, but mistakenly uses a picture of an unrelated individual $p$, $x_p$, as the header image. This creates a strong, albeit false, association between this person's likeness and the (perhaps negative) content of the article. When prompted with a string similar to the headline of the news article, the model generates an image (or a similar image) of person $p$, algorithmically cementing a false narrative about person $p$.

# F  LIMITATIONS AND FUTURE DIRECTIONS

Notably, our presented method only applies to classification. While we provide an example of TTP in the generative setting in Appx. E, we leave extending TTP to e.g. diffusion models (Song et al., 2021) to future work. This extension is nontrivial, since there are uncountably infinitely many instances to induce uniformity over. Furthermore, following the newspaper example given in Appx. E, the "forget set" is often a *concept* or *association*, not a discrete set of instances. This holds similarly for sequence-to-sequence generation (Vaswani et al., 2017). Furthermore, even in the discriminative setting, we focus our method on image classification. Extending our methods to the text setting, which is nontrivial due to discrete inputs, remains as future work.

From an algorithmic perspective, in Alg. 1, we use linear scalarization to design our objective (Hwang and Masud, 2012). One can instead design an objective using $\varepsilon$-constraints (Miettinen, 1999), which can then be solved by an augmented Lagrangian method (Nocedal and Wright, 1999). Furthermore, our certified algorithms, especially Alg. 3, are very sensitive to hyperparameters. Specifically, $\lambda$ and $J$ must be nearly precisely $|\lambda_{\min}|$ and $|\lambda_{\max} + \lambda|$. Otherwise, the estimator diverges aggressively. Furthermore, other Hessian estimators (Mehta et al., 2022) do not admit bounds which are necessary for our certified Gaussian mechanism. Finally, taking $\lambda$ large to ensure that the Hessian is p.d. and hence invertible, i.e. local convex approximation, is problematic in our multiobjective setting since it results in a step off the Pareto frontier. These limitations can be overcome by adopting recent Hessian-free approaches from certified unlearning (Qiao et al., 2025; Mu and Klabjan, 2025; Koloskova et al., 2025). However, adopting these approaches is nontrivial, since they rely on assumptions about the

---

[5]Recently, ML model providers have been involved in privacy cases involving criminal records (of Privacy Professionals, 2020), making this threat pertinent.

[6]Note that recent, the Department of Motor Vehicles in America has been selling driving records, making this threat pertinent (White, 2020).

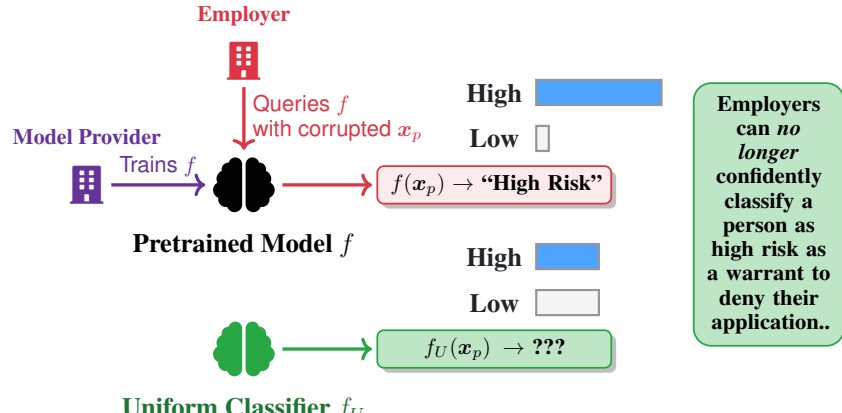

Figure F.4: An adversary, like an employer (🏢), can query a pretrained model $f$ (🧠) and use its outputs to make harmful decisions. However, after running our algorithm, the new model $f_U$ (🧠) provides maximal uncertainty, protecting against such an adversary. This is a duplicate of Fig. 1, to make clear how TTP extends to other settings.

retain-from-scratch (target) model which do not hold for the Pareto model, e.g. the structure of its gradient updates. As such, we leave this to future work.

Finally, while we proved an upper bound on the retain accuracy drop for our Pareto learner without assuming convexity in Sec. 5, the general theoretical qualities of TTP still remain to be studied. In particular, one could prove lower bounds on the generalization error or the sample complexity of TTP. This would allow us to design more principled TTP algorithms which e.g. achieve the minimax rate. However, obtaining such lower bounds is nontrivial. For example, current approaches which prove lower bounds for deletion capacity (sample complexity) in the related setting of certified unlearning (Allouah et al., 2025) rely heavily on the fact that $(\varepsilon, \delta)$-unlearning aims to mimic $\mathcal{A}(\mathcal{D}_r)$, whereas $(\varepsilon, \delta)$-TTP aims to mimic $\mathcal{M}_\theta(\mathcal{D})$. Furthermore, these works assume strongly convex losses, making them inapplicable to the deep learning setting. We thus leave this to future work.

## G DESIGNING CERTIFIED ALGORITHMS

In what follows, we design $(\varepsilon, \delta, \theta)$-certified Pareto learners. A symbol table can be found at Appx. N.

In our setting, the original model is obtained using ERM over some loss function $\mathcal{L}_\mathcal{A}$, some dataset $\mathcal{D}$, and some parameter space $\mathcal{W}$. Furthermore, we consider the common scenario where the cumulative loss $\mathcal{L}_\mathcal{A}$ over the dataset is a finite sum of individual losses $\ell_\mathcal{A}$. Thus, we denote the pretrained model as:

$$\boldsymbol{w}^* = \mathcal{A}(D) := \arg\min_{\boldsymbol{w} \in \mathcal{W}} \mathcal{L}_\mathcal{A}(\boldsymbol{w}, \mathcal{D}) = \arg\min_{\boldsymbol{w} \in \mathcal{W}} \sum_{i=1}^{|\mathcal{D}|} \ell_\mathcal{A}(\boldsymbol{w}, \mathcal{D}^{(i)}). \tag{G.13}$$

By Prop. 4.2, we can similarly obtain a uniform learner through ERM with respect to some loss function $\mathcal{L}_\mathcal{K}$. Furthermore, in our setting, we have the forget set $\mathcal{D}_f$ and retain set $\mathcal{D}_r = \mathcal{D} \setminus \mathcal{D}_f$. Thus, the uniform learner over the forget set can be characterized as:

$$\mathcal{K}(\mathcal{D}_f) := \arg\min_{\boldsymbol{w} \in \mathcal{W}} \mathcal{L}_\mathcal{K}(\boldsymbol{w}, \mathcal{D}) = \sum_{i=1}^{|\mathcal{D}_f|} \ell_\mathcal{K}(\boldsymbol{w}, \mathcal{D}_f^{(i)}). \tag{G.14}$$

Let $\theta \in (0, 1)$ be a tradeoff parameter between uniformity over the forget set and utility over the retain set. This yields a concrete characterization of $\mathcal{M}_\theta$ as:

$$\tilde{\boldsymbol{w}}^* = \mathcal{M}_\theta(D) := \arg\min_{\boldsymbol{w}\in\mathcal{W}} \theta\mathcal{L}_\mathcal{K}(\boldsymbol{w},\mathcal{D}_f) + (1-\theta)\mathcal{L}_\mathcal{A}(\boldsymbol{w},\mathcal{D}_r), \tag{G.15}$$

$$= \arg\min_{\boldsymbol{w}\in\mathcal{W}} \theta\sum_{i=1}^{|\mathcal{D}_f|} \ell_\mathcal{K}(\boldsymbol{w},\mathcal{D}_f^{(i)}) + (1-\theta)\sum_{i=1}^{|\mathcal{D}_r|} \ell_\mathcal{A}(\boldsymbol{w},\mathcal{D}_r^{(i)}). \tag{G.16}$$

as in Eq. (6).

To design an algorithm which takes in $\mathcal{D}, \mathcal{D}_r$, and $\boldsymbol{w}^*$ and outputs a parameter which satisfies Def. 4.6, we follow the methodology of certified unlearning Zhang et al. (2024), which seeks to satisfy Def. 4.5.

First, we simplify the problem of deriving a model that satisfies Def. 4.6:

**Theorem G.1.** *(Certification Guarantee) Let $\tilde{\boldsymbol{w}} := \mathcal{F}(\mathcal{D},\mathcal{D}_f,\boldsymbol{w}^*)$ be an approximation to $\tilde{\boldsymbol{w}}^*$. Suppose $||\tilde{\boldsymbol{w}} - \tilde{\boldsymbol{w}}^*||_2 \le \Delta$. Then, $\mathcal{U}(\mathcal{D},\mathcal{D}_f,A(D)) = \boldsymbol{w}^- = \tilde{\boldsymbol{w}} + Y$ is a $(\epsilon,\delta,\theta)$ certified uniformity algorithm, where $Y \sim \mathcal{N}(0,\sigma^2\boldsymbol{I})$ and $\sigma \ge \frac{\Delta}{\epsilon}\sqrt{2\ln(1.25/\delta)}$.*

*Proof:* See Appx. H.4.

Thus, it then suffices to find an approximation of $\tilde{\boldsymbol{w}}^*$, i.e. a form for $\mathcal{F}(\mathcal{D},\mathcal{D}_f,\boldsymbol{w}^*)$ and its associated $\Delta$. To do so, we consider the two assumptions Asm. 5.3 and Asm. 5.4.

For any $\boldsymbol{w} \in \mathcal{W}$, denote $\nabla_{\boldsymbol{w},\mathcal{K},\mathcal{A}} := \nabla_{\boldsymbol{w}}(\theta\mathcal{L}_\mathcal{K}(\boldsymbol{w},\mathcal{D}_f) + (1-\theta)\mathcal{L}_\mathcal{A}(\boldsymbol{w},\mathcal{D}_r))$, the gradient of the objective of $\mathcal{M}_\theta$ with respect to $\boldsymbol{w}$, and $\boldsymbol{H}_{\boldsymbol{w},\mathcal{K},\mathcal{A}} := \nabla_w^2(\theta\mathcal{L}_\mathcal{K}(\boldsymbol{w},\mathcal{D}_f) + (1-\theta)\mathcal{L}_\mathcal{A}(\boldsymbol{w},\mathcal{D}_r))$, the Hessian of the objective of $\mathcal{M}_\theta$ with respect to $\boldsymbol{w}$. We thus have $\nabla_{\boldsymbol{w},\mathcal{K},\mathcal{A}} = \theta\nabla_{\boldsymbol{w},\mathcal{K}} + (1-\theta)\nabla_{\boldsymbol{w},\mathcal{A}}$, and similarly for the Hessian.

Next, letting $g(\boldsymbol{w}) := \nabla_{\boldsymbol{w},\mathcal{K},\mathcal{A}}$, by Taylor's theorem, expanding $g(\tilde{\boldsymbol{w}}^*)$ around $\boldsymbol{w}^*$, we have that:

$$g(\tilde{\boldsymbol{w}}^*) \approx g(\boldsymbol{w}^*) + Dg|_{\boldsymbol{w}^*}(\tilde{\boldsymbol{w}}^* - \boldsymbol{w}^*). \tag{G.17}$$

.

Note that $g(\tilde{\boldsymbol{w}}^*) = 0$, since $\tilde{\boldsymbol{w}}^*$ is the minimizer of the objective in $\mathcal{M}_\theta$. Isolating $\tilde{\boldsymbol{w}}^*$ and using the definition of $g$, we then have that:

$$\tilde{\boldsymbol{w}}^* \approx \boldsymbol{w}^* - \boldsymbol{H}_{\boldsymbol{w}^*,\mathcal{K},\mathcal{A}}^{-1}\nabla_{\boldsymbol{w}^*,\mathcal{K},\mathcal{A}}. \tag{G.18}$$

.

Thus, we let $\tilde{\boldsymbol{w}} = \boldsymbol{w}^* - \boldsymbol{H}_{\boldsymbol{w}^*,\mathcal{K},\mathcal{A}}^{-1}\nabla_{\boldsymbol{w}^*,\mathcal{K},\mathcal{A}}$. This yields the following general form of $\Delta$:

**Proposition G.2.** *Suppose Asm. 5.3 and Asm. 5.4 hold. Suppose $\tilde{\boldsymbol{w}} = \boldsymbol{w}^* - \boldsymbol{H}_{\boldsymbol{w}^*,\mathcal{K},\mathcal{A}}^{-1}\nabla_{\boldsymbol{w}^*,\mathcal{K},\mathcal{A}}$. Then,*

$$||\tilde{\boldsymbol{w}}^* - \tilde{\boldsymbol{w}}||_2 \le \frac{\theta F_\mathcal{K} + (1-\theta)F_\mathcal{A}}{2}||\boldsymbol{H}_{\boldsymbol{w}^*,\mathcal{K},\mathcal{A}}^{-1}||_2||\boldsymbol{w}^* - \tilde{\boldsymbol{w}}^*||_2^2. \tag{G.19}$$

*Proof*: See Appx. H.5.

We then use local convex approximation (Nocedal and Wright, 1999) to bound $||\boldsymbol{H}_{\boldsymbol{w}^*,\mathcal{K},\mathcal{A}}^{-1}||_2$. To that end, we let the objective of $\mathcal{M}_\theta$ have a regularization term $\frac{\lambda}{2}||\boldsymbol{w}||_2^2$, yielding the inverse Hessian $||(\boldsymbol{H}_{\boldsymbol{w}^*,\mathcal{K},\mathcal{A}} + \lambda\boldsymbol{I})^{-1}||_2$; thus, in Prop. G.2, the norm of the inverse Hessian is replaced by $||(\boldsymbol{H}_{\boldsymbol{w}^*,\mathcal{K},\mathcal{A}} + \lambda\boldsymbol{I})^{-1}||_2$. It then suffices to bound this term.

Additionally, note that since the objective of $\mathcal{M}_\theta$ is nonconvex, the Hessian may not be invertible, i.e. $\lambda_{\min}(\boldsymbol{H}_{\boldsymbol{w}^*,\mathcal{K},\mathcal{A}}) < 0$. However, $\lambda_{\min}(\boldsymbol{H}_{\boldsymbol{w}^*,\mathcal{K},\mathcal{A}} + \lambda\boldsymbol{I}) = \lambda_{\min}(\boldsymbol{H}_{\boldsymbol{w}^*,\mathcal{K},\mathcal{A}}) + \lambda$. Thus, for $\lambda$ sufficiently large, we can make $\boldsymbol{H}_{\boldsymbol{w}^*,\mathcal{K},\mathcal{A}} + \lambda\boldsymbol{I}$ positive definite and hence invertible, resolving this issue. In particular, we can take $\lambda > ||\boldsymbol{H}_{\boldsymbol{w}^*,\mathcal{K},\mathcal{A}}||_2$.

Furthermore, we let $||\boldsymbol{w}||_2 \le C$ in $\mathcal{M}_\theta$ and $\mathcal{A}$, i.e. $\mathcal{M}_\theta = \arg\min_{||\boldsymbol{w}||_2\le C,\boldsymbol{w}\in\mathcal{W}} \theta\mathcal{L}_\mathcal{K}(\boldsymbol{w},\mathcal{D}_f) + (1-\theta)\mathcal{L}_\mathcal{A}(\boldsymbol{w},\mathcal{D}_A) + \frac{\lambda}{2}||\boldsymbol{w}||_2^2$ and $\mathcal{A}(\mathcal{D}) = \arg\min_{\boldsymbol{w}\in\mathcal{W},||\boldsymbol{w}||\le C} \mathcal{L}_\mathcal{A}(\boldsymbol{w},\mathcal{D})$. Note that, as mentioned in (Zhang et al., 2024), unlearning methods implicitly assume this.

Together, these two methods yield a tractable form of $\Delta$:

**Proposition G.3.** *Suppose Asm. 5.3 and Asm. 5.4 hold. Suppose $\tilde{\boldsymbol{w}} = \boldsymbol{w}^* - (\boldsymbol{H}_{\boldsymbol{w}^*,\mathcal{K},\mathcal{A}} + \lambda\boldsymbol{I})^{-1}\nabla_{\boldsymbol{w}^*,\mathcal{K},\mathcal{A}}$ where $||\boldsymbol{w}^*||_2, ||\tilde{\boldsymbol{w}}^*||_2 \leq C$. Let $\lambda_{\min} := \lambda_{\min}(\boldsymbol{H}_{\boldsymbol{w}^*,\mathcal{K},\mathcal{A}})$. Suppose $\lambda > ||\boldsymbol{H}_{\boldsymbol{w}^*,\mathcal{K},\mathcal{A}}||_2$. Then,*

$$||\tilde{\boldsymbol{w}}^* - \tilde{\boldsymbol{w}}||_2 \leq \frac{2C((\theta M_K + (1-\theta)M_A)C + \lambda)}{\lambda + \lambda_{\min}}. \tag{G.20}$$

*Proof:* See Appx. H.6.

While Prop. G.3 does yield a form of $\mathcal{F}$ and $\Delta$, the computation of $\tilde{\boldsymbol{w}}$ requires obtaining the exact inverse Hessian, which has runtime $\mathcal{O}(dz^2 + z^3)$, where $z$ is the number of learnable parameters. Furthermore, computing the gradient product with the inverse Hessian is $\mathcal{O}(z^2)$. Finally, computing the gradient $\nabla_{\boldsymbol{w}^*,\mathcal{K},\mathcal{A}}$ is $\mathcal{O}(|\mathcal{D}|z)$. Thus, the algorithm yielded by Prop. G.3 has a runtime complexity of $\mathcal{O}(dz^2 + z^3 + z^2 + |\mathcal{D}|z)$.

If we consider the additional assumption of convexity, we can take $\lambda$ very small to ensure the Hessian is invertible, since we have $\lambda_{\min} = 0$. Thus, for convex models e.g. logistic regression with a mean-square uniform loss, this is tractable. This yields Alg. 2.

However, for nonconvex models e.g. large scale neural networks, this is computationally intractable. Thus, to provide better runtime, we derive an asymptotically unbiased estimator of the inverse Hessian. However, the estimator in Zhang et al. (2024) does not trivially extend to our case. In particular, we cannot glean Hessian samples using sampled i.i.d. data from the retain set, because the Hessian in our setting is defined over the forget set as well. Thus, we must derive an unbiased estimator while sampling Hessians from *both* the retain and forget set. As such, following the techniques of (Agarwal et al., 2016), we design an unbiased estimator as follows:

**Theorem G.4.** *Suppose we have $n$ i.i.d. data samples $(X_1, ..., X_n)$ drawn from $D_f$ and $D_r$, uniformly at random, with probabilities $\theta$ and $1 - \theta$ respectively. Then, suppose $||\boldsymbol{H}_{\boldsymbol{w}^*,\mathcal{K},\mathcal{A}} + \lambda I||_2 \leq J$. For $t = 1, ..., n$, if $X_t \sim D_f$ let $\boldsymbol{H}_{t,\lambda} = \boldsymbol{H}_{\boldsymbol{w}^*,\mathcal{K},t} + \frac{\lambda I}{2\theta}$ and if $X_t \sim D_r$ let $\boldsymbol{H}_{t,\lambda} = \boldsymbol{H}_{\boldsymbol{w}^*,\mathcal{A},t} + \frac{\lambda I}{2(1-\theta)}$. Suppose $\lambda > ||\boldsymbol{H}_{\boldsymbol{w}^*,\mathcal{K},\mathcal{A}}||_2$. Then, compute:*

$$\tilde{\boldsymbol{H}}_{t,\lambda}^{-1} = \boldsymbol{I} + (\boldsymbol{I} - \frac{\boldsymbol{H}_{t,\lambda}}{J})\tilde{\boldsymbol{H}}_{t-1,\lambda}^{-1}, \ \ \tilde{\boldsymbol{H}}_{0,\lambda}^{-1} = \boldsymbol{I}. \tag{G.21}$$

*Then, $\frac{\tilde{\boldsymbol{H}}_{n,\lambda}^{-1}}{J}$ is an asymptotically unbiased estimator for $(\boldsymbol{H}_{\boldsymbol{w}^*,\mathcal{K},\mathcal{A}} + \lambda\boldsymbol{I})^{-1}$*

*Proof:* See Appx. H.7

One simple choice of $J$ is $J = 2\lambda$, by Lemma H.1. However, we let $J$ be free. The computation of the estimator in theorem G.4 has a runtime complexity of $\mathcal{O}(nz^2)$, a great speedup over the original $\mathcal{O}(dz^2 + z^3)$. Furthermore, with Hessian vector product (HVP) techniques (Pearlmutter, 1994), we obtain a space complexity of $\mathcal{O}(z)$ instead of $\mathcal{O}(z^2)$, since we do not have to compute the sample Hessians explicitly. Furthermore, computing $\tilde{\boldsymbol{H}}_{n,\lambda}^{-1}\nabla_{\boldsymbol{w},\mathcal{K},\mathcal{A}}$ recursively reduces $\mathcal{O}(z^2)$ to $\mathcal{O}(nz)$.

Additionally, following Agarwal et al. (2016), we can average $b$ unbiased estimators $\frac{\tilde{\boldsymbol{H}}_{t,\lambda}^{-1}}{J}$ as $\frac{1}{b}\sum_{i=1}^{b}\frac{\tilde{\boldsymbol{H}}_{t,\lambda}^{-1,(i)}}{J}$ to achieve better concentration. Altogether, we achieve a final runtime complexity of $\mathcal{O}(bnz^2 + bnz + |\mathcal{D}|z)$.

Furthermore, we relax the assumption that $\boldsymbol{w}^*$ and $\tilde{\boldsymbol{w}}^*$ are the global minimizers of $\mathcal{L}_\mathcal{A}$ and $\theta\mathcal{L}_\mathcal{A} + (1-\theta)\mathcal{L}_\mathcal{K}$. We do so because, in practice, it is possible that the data controller trained their model with early stopping, i.e. they did not reach the global minimizer. Altogether, this yields a final form of $\Delta$ as:

**Theorem G.5.** *Let $\tilde{\boldsymbol{w}}^*$ and $\boldsymbol{w}^*$ not be empirical risk minimizers of their respective losses, but rather approximations thereof. Suppose Asm. 5.3 and Asm. 5.4 hold. Suppose $||\boldsymbol{w}^*||_2, ||\tilde{\boldsymbol{w}}^*||_2 \leq C$. Let $\lambda_{\min} := \lambda_{\min}(\boldsymbol{H}_{\boldsymbol{w}^*,\mathcal{K},\mathcal{A}})$. Suppose $\lambda > ||\boldsymbol{H}_{\boldsymbol{w}^*,\mathcal{K},\mathcal{A}}||_2$. Let $\tilde{\boldsymbol{w}} = \boldsymbol{w}^* - \frac{\tilde{\boldsymbol{H}}_{t,\lambda}^{-1}}{J}\nabla_{\boldsymbol{w}^*,\mathcal{K},\mathcal{A}}$. Let*

---

**Algorithm 3** $(\epsilon, \delta, \theta)$-Certified Uniformity with Inverse Hessian Estimator

---

**Require:** Dataset $\mathcal{D}$; forget set $\mathcal{D}_f$; pretrained model $\boldsymbol{w}^* = \mathcal{A}(\mathcal{D})$; privacy budgets $\epsilon$ and $\delta$; uniformity-utility tradeoff coefficient $\theta$; estimator concentration $b$; sample size $n$; local convex coefficient $\lambda$; norm upper bound $C$; cumulative Hessian upper bound $H$; individual Hessian minimum eigenvalue upper bound $\zeta_{\min}$; gradient norm upper bound $G$; bound looseness probability $\rho$.

$\boldsymbol{P}_{0,\lambda}^{(0)} \leftarrow \nabla_{\boldsymbol{w}^*, \mathcal{K}, \mathcal{A}}$
**for** $j = 1, ..., b$ **do**
    **for** $t = 1, ..., n$ **do**
        Sample $X_t$ from $D_f$ uniformly with probability $\theta$ or,
        sample $X_t$ from $D_r$ uniformly with probability $1 - \theta$ .
        **if** $X_t \sim D_f$ **then**
            $\boldsymbol{H}_{t,\lambda}^{(j)} \leftarrow \nabla_{\boldsymbol{w}}^2 \mathcal{L}_{\mathcal{K}}(\boldsymbol{w}^*, X_i) + \frac{\lambda \boldsymbol{I}}{2\theta}$.
        **else if** $X_t \sim D_r$ **then**
            $\boldsymbol{H}_{t,\lambda}^{(j)} \leftarrow \nabla_{\boldsymbol{w}}^2 \mathcal{L}_{\mathcal{A}}(\boldsymbol{w}^*, X_i) + \frac{\lambda \boldsymbol{I}}{2(1-\theta)}$.
        **end if**
        $\boldsymbol{P}_{t,\lambda}^{(j)} = \boldsymbol{P}_{0,\lambda}^{(0)} + (\boldsymbol{I} - \frac{\boldsymbol{H}_{t,\lambda}^{(j)}}{H})\boldsymbol{P}_{t-1,\lambda}^{(j)}$.
    **end for**
**end for**
$\boldsymbol{P}_{n,\lambda} \leftarrow \frac{1}{b} \sum_{j=1}^{b} \boldsymbol{P}_{n,\lambda}^{(j)}$.
$\tilde{\boldsymbol{w}} \leftarrow \boldsymbol{w}^* - \frac{\boldsymbol{P}_{n,\lambda}}{H}$.
Compute $\Delta$ as the bound in Eq. (G.23).
$\sigma = \frac{\Delta}{\epsilon}\sqrt{2\ln(1.25/\delta)}$
$\boldsymbol{w}^- \leftarrow \tilde{\boldsymbol{w}} + Y$ where $Y \sim \mathcal{N}(\boldsymbol{0}, \sigma^2 \boldsymbol{I})$.
**return** $\boldsymbol{w}^-$.

---

$b$ *be the number of inverse Hessian estimators we average. Letting* $n$ *be the number of steps taken during unbiased estimation of the inverse Hessian, require* $n \geq 2\frac{B}{\lambda + \lambda_{\min}} \ln(\frac{B}{\lambda + \lambda_{\min}} b)$ *where* $B = \max\{\frac{\theta P_{\mathcal{K}} + \lambda}{|\mathcal{D}_f|}, \frac{(1-\theta)P_{\mathcal{A}} + \lambda}{|\mathcal{D}_r|}\}$. *Suppose* $||\nabla_{\boldsymbol{w}^*, \mathcal{K}, \mathcal{A}}||_2, ||\nabla_{\tilde{\boldsymbol{w}}^*, \mathcal{K}, \mathcal{A}}||_2 \leq G$, *With probability larger than* $1 - \rho$, *we have that:*

$$||\tilde{\boldsymbol{w}}^* - \tilde{\boldsymbol{w}}||_2 \leq \frac{2C((\theta F_{\mathcal{K}} + (1-\theta)F_{\mathcal{A}})C + \lambda) + G}{\lambda + \lambda_{\min}} \tag{G.22}$$

$$+ (16\frac{B}{\zeta_{\min}}\sqrt{\frac{\ln(\frac{d}{\rho})}{b}} + \frac{1}{16})(2C(\theta P_{\mathcal{K}} + (1-\theta)P_{\mathcal{A}}) + G). \tag{G.23}$$

*where* $\zeta_{\min} \geq \min_i \lambda_{\min}(\nabla_{\boldsymbol{w}}^2 \tilde{\ell}_{\mathcal{K}, \mathcal{A}}(\boldsymbol{w}, \mathcal{D}^{(i)}))$.

*Proof:* See Appx. H.8.

Note that if we let $\boldsymbol{w}^*$ be an ERM in theorem G.5, we can use $\nabla_{\boldsymbol{w}, \mathcal{K}, \mathcal{A}}$ and obtain the same result. Altogether, this yields Alg. 3.

# H PROOFS

## H.1 HELPFUL LEMMAS

**Lemma H.1.** *Given Asm. 5.3, the gradients* $\nabla_{\boldsymbol{w}, \mathcal{K}}$ *and* $\nabla_{\boldsymbol{w}, \mathcal{A}}$ *exist and are Lipschitz with constants* $P_{\mathcal{K}}$ *and* $P_{\mathcal{A}}$, *respectively. Furthermore, given Asm. 5.4, the Hessians* $\boldsymbol{H}_{\boldsymbol{w}, \mathcal{K}}$ *and* $\boldsymbol{H}_{\boldsymbol{w}, \mathcal{A}}$ *exist and are Lipschitz with constants* $F_{\mathcal{K}}$ *and* $F_{\mathcal{A}}$, *respectively.*

*Proof.*

$$||\nabla_{\boldsymbol{w}_1,\mathcal{K}} - \nabla_{\boldsymbol{w}_2,\mathcal{K}}||_2 = ||\sum_{i=1}^{|\mathcal{D}_f|} \nabla^2 \ell_{\mathcal{K}}^{(i)}(\boldsymbol{w}_1, \mathcal{D}_f^{(i)}) - \sum_{i=1}^{|\mathcal{D}_f|} \nabla^2 \ell_{\mathcal{K}}^{(i)}(\boldsymbol{w}_2, \mathcal{D}_f^{(i)})||_2 \quad \text{(H.24)}$$

$$\leq \sum_{i=1}^{|\mathcal{D}_f|} ||\nabla^2 \ell_{\mathcal{K}}^{(i)}(\boldsymbol{w}_1, \mathcal{D}_f^{(i)}) - \nabla^2 \ell_{\mathcal{K}}^{(i)}(\boldsymbol{w}_2, \mathcal{D}_f^{(i)})||_2, \text{ triangle inequality} \quad \text{(H.25)}$$

$$\leq \sum_{i=1}^{|\mathcal{D}_f|} \frac{P_{\mathcal{K}}}{|\mathcal{D}_f|} ||\boldsymbol{w}_1 - \boldsymbol{w}_2||_2, \text{Asm. 5.3} \quad \text{(H.26)}$$

$$= P_{\mathcal{K}} ||\boldsymbol{w}_1 - \boldsymbol{w}_2||_2 \quad \text{(H.27)}$$

This follows similarly for $\nabla_{\boldsymbol{w},\mathcal{A}}$, $\boldsymbol{H}_{\boldsymbol{w},\mathcal{K}}$, and $\boldsymbol{H}_{\boldsymbol{w},\mathcal{A}}$. $\qquad\square$

**Lemma H.2.** *Given Asm. 5.3, for any dataset $\mathcal{D} \subset \mathcal{Z}^n$, $\mathcal{L}_A$ satisfies:*

$$|\mathcal{L}_A(\boldsymbol{w}_1, \mathcal{D}) - \mathcal{L}_A(\boldsymbol{w}_2, \mathcal{D})| \leq \frac{P_{\mathcal{K}}}{2}||\boldsymbol{w}_1 - \boldsymbol{w}_2||_2^2 + ||\nabla_{\boldsymbol{w}_2,\mathcal{A}}||_2 ||\boldsymbol{w}_1 - \boldsymbol{w}_2||_2 \quad \text{(H.28)}$$

*Proof.* By the fundamental theorem of calculus, we have the path integral:

$$\int_0^1 \langle \nabla_{\boldsymbol{w}_2 + t(\boldsymbol{w}_1 - \boldsymbol{w}_2),\mathcal{A}}, \boldsymbol{w}_1 - \boldsymbol{w}_2 \rangle dt = \mathcal{L}_A(\boldsymbol{w}_1, \mathcal{D}) - \mathcal{L}_A(\boldsymbol{w}_2, \mathcal{D}) \quad \text{(H.29)}$$

We have that:

$$\int_0^1 \langle \nabla_{\boldsymbol{w}_2 + t(\boldsymbol{w}_1 - \boldsymbol{w}_2),\mathcal{A}}, \boldsymbol{w}_1 - \boldsymbol{w}_2 \rangle dt = \int_0^1 \langle \nabla_{\boldsymbol{w}_2,\mathcal{A}} - \nabla_{\boldsymbol{w}_2,\mathcal{A}} + \nabla_{\boldsymbol{w}_2 + t(\boldsymbol{w}_1 - \boldsymbol{w}_2),\mathcal{A}}, \boldsymbol{w}_1 - \boldsymbol{w}_2 \rangle dt$$
$$\text{(H.30)}$$

$$= \int_0^1 \langle \nabla_{\boldsymbol{w}_2,\mathcal{A}}, \boldsymbol{w}_1 - \boldsymbol{w}_2 \rangle dt \quad \text{(H.31)}$$

$$+ \int_0^1 \langle \nabla_{\boldsymbol{w}_2 + t(\boldsymbol{w}_1 - \boldsymbol{w}_2),\mathcal{A}} - \nabla_{\boldsymbol{w}_2,\mathcal{A}}, \boldsymbol{w}_1 - \boldsymbol{w}_2 \rangle dt \quad \text{(H.32)}$$

The first term can be bounded by Cauchy-Schwarz as:

$$\int_0^1 \langle \nabla_{\boldsymbol{w}_2,\mathcal{A}}, \boldsymbol{w}_1 - \boldsymbol{w}_2 \rangle dt \leq ||\nabla_{\boldsymbol{w}_2,\mathcal{A}}||_2 ||\boldsymbol{w}_1 - \boldsymbol{w}_2||_2 \quad \text{(H.33)}$$

and similarly the second term can be bounded by Cauchy-Schwarz as:

$$\int_0^1 \langle \nabla_{\boldsymbol{w}_2 + t(\boldsymbol{w}_1 - \boldsymbol{w}_2),\mathcal{A}} - \nabla_{\boldsymbol{w}_2,\mathcal{A}}, \boldsymbol{w}_1 - \boldsymbol{w}_2 \rangle dt \leq \int_0^1 ||\nabla_{\boldsymbol{w}_2 + t(\boldsymbol{w}_1 - \boldsymbol{w}_2),\mathcal{A}} - \nabla_{\boldsymbol{w}_2,\mathcal{A}}||_2 ||\boldsymbol{w}_1 - \boldsymbol{w}_2||_2 dt$$
$$\text{(H.34)}$$

$$\leq \int_0^1 P_{\mathcal{K}} t ||\boldsymbol{w}_1 - \boldsymbol{w}_2||_2, \text{by Lemma H.1} \quad \text{(H.35)}$$

$$\leq \frac{P_{\mathcal{A}}}{2} ||\boldsymbol{w}_1 - \boldsymbol{w}_2||_2 \quad \text{(H.36)}$$

Incorporating these bounds into Eq. (H.32) and Eq. (H.29), upon applying the triangle inequality, yields:

$$|\mathcal{L}_A(\boldsymbol{w}_1, \mathcal{D}) - \mathcal{L}_{\mathcal{A}}(\boldsymbol{w}_2, \mathcal{D})| \leq \frac{P_{\mathcal{K}}}{2}||\boldsymbol{w}_1 - \boldsymbol{w}_2||_2^2 + ||\nabla_{\boldsymbol{w}_2, \mathcal{A}}||_2 ||\boldsymbol{w}_1 - \boldsymbol{w}_2||_2 \tag{H.37}$$

as desired.

$\square$

**Lemma H.3.** *Given Asm. 5.4, the Hessians $\boldsymbol{H}_{\boldsymbol{w}, \mathcal{K}}$, $\boldsymbol{H}_{\boldsymbol{w}, \mathcal{A}}$, and $\boldsymbol{H}_{\boldsymbol{w}, \mathcal{K}, \mathcal{A}}$ are symmetric.*

*Proof.* By Lemma H.1, the Hessians $\boldsymbol{H}_{\boldsymbol{w}, \mathcal{K}}$ and $\boldsymbol{H}_{\boldsymbol{w}, \mathcal{A}}$ are continuous, and thus $\boldsymbol{H}_{\boldsymbol{w}, \mathcal{K}, \mathcal{A}}$ is continuous by linearity. Hence, all second-order partial derivatives contained in the Hessians are continuous, so by Schwartz's theorem all Hessians are symmetric. Importantly, for e.g. $\boldsymbol{H}_{\boldsymbol{w}, \mathcal{K}, \mathcal{A}}$, $||\boldsymbol{H}_{\boldsymbol{w}, \mathcal{K}, \mathcal{A}}||_2 = \max_i |\lambda_i(\boldsymbol{H}_{\boldsymbol{w}, \mathcal{K}, \mathcal{A}})|$, where $\lambda_i$ denotes the $i$th eigenvalue. $\square$

**Lemma H.4.** *(Corollary of Theorem A.1 in (Dwork et al., 2014)) Let $X \sim \mathcal{N}(\lambda, \sigma^2 \boldsymbol{I})$ and $\mathcal{Y} \sim \mathcal{N}(\lambda', \sigma^2 \boldsymbol{I})$. Suppose $||\lambda - \lambda'||_2 \leq \Delta$. Then for any $\delta > 0$, $X$ and $Y$ are $(\varepsilon, \delta)$-indistinguishable if $\sigma \geq \frac{\Delta}{\varepsilon}\sqrt{2 \ln(1.25/\delta)}$.*

**Lemma H.5.** *Suppose we have $n$ i.i.d. data samples $(X_1, ..., X_n)$ drawn from $D_f$ and $D_r$ with probabilities $\theta$ and $1 - \theta$ respectively. For $t = 1, ..., n$, if $X_t \sim D_f$ let $\boldsymbol{H}_{t, \lambda} = \boldsymbol{H}_{\boldsymbol{w}^*, \mathcal{K}, t} + \frac{\lambda I}{2\theta}$ and if $X_t \sim D_r$ let $\boldsymbol{H}_{t, \lambda} = \boldsymbol{H}_{\boldsymbol{w}^*, \mathcal{A}, t} + \frac{\lambda I}{2(1-\theta)}$. Then, $\mathbb{E}[\boldsymbol{H}_{t, \lambda}] = \boldsymbol{H}_{\boldsymbol{w}^*, \mathcal{K}, \mathcal{A}} + \lambda I$ i.e. $\boldsymbol{H}_{t, \lambda}$ is an unbiased estimator of our Hessian of interest.*

*Proof.* At time $t$, we have sample $X_t$ s.t. $X_t \sim D_r$ or $X_t \sim D_f$. Note that $\mathbb{E}_{X_t \sim D_f}[\boldsymbol{H}_{\boldsymbol{w}^*, \mathcal{K}, t} + \frac{\lambda I}{2\theta}] = \boldsymbol{H}_{\boldsymbol{w}^*, \mathcal{K}} + \frac{\lambda I}{2\theta}$, and likewise $\mathbb{E}_{X_t \sim D_r}[\boldsymbol{H}_{\boldsymbol{w}^*, \mathcal{A}, t} + \frac{\lambda I}{2(1-\theta)}] = \boldsymbol{H}_{\boldsymbol{w}^*, \mathcal{A}} + \frac{\lambda I}{2(1-\theta)}$.

By the law of iterated expectation, we have that:

$$\mathbb{E}[\boldsymbol{H}_{t, \lambda}] = \mathbb{E}[\boldsymbol{H}_{t, \lambda} | X_t \sim \mathcal{D}_f] \Pr(X_t \sim \mathcal{D}_f) + \mathbb{E}[\boldsymbol{H}_{t, \lambda} | X_t \sim \mathcal{D}_r] \Pr(X_t \sim \mathcal{D}_r)$$

$$= \theta \mathbb{E}_{X_t \sim D_f}[\boldsymbol{H}_{\boldsymbol{w}^*, \mathcal{K}, t} + \frac{\lambda I}{2\theta}] + (1 - \theta)\mathbb{E}_{X_t \sim D_r}[\boldsymbol{H}_{\boldsymbol{w}^*, \mathcal{A}, t} + \frac{\lambda I}{2(1 - \theta)}]$$

$$= \theta(\boldsymbol{H}_{\boldsymbol{w}^*, \mathcal{K}} + \frac{\lambda I}{2\theta}) + (1 - \theta)(\boldsymbol{H}_{\boldsymbol{w}^*, \mathcal{A}} + \frac{\lambda I}{2(1 - \theta)})$$

$$= \theta \boldsymbol{H}_{\boldsymbol{w}^*, \mathcal{K}} + (1 - \theta)\boldsymbol{H}_{\boldsymbol{w}^*, \mathcal{A}} + \lambda I$$

$$= \boldsymbol{H}_{\boldsymbol{w}^*, \mathcal{K}, \mathcal{A}} + \lambda \boldsymbol{I}$$

as desired. $\square$

**Lemma H.6.** *Suppose Assumptions 5.3 and 5.4 hold. Let local condition number $\hat{\kappa}_l$ and maximum local condition number $\hat{\kappa}_l^{\max}$ correspond to the definitions of Agarwal et al. (2016) with respect to the Hessian of the loss of $\mathcal{M}_\theta$ after local convex approximation. Then, $\hat{\kappa}_l \leq \frac{B}{\lambda + \lambda_{\min}}$ and $\hat{\kappa}_l^{\max} \leq \frac{B}{\zeta_{\min}}$ where $B = \max\{\frac{\theta P_{\mathcal{K}} + \lambda}{|\mathcal{D}_f|}, \frac{(1-\theta)P_{\mathcal{A}} + \lambda}{|\mathcal{D}_r|}\}$ and where $\zeta_{\min} \geq \min_i \lambda_{\min}(\nabla_{\boldsymbol{w}}^2 \tilde{\ell}_{\mathcal{K}, \mathcal{A}}^{(i)}(\boldsymbol{w}, \mathcal{D}^{(i)}))$.*

*Proof.* By Eq. (6) and our local convex approximation technique, we have that:

$$M_\theta(D) = \arg \min_{\boldsymbol{w} \in \mathcal{W}} \theta \sum_{i=1}^{|\mathcal{D}_f|} \ell_{\mathcal{K}}^{(i)}(\boldsymbol{w}, \mathcal{D}_f^{(i)}) + (1 - \theta) \sum_{i=1}^{|\mathcal{D}_r|} \ell_{\mathcal{A}}^{(i)}(\boldsymbol{w}, \mathcal{D}_r^{(i)}) + \frac{\lambda}{2}||\boldsymbol{w}||_2^2 \tag{H.38}$$

$$= \arg \min_{w \in \mathcal{W}} \sum_{i=1}^{|D_f|}(\theta \ell_{\mathcal{K}}^{(i)}(\boldsymbol{w}, \mathcal{D}_f^{(i)}) + \frac{\lambda}{2|\mathcal{D}_f|}||\boldsymbol{w}||_2^2) + \sum_{i=1}^{|D_r|}(\theta \ell_{\mathcal{A}}^{(i)}(\boldsymbol{w}, \mathcal{D}_r^{(i)}) + \frac{\lambda}{2|\mathcal{D}_r|}||\boldsymbol{w}||_2^2)$$

$$\tag{H.39}$$

$$= \arg \min_{w \in \mathcal{W}} \sum_{i=1}^{|D|} \tilde{\ell}_{\mathcal{K}, \mathcal{A}}^{(i)}(\boldsymbol{w}, \mathcal{D}^{(i)}) \tag{H.40}$$

where

$$\tilde{\ell}_{\mathcal{K},\mathcal{A}}^{(i)}(\boldsymbol{w}, \mathcal{D}^{(i)}) = \begin{cases} \theta \ell_{\mathcal{K}}^{(i)}(\boldsymbol{w}, \mathcal{D}_f^{(i)}) + \frac{\lambda}{2|\mathcal{D}_f|}||\boldsymbol{w}||_2^2, \ 1 \leq i \leq |\mathcal{D}_f| \\ (1-\theta)\ell_{\mathcal{K}}^{(i-|\mathcal{D}_f|)}(\boldsymbol{w}, \mathcal{D}_r^{(i-|\mathcal{D}_f|)}) + \frac{\lambda}{2|\mathcal{D}_r|}||\boldsymbol{w}||_2^2, \ |\mathcal{D}_f| + 1 \leq i \leq |\mathcal{D}| \end{cases} \tag{H.41}$$

By the definitions provided in Agarwal et al. (2016), we have:

$$\hat{\kappa}_l = \max_{\boldsymbol{w} \in \mathcal{W}} \frac{\max_i \lambda_{\max}(\nabla_{\boldsymbol{w}}^2 \tilde{\ell}_{\mathcal{K},\mathcal{A}}(\boldsymbol{w}, \mathcal{D}^{(i)}))}{\lambda_{\min}(\boldsymbol{H}_{\boldsymbol{w},\mathcal{K},\mathcal{A}} + \lambda \boldsymbol{I})} \tag{H.42}$$

and

$$\hat{\kappa}_l^{\max} = \max_{\boldsymbol{w} \in \mathcal{W}} \frac{\max_i \lambda_{\max}(\nabla_{\boldsymbol{w}}^2 \tilde{\ell}_{\mathcal{K},\mathcal{A}}(\boldsymbol{w}, \mathcal{D}^{(i)}))}{\min_i \lambda_{\min}(\nabla_{\boldsymbol{w}}^2 \tilde{\ell}_{\mathcal{K},\mathcal{A}}(\boldsymbol{w}, \mathcal{D}^{(i)}))} \tag{H.43}$$

We then have that, for any $i$,

$$\lambda_{\max}(\nabla_{\boldsymbol{w}}^2 \tilde{\ell}_{\mathcal{K},\mathcal{A}}^{(i)}) \leq ||\nabla_{\boldsymbol{w}}^2 \tilde{\ell}_{\mathcal{K},\mathcal{A}}^{(i)}||_2, \text{ by Lemma H.3} \tag{H.44}$$

$$= \max\{||\theta \nabla_{\boldsymbol{w}}^2 \ell_{\mathcal{K}}^{(i)} + \frac{\lambda \boldsymbol{I}}{|\mathcal{D}_f|}||_2, ||(1-\theta)\nabla_{\boldsymbol{w}}^2 \ell_{\mathcal{A}}^{(i)} + \frac{\lambda \boldsymbol{I}}{|\mathcal{D}_r|}||_2\} \tag{H.45}$$

$$\tag{H.46}$$

Furthermore, by Asm. 5.3 and the triangle inequality:

$$||\theta \nabla_{\boldsymbol{w}}^2 \ell_{\mathcal{K}}^{(i)} + \frac{\lambda \boldsymbol{I}}{|\mathcal{D}_f|}||_2 \leq \frac{\theta P_{\mathcal{K}} + \lambda}{|\mathcal{D}_f|} \tag{H.47}$$

and

$$||(1-\theta)\nabla_{\boldsymbol{w}}^2 \ell_{\mathcal{A}}^{(i)} + \frac{\lambda \boldsymbol{I}}{|\mathcal{D}_r|}||_2 \leq \frac{(1-\theta) P_{\mathcal{A}} + \lambda}{|\mathcal{D}_r|} \tag{H.48}$$

Taking max over all $i$, we obtain that

$$\max_i \lambda_{\max}(\nabla_{\boldsymbol{w}}^2 \tilde{\ell}_{\mathcal{K},\mathcal{A}}(\boldsymbol{w}, \mathcal{D}^{(i)})) \leq \max\{\frac{\theta P_{\mathcal{K}} + \lambda}{|\mathcal{D}_f|}, \frac{(1-\theta) P_{\mathcal{A}} + \lambda}{|\mathcal{D}_r|}\} \tag{H.49}$$

which we denote by $B$.

Then, we obtain that $\hat{\kappa}_l \leq \frac{B}{\lambda + \lambda_{\min}}$ and $\hat{\kappa}_l^{\max} \leq \frac{B}{\zeta_{\min}}$ as desired, since

$\square$

**Lemma H.7.** *(Lemma 3.6 adapted from Agarwal et al. (2016)) Suppose Asm. 5.3 and Asm. 5.4 hold. Consider the estimator $\frac{\tilde{\boldsymbol{H}}_{n,\lambda}^{-1}}{H}$ in theorem G.4. Let $b$ be the number of inverse Hessian estimators we obtain. Suppose $n \geq 2\frac{B}{\lambda + \lambda_{\min}} \ln(\frac{B}{\lambda + \lambda_{\min}} b)$, where $B = \max\{\frac{\theta P_{\mathcal{K}} + \lambda}{|\mathcal{D}_f|}, \frac{(1-\theta) P_{\mathcal{A}} + \lambda}{|\mathcal{D}_r|}\}$. Then, we have that:*

$$\Pr[||\boldsymbol{H}_{\boldsymbol{w}^*,\mathcal{K},\mathcal{A}} + \lambda \boldsymbol{I})^{-1} - \frac{\tilde{\boldsymbol{H}}_{n,\lambda}^{-1}}{H}||_2 \leq 16\frac{B}{\zeta_{\min}}\sqrt{\frac{\ln(\frac{d}{\rho})}{b}} + \frac{1}{16}] \geq 1 - \rho \tag{H.50}$$

*where $\zeta_{\min} \geq \min_i \lambda_{\min}(\nabla_{\boldsymbol{w}}^2 \tilde{\ell}_{\mathcal{K},\mathcal{A}}^{(i)}(\boldsymbol{w}, \mathcal{D}^{(i)}))$.*

*Proof.* Note that $b = S_1$ in our setting. In our setting, following the subsequent steps of the proof in Agarwal et al. (2016) after plugging in the bounds in Lemma H.6 in place of $\hat{\kappa}_l, \hat{\kappa}_l^{\max}$, noting that we choose $n = S_2 \geq 2\frac{B}{\lambda + \lambda_{\min}} \ln(\frac{B}{\lambda + \lambda_{\min}} b)$, we obtain the exact same result for the Neumann series bound of $\frac{1}{16}$. Using the fact that $\frac{B}{\zeta_{\min}}$ is an upper bound on $\hat{\kappa}_l^{\max}$ by Lemma H.6, the rest of the proof follows similarly. $\qquad \square$

**Lemma H.8.** *(Proposition 2.1 in Dwork et al. (2014)) Let $\mathcal{M} : \mathbb{N}^{|\mathcal{X}|} \to R$ be a randomized algorithm that is $(\varepsilon, \delta)$-differentially private. Let $f : R \to R'$ be an arbitrary mapping. Then, $f \circ \mathcal{M} : \mathbb{N}^{|\mathcal{X}|} \to R'$ is $(\varepsilon, \delta)$-differentially private.*

Note that, in the proof of Lemma H.8, one proves this fact for deterministic mappings, so this holds for both randomized and deterministic $f$.

**Lemma H.9.** *Consider the mapping $\mathcal{J} : \mathcal{W} \to R$, and suppose $\mathcal{G} : \mathcal{Z}^n \times \mathcal{Z}^n \times \mathcal{W} \to \mathcal{W}$ satisfies Def. 4.6. Then, $\forall C \subset R$:*

$$\Pr(\mathcal{J}(\mathcal{G}(\mathcal{D}, \mathcal{D}_f, \mathcal{A}(\mathcal{D}))) \in C) \leq e^{\varepsilon} \Pr(\mathcal{J}(\mathcal{M}_\theta(\mathcal{D})) \in C) + \delta \qquad (\text{H.51})$$
$$\Pr(\mathcal{J}(\mathcal{M}_\theta(\mathcal{D})) \in C) \leq e^{\varepsilon} \Pr(\mathcal{J}(\mathcal{G}(\mathcal{D}, \mathcal{D}_f, \mathcal{A}(\mathcal{D}))) \in C) + \delta \qquad (\text{H.52})$$

*Proof.* Immediate from Lemma H.8. $\qquad \square$

## H.2 PROOF OF PROPOSITION 4.2

*Proof.* Fix a dataset $\mathcal{D} \subset \mathcal{Z}^n$.

Suppose we have an $K$−layer function $f_{\boldsymbol{w}} : \mathbb{R}^d \to \mathbb{R}^o$ parameterized by $\boldsymbol{w} \in \mathcal{W}$ of the form $f(\boldsymbol{x}) = L_1 \circ ... \circ L_{K-1} \circ L_K$ where $L_{K-1}(\boldsymbol{x}) = \boldsymbol{W}_{K-1}^T \boldsymbol{x} + \boldsymbol{b}_{K-1}$ and $L_K(\boldsymbol{x}) = \text{softmax}(\boldsymbol{x})$, i.e. $L_{K-1}(\boldsymbol{x})_i = \frac{e^{x_i}}{\sum_{j=1}^{|\mathcal{Y}|} e^{x_j}}$. Thus, $f_{\boldsymbol{w}} \in \mathcal{H}_{\mathcal{W}}$. Then, let $\boldsymbol{W}_{K-1} = \boldsymbol{0}$ and $\boldsymbol{b}_{K-1} = \boldsymbol{0}$.

Fix $\boldsymbol{z} \in \mathcal{D}$. This yields, for $j = 1, ..., |\mathcal{Y}|$, $f(\boldsymbol{z})_j = \frac{e^0}{\sum_{j=1}^{|\mathcal{Y}|} e^0} = \frac{e^0}{|\mathcal{Y}| e^0} = \frac{1}{|\mathcal{Y}|}$. Hence, since $\boldsymbol{z}$ was arbitrary, $f_{\boldsymbol{w}}(\boldsymbol{z}) = (\underbrace{\frac{1}{|\mathcal{Y}|}, ..., \frac{1}{|\mathcal{Y}|}}_{|\mathcal{Y}| \text{ times}}) \; \forall \boldsymbol{z} \in \mathcal{D}$. Since $\mathcal{D}$ was arbitrary, by definition of a uniform learner over $\mathcal{D}$, $f_{\mathcal{K}(\mathcal{D})} \in \mathcal{H}_{\mathcal{W}} \; \forall \mathcal{D} \subset \mathcal{Z}^n$ as desired.

$\qquad \square$

## H.3 PROOF OF PROPOSITION 4.3

We use the following definition of global Pareto optimality:

**Definition H.10.** *(Chapter 1 of Pardalos et al. (2017)) Suppose we have a multiobjective optimization problem $\min \boldsymbol{f}(\boldsymbol{x})$ s.t. $\boldsymbol{x} \in A$, where $\boldsymbol{f}(\boldsymbol{x}) = (f_1(\boldsymbol{x}), f_2(\boldsymbol{x}), ..., f_m(\boldsymbol{x}))$. $\boldsymbol{x}^* \in A$ with $\boldsymbol{f}(\boldsymbol{x}^*)$ is called globally Pareto optimal if and only if there exists no $\boldsymbol{x} \in A$ such that $f_i(\boldsymbol{x}) \leq f_i(\boldsymbol{x}^*)$ for all $i = 1, 2, ..., m$ and $f_j(\boldsymbol{x}) < f_j(\boldsymbol{x}^*)$ for at least one $j \in \{1, ..., m\}$.*

We can then prove the statement:

*Proof.* Let $\theta \in (0, 1)$. Fix $\mathcal{D} \subset \mathcal{Z}^n$, $\mathcal{D}_f \subset \mathcal{D}$, and $\mathcal{D}_r = \mathcal{D} \setminus \mathcal{D}_f$.

Suppose, for the sake of contradiction, that $\tilde{\boldsymbol{w}}^* = \mathcal{M}_\theta(\mathcal{D}) = \text{argmin}_{\boldsymbol{w}} \theta \mathcal{L}_{\mathcal{K}}(\boldsymbol{w}, \mathcal{D}_f) + (1 - \theta)\mathcal{L}_{\mathcal{A}}(\boldsymbol{w}, \mathcal{D}_r)$, a global minimizer, is not globally Pareto optimal with respect to $\mathcal{L}_{\mathcal{K}}(\boldsymbol{w}, \mathcal{D}_f)$ and $\mathcal{L}_{\mathcal{A}}(\boldsymbol{w}, \mathcal{D}_r)$. Then, exists $\boldsymbol{w}'$ s.t. $\mathcal{L}_{\mathcal{K}}(\boldsymbol{w}', \mathcal{D}_f) \leq \mathcal{L}_{\mathcal{K}}(\tilde{\boldsymbol{w}}^*, \mathcal{D}_f)$ and $\mathcal{L}_{\mathcal{A}}(\boldsymbol{w}', \mathcal{D}_r) \leq \mathcal{L}_{\mathcal{A}}(\tilde{\boldsymbol{w}}^*, \mathcal{D}_r)$, with at least one of these inequalities being strict.

Then, since $\theta \in (0, 1)$ and $(1 - \theta) \in (0, 1)$, we have that $\theta \mathcal{L}_{\mathcal{K}}(\boldsymbol{w}', \mathcal{D}_f) + (1 - \theta)\mathcal{L}_{\mathcal{A}}(\boldsymbol{w}', \mathcal{D}_r) < \theta \mathcal{L}_{\mathcal{K}}(\tilde{\boldsymbol{w}}^*, \mathcal{D}_f) + (1 - \theta)\mathcal{L}_{\mathcal{A}}(\tilde{\boldsymbol{w}}^*, \mathcal{D}_r)$, contradicting optimality of $\tilde{\boldsymbol{w}}^*$. As such, $\mathcal{M}_\theta(\mathcal{D})$ is globally Pareto optimal respect to $\mathcal{L}_{\mathcal{K}}(\boldsymbol{w}, \mathcal{D}_f)$ and $\mathcal{L}_{\mathcal{A}}(\boldsymbol{w}, \mathcal{D}_r)$ as desired.

### H.4 PROOF OF THEOREM G.1

*Proof.* The proof follows similarly to Lemma 10 in Sekhari et al. (2021); for completeness, we adapt their proof to our setting.

Let $\boldsymbol{w}^* := A(D), \boldsymbol{w}^- := \mathcal{G}(\mathcal{D}, \mathcal{D}_f, \boldsymbol{w}^*), \tilde{\boldsymbol{w}} := \mathcal{F}(\mathcal{D}, \mathcal{D}_f, \boldsymbol{w}^*)$. Departing from the notation of the theorem for clarity, let $\hat{\boldsymbol{w}}^* := \mathcal{M}_\theta(\mathcal{D}), \hat{\boldsymbol{w}^-} := \mathcal{G}(\mathcal{D}, \emptyset, \hat{\boldsymbol{w}}^*), \hat{\tilde{\boldsymbol{w}}} := \mathcal{F}(\mathcal{D}, \emptyset, \hat{\boldsymbol{w}}^*)$.

Note that $\hat{\tilde{\boldsymbol{w}}} = \hat{\boldsymbol{w}}^*$. We then have that $||\tilde{\boldsymbol{w}} - \hat{\tilde{\boldsymbol{w}}}||_2 = ||\tilde{\boldsymbol{w}} - \hat{\boldsymbol{w}}^*||_2 \leq \Delta$, by definition of $\Delta$.

By definition of $\mathcal{G}$, we have that $\boldsymbol{w}^- = \tilde{\boldsymbol{w}} + Y$ and $\hat{\boldsymbol{w}^-} = \hat{\tilde{\boldsymbol{w}}} + Y$, where $Y \sim \mathcal{N}(0, \sigma^2 \boldsymbol{I})$ s.t. $\sigma \geq \frac{\Delta}{\varepsilon}\sqrt{2\ln(1.25/\delta)}$.

As such, $\boldsymbol{w}^- = \mathcal{N}(\tilde{\boldsymbol{w}}, \sigma^2 \boldsymbol{I})$ and $\hat{\boldsymbol{w}^-} \sim \mathcal{N}(\hat{\tilde{\boldsymbol{w}}}, \sigma^2 \boldsymbol{I})$.

Thus, by Lemma H.4, $\boldsymbol{w}^-, \hat{\boldsymbol{w}^-}$ are $(\varepsilon, \delta)$-indistinguishable. In particular, since $\hat{\boldsymbol{w}^-} = \hat{\boldsymbol{w}}^*$ by construction, $\mathcal{G}(\mathcal{D}, \mathcal{D}_f, \mathcal{A}(\mathcal{D}))$ and $\mathcal{M}_\theta(\mathcal{D})$ are $(\varepsilon, \delta)$-indistinguishable, as desired.

□

### H.5 PROOF OF PROPOSITION G.2

*Proof.* By the same token as Lemma 3.3 in (Zhang et al., 2024), we have that:

$$||\tilde{\boldsymbol{w}} - \tilde{\boldsymbol{w}}^*||_2 \leq ||\boldsymbol{H}^{-1}_{\boldsymbol{w}^*,\mathcal{K},\mathcal{A}}||_2 \int_0^1 ||\boldsymbol{H}_{\boldsymbol{w}^*,\mathcal{K},\mathcal{A}} - \boldsymbol{H}_{\boldsymbol{w}^*+t(\tilde{\boldsymbol{w}}-\boldsymbol{w}^*),\mathcal{K},\mathcal{A}}||_2 ||\boldsymbol{w}^* - \tilde{\boldsymbol{w}}^*||_2 dt \quad (\text{H.53})$$

.

Let $\boldsymbol{w}' = \boldsymbol{w}^* + t(\tilde{\boldsymbol{w}} - \boldsymbol{w}^*)$. We have that $||\boldsymbol{w}^* - \boldsymbol{w}'||_2 = ||\boldsymbol{w}^* - \boldsymbol{w}^* + t(\tilde{\boldsymbol{w}}^* - \boldsymbol{w}^*)||_2 = t||\boldsymbol{w}^* - \tilde{\boldsymbol{w}}^*||_2$.

Furthermore, by linearity of $\boldsymbol{H}_{\boldsymbol{w}}$ and the triangle inequality, we have that:

$$||\boldsymbol{H}_{\boldsymbol{w}^*,\mathcal{K},\mathcal{A}} - \boldsymbol{H}_{\boldsymbol{w}',\mathcal{K},\mathcal{A}}||_2 = ||\theta \boldsymbol{H}_{\boldsymbol{w}^*,\mathcal{K}} + (1-\theta)\boldsymbol{H}_{\boldsymbol{w}^*,\mathcal{A}} - \theta \boldsymbol{H}_{\boldsymbol{w}',\mathcal{K}} - (1-\theta)\boldsymbol{H}_{\boldsymbol{w}',\mathcal{A}}||_2 \quad (\text{H.54})$$

$$\leq \theta ||\boldsymbol{H}_{\boldsymbol{w}^*,\mathcal{K}} - \theta \boldsymbol{H}_{\boldsymbol{w}',\mathcal{K}}||_2 + (1-\theta)||\boldsymbol{H}_{\boldsymbol{w}^*,\mathcal{A}} - \boldsymbol{H}_{\boldsymbol{w}',\mathcal{A}}||_2 \quad (\text{H.55})$$

$$= \theta F_{\mathcal{K}} ||\boldsymbol{w}^* - \boldsymbol{w}'||_2 + (1-\theta)F_{\mathcal{A}} ||\boldsymbol{w}^* - \boldsymbol{w}'||_2, \text{ by Lemma H.1} \quad (\text{H.56})$$

$$= \theta t F_{\mathcal{K}} ||\boldsymbol{w}^* - \tilde{\boldsymbol{w}}^*||_2 + (1-\theta)t F_{\mathcal{A}} ||\boldsymbol{w}^* - \tilde{\boldsymbol{w}}^*||_2, \quad (\text{H.57})$$

This yields that:

$$||\tilde{\boldsymbol{w}} - \tilde{\boldsymbol{w}}^*||_2 \leq ||\boldsymbol{H}^{-1}_{\boldsymbol{w}^*,\mathcal{K},\mathcal{A}}||_2 \int_0^1 ||\boldsymbol{H}_{\boldsymbol{w}^*,\mathcal{K},\mathcal{A}} - \boldsymbol{H}_{\boldsymbol{w}^*+t(\tilde{\boldsymbol{w}}^*-\boldsymbol{w}^*),\mathcal{K},\mathcal{A}}||_2 ||\boldsymbol{w}^* - \tilde{\boldsymbol{w}}^*||_2 dt \quad (\text{H.58})$$

$$\leq ||\boldsymbol{H}^{-1}_{\boldsymbol{w}^*,\mathcal{K},\mathcal{A}}||_2 \int_0^1 (\theta t F_{\mathcal{K}} + (1-\theta)t F_{\mathcal{A}}) ||\boldsymbol{w}^* - \tilde{\boldsymbol{w}}^*||_2^2 \quad (\text{H.59})$$

$$= \frac{\theta F_{\mathcal{K}} + (1-\theta)F_{\mathcal{A}}}{2} ||\boldsymbol{H}^{-1}_{\boldsymbol{w}^*,\mathcal{K},\mathcal{A}}||_2 ||\boldsymbol{w}^* - \tilde{\boldsymbol{w}}^*||_2^2 \quad (\text{H.60})$$

as desired.

□

### H.6 PROOF OF PROPOSITION G.3

*Proof.* See the proof of theorem 3.4 in (Zhang et al., 2024), noting that in our setting $M = F = \theta F_{\mathcal{K}} + (1-\theta)F_{\mathcal{A}}$ by Eq. (H.57). $\qquad\square$

### H.7 PROOF OF THEOREM G.4

*Proof.* First, we have that:

$$\mathbb{E}[\tilde{H}_{t,\lambda}^{-1}] = \mathbb{E}[I + \tilde{H}_{t-1,\lambda}^{-1} - \frac{1}{J}H_{t,\lambda}\tilde{H}_{t-1,\lambda}^{-1}], \text{ by definition} \tag{H.61}$$

$$= I + \mathbb{E}[\tilde{H}_{t-1,\lambda}^{-1}] - \frac{1}{J}\mathbb{E}[H_{t,\lambda}\tilde{H}_{t-1,\lambda}], \text{ linearity of expectation} \tag{H.62}$$

$$= I + \mathbb{E}[\tilde{H}_{t-1,\lambda}^{-1}] - \frac{1}{J}\mathbb{E}[H_{t,\lambda}]\mathbb{E}[\tilde{H}_{t-1,\lambda}], \text{i.i.d. samples} \tag{H.63}$$

$$= I + \mathbb{E}[\tilde{H}_{t-1,\lambda}^{-1}] - \frac{H_{w^*,\mathcal{K},\mathcal{A}} + \lambda I}{J}\mathbb{E}[\tilde{H}_{t-1,\lambda}], \text{ by Lemma H.5} \tag{H.64}$$

Denote $H_* := H_{w^*,\mathcal{K},\mathcal{A}}$ and $E_t := \mathbb{E}[\tilde{H}_{t,\lambda}^{-1}]$. We thus have that:

$$E_t = I + E_{t-1} - \frac{H_*}{J}E_{t-1} \tag{H.65}$$

$$= I + E_{t-1}(I - \frac{H_*}{J}) \tag{H.66}$$

$$= I + (I - M)E_{t-1}, \text{ letting } M := \frac{H_*}{J} \tag{H.67}$$

We then know that, by assumption, $\lambda > ||H_*||_2$, where $H_*$ is a symmetric Hessian by Lemma H.3; as such, $H_* + \lambda I$ is positive definite and has all positive eigenvalues. We also know that $||H_*||_2 < J \implies ||M||_2 < 1$, so we have that $0 < \lambda_i(M) < 1$ for all eigenvalues $\lambda_i$. Furthermore, $I - M$ has eigenvalues $1 - \lambda_i(M)$, so we have that $0 < \lambda_i(I - M) < 1$, so $||I - M||_2 < 1$, since $I - M$ is symmetric. Since $I - M$ has spectral radius less than 1, the Neumann series $\sum_{k=0}^{\infty}(I - M)^k$ converges. (Mayer, 1985). Thus, the Neumann series is Cauchy.

Fix $\varepsilon > 0$. Let $s_n = \sum_{k=0}^{n}(I - M)^k$. We know that $\exists N \in \mathbb{N}$ s.t. $m > n \geq N \implies ||s_m - s_n||_2 = ||\sum_{k=n+1}^{m}(I - M)^k||_2 < \varepsilon$. For $m > n \geq N$, we have that $||E_m - E_n||_2 = ||\sum_{k=n+1}^{m}(I - M)^k||_2 < \varepsilon$. As such, $\{E_n\}$ is Cauchy; since it is real, it converges. As such, $E_\infty = \lim_{t \to \infty} E_n$ exists.

Taking limits on both sides, we then have:

$$\mathbb{E}[\tilde{H}_{\infty,\lambda}^{-1}] = I + \mathbb{E}[\tilde{H}_{\infty,\lambda}^{-1}] + \frac{H_{w^*,\mathcal{K},\mathcal{A}} + \lambda I}{J}\mathbb{E}[\tilde{H}_{\infty,\lambda}] \tag{H.68}$$

$$\iff \mathbb{E}[\frac{\tilde{H}_{\infty,\lambda}^{-1}}{J}] = (H_{w^*,\mathcal{K},\mathcal{A}} + \lambda I)^{-1} \tag{H.69}$$

rearranging using linearity of expectation and noting that $\lambda$ was chosen such that $H_{w^*,\mathcal{K},\mathcal{A}} + \lambda I$ is invertible, as desired. $\qquad\square$

### H.8 PROOF OF THEOREM G.5

This follows similarly to theorem 3.6 and proposition 4.1 in Zhang et al. (2024), noting that we apply Lemma H.7 instead of applying lemma 3.6 from Agarwal et al. (2016). Furthermore, note that $L = \theta P_{\mathcal{K}} + (1-\theta)P_{\mathcal{A}}$ in our setting. For completeness, we provide the full proof below.

*Proof.*

$$\tilde{\boldsymbol{w}} - \tilde{\boldsymbol{w}}^* = \boldsymbol{w}^* - \frac{\tilde{\boldsymbol{H}}_{n,\lambda}^{-1}}{H}\nabla_{\boldsymbol{w}^*,\mathcal{K},\mathcal{A}} - \tilde{\boldsymbol{w}}^* \tag{H.70}$$

$$= \boldsymbol{w}^* - \tilde{\boldsymbol{w}}^* - \frac{\tilde{\boldsymbol{H}}_{n,\lambda}^{-1}}{H}(\nabla_{\boldsymbol{w}^*,\mathcal{K},\mathcal{A}} - \nabla_{\tilde{\boldsymbol{w}}^*,\mathcal{K},\mathcal{A}}) - \frac{\tilde{\boldsymbol{H}}_{n,\lambda}^{-1}}{H}\nabla_{\tilde{\boldsymbol{w}}^*,\mathcal{K},\mathcal{A}} \tag{H.71}$$

By the triangle inequality, this yields:

$$||\tilde{\boldsymbol{w}} - \tilde{\boldsymbol{w}}^*||_2 \leq ||\boldsymbol{w}^* - \tilde{\boldsymbol{w}}^* - \frac{\tilde{\boldsymbol{H}}_{n,\lambda}^{-1}}{H}(\nabla_{\boldsymbol{w}^*,\mathcal{K},\mathcal{A}} - \nabla_{\tilde{\boldsymbol{w}}^*,\mathcal{K},\mathcal{A}})||_2 + ||\frac{\tilde{\boldsymbol{H}}_{n,\lambda}^{-1}}{H}\nabla_{\tilde{\boldsymbol{w}}^*,\mathcal{K},\mathcal{A}}||_2 \tag{H.72}$$

The first term in Eq. (H.72) can be bounded by the triangle inequality as:

$$||\boldsymbol{w}^* - \tilde{\boldsymbol{w}}^* - \frac{\tilde{\boldsymbol{H}}_{n,\lambda}^{-1}}{H}(\nabla_{\boldsymbol{w}^*,\mathcal{K},\mathcal{A}} - \nabla_{\tilde{\boldsymbol{w}}^*,\mathcal{K},\mathcal{A}})||_2 \tag{H.73}$$

$$= ||\boldsymbol{w}^* - \tilde{\boldsymbol{w}}^* - ((\boldsymbol{H}_{\boldsymbol{w}^*,\mathcal{K},\mathcal{A}} + \lambda\boldsymbol{I})^{-1} + \frac{\tilde{\boldsymbol{H}}_{n,\lambda}^{-1}}{H} - (\boldsymbol{H}_{\boldsymbol{w}^*,\mathcal{K},\mathcal{A}} + \lambda\boldsymbol{I})^{-1})(\nabla_{\boldsymbol{w}^*,\mathcal{K},\mathcal{A}} - \nabla_{\tilde{\boldsymbol{w}}^*,\mathcal{K},\mathcal{A}})||_2 \tag{H.74}$$

$$\leq ||\boldsymbol{w}^* - \tilde{\boldsymbol{w}}^* - (\boldsymbol{H}_{\boldsymbol{w}^*,\mathcal{K},\mathcal{A}} + \lambda\boldsymbol{I})^{-1}(\nabla_{\boldsymbol{w}^*,\mathcal{K},\mathcal{A}} - \nabla_{\tilde{\boldsymbol{w}}^*,\mathcal{K},\mathcal{A}})||_2 \tag{H.75}$$

$$+ ||((\boldsymbol{H}_{\boldsymbol{w}^*,\mathcal{K},\mathcal{A}} + \lambda\boldsymbol{I})^{-1} - \frac{\tilde{\boldsymbol{H}}_{n,\lambda}^{-1}}{H})(\nabla_{\boldsymbol{w}^*,\mathcal{K},\mathcal{A}} - \nabla_{\tilde{\boldsymbol{w}}^*,\mathcal{K},\mathcal{A}})||_2 \tag{H.76}$$

In the setting of Prop. G.3, we have that $\tilde{\boldsymbol{w}} - \tilde{\boldsymbol{w}}^* = \boldsymbol{w}^* - \tilde{\boldsymbol{w}}^* - (\boldsymbol{H}_{\boldsymbol{w}^*,\mathcal{K},\mathcal{A}} + \lambda\boldsymbol{I})^{-1}(\nabla_{\boldsymbol{w}^*,\mathcal{K},\mathcal{A}} - \nabla_{\tilde{\boldsymbol{w}}^*,\mathcal{K},\mathcal{A}})$. Hence, by Prop. G.3, we have that:

$$||\boldsymbol{w}^* - \tilde{\boldsymbol{w}}^* - (\boldsymbol{H}_{\boldsymbol{w}^*,\mathcal{K},\mathcal{A}} + \lambda\boldsymbol{I})^{-1}(\nabla_{\boldsymbol{w}^*,\mathcal{K},\mathcal{A}} - \nabla_{\tilde{\boldsymbol{w}}^*,\mathcal{K},\mathcal{A}})||_2 \tag{H.77}$$

$$\leq \frac{2C((\theta F_{\mathcal{K}} + (1-\theta)F_{\mathcal{A}})C + \lambda)}{\lambda + \lambda_{\min}} \tag{H.78}$$

Furthermore, we have:

$$||((\boldsymbol{H}_{\boldsymbol{w}^*,\mathcal{K},\mathcal{A}} + \lambda\boldsymbol{I})^{-1} - \frac{\tilde{\boldsymbol{H}}_{n,\lambda}^{-1}}{H})(\nabla_{\boldsymbol{w}^*,\mathcal{K},\mathcal{A}} - \nabla_{\tilde{\boldsymbol{w}}^*,\mathcal{K},\mathcal{A}})||_2 \tag{H.79}$$

$$\leq ||((\boldsymbol{H}_{\boldsymbol{w}^*,\mathcal{K},\mathcal{A}} + \lambda\boldsymbol{I})^{-1} - \frac{\tilde{\boldsymbol{H}}_{n,\lambda}^{-1}}{H})||_2 ||(\nabla_{\boldsymbol{w}^*,\mathcal{K},\mathcal{A}} - \nabla_{\tilde{\boldsymbol{w}}^*,\mathcal{K},\mathcal{A}})||_2, \text{ property of op norm} \tag{H.80}$$

$$\leq (16\frac{B}{\zeta_{\min}}\sqrt{\frac{\ln\frac{d}{\rho}}{b}} + \frac{1}{16})||(\nabla_{\boldsymbol{w}^*,\mathcal{K},\mathcal{A}} - \nabla_{\tilde{\boldsymbol{w}}^*,\mathcal{K},\mathcal{A}})||_2, \text{ Lemma H.7} \tag{H.81}$$

$$\leq (16\frac{B}{\zeta_{\min}}\sqrt{\frac{\ln\frac{d}{\rho}}{b}} + \frac{1}{16})2C(\theta P_{\mathcal{K}} + (1-\theta)P_{\mathcal{A}}, \text{ Lemma H.1} \tag{H.82}$$

with probability at least 1 - $\rho$. Incorporating this into equation Eq. (H.76), we have that:

$$||\boldsymbol{w}^* - \tilde{\boldsymbol{w}}^* - \frac{\tilde{\boldsymbol{H}}_{n,\lambda}^{-1}}{H}(\nabla_{\boldsymbol{w}^*,\mathcal{K},\mathcal{A}} - \nabla_{\tilde{\boldsymbol{w}}^*,\mathcal{K},\mathcal{A}})||_2 \leq \frac{2C((\theta F_{\mathcal{K}} + (1-\theta)F_{\mathcal{A}})C + \lambda)}{\lambda + \lambda_{\min}} \tag{H.83}$$

$$+ (32\frac{B}{\zeta_{\min}}\sqrt{\frac{\ln\frac{d}{\rho}}{b}} + \frac{1}{8})C(\theta P_{\mathcal{K}} + (1-\theta)P_{\mathcal{A}}) \tag{H.84}$$

It then suffices to bound the second term in Eq. (H.72). We have that:

$$\left\|\tfrac{\tilde{\boldsymbol{H}}_{n,\lambda}^{-1}}{H}\,\nabla_{\tilde{\boldsymbol{w}}^*,\mathcal{K},\mathcal{A}}\right\|_2 = \left\|\left[(\boldsymbol{H}_{w^*,\mathcal{K},\mathcal{A}}+\lambda\boldsymbol{I})^{-1} - (\boldsymbol{H}_{w^*,\mathcal{K},\mathcal{A}}+\lambda\boldsymbol{I})^{-1}\right.\right.$$
$$\left.\left.+\;\tfrac{\tilde{\boldsymbol{H}}_{n,\lambda}^{-1}}{H}\right]\nabla_{\tilde{\boldsymbol{w}}^*,\mathcal{K},\mathcal{A}}\right\|_2 \tag{H.85}$$

$$= \left\|(\boldsymbol{H}_{w^*,\mathcal{K},\mathcal{A}}+\lambda\boldsymbol{I})^{-1}\,\nabla_{\tilde{\boldsymbol{w}}^*,\mathcal{K},\mathcal{A}}\right.$$
$$\left.+\;\big(\tfrac{\tilde{\boldsymbol{H}}_{n,\lambda}^{-1}}{H}-(\boldsymbol{H}_{w^*,\mathcal{K},\mathcal{A}}+\lambda\boldsymbol{I})^{-1}\big)\,\nabla_{\tilde{\boldsymbol{w}}^*,\mathcal{K},\mathcal{A}}\right\|_2 \tag{H.86}$$

$$\le \|(\boldsymbol{H}_{w^*,\mathcal{K},\mathcal{A}}+\lambda\boldsymbol{I})^{-1}\|_2\,\|\nabla_{\tilde{\boldsymbol{w}}^*,\mathcal{K},\mathcal{A}}\|_2$$
$$+\;\big\|\tfrac{\tilde{\boldsymbol{H}}_{n,\lambda}^{-1}}{H}-(\boldsymbol{H}_{w^*,\mathcal{K},\mathcal{A}}+\lambda\boldsymbol{I})^{-1}\big\|_2\,\|\nabla_{\tilde{\boldsymbol{w}}^*,\mathcal{K},\mathcal{A}}\|_2 \tag{H.87}$$

$$\le \frac{G}{\lambda+\lambda_{\min}} + \Big(16\frac{B}{\zeta_{\min}}\sqrt{\tfrac{\ln(d/\rho)}{b}}+\tfrac{1}{16}\Big)G. \tag{H.88}$$

by definition of $\lambda$, Lemma H.7, and that $\|\nabla_{\tilde{\boldsymbol{w}}^*,\mathcal{K},\mathcal{A}}\|_2 \le G$.

Incorporating the above into Eq. (H.72), this yields that:

$$\|\tilde{\boldsymbol{w}}-\tilde{\boldsymbol{w}}^*\|_2 \le \frac{2C\big((\theta F_{\mathcal{K}}+(1-\theta)F_{\mathcal{A}})\,C+\lambda\big)}{\lambda+\lambda_{\min}} \tag{H.89}$$
$$+\Big(32\frac{B}{\zeta_{\min}}\sqrt{\tfrac{\ln(d/\rho)}{b}}+\tfrac{1}{8}\Big)C\Big(\theta P_{\mathcal{K}}+(1-\theta)P_{\mathcal{A}}+\frac{G}{\lambda+\lambda_{\min}}$$
$$+\Big(16\frac{B}{\zeta_{\min}}\sqrt{\tfrac{\ln(d/\rho)}{b}}+\tfrac{1}{16}\Big)G\Big) \tag{H.90}$$

$$= \frac{2C\big((\theta F_{\mathcal{K}}+(1-\theta)F_{\mathcal{A}})\,C+\mu\big)\;+\;G}{\mu+\mu_{\min}} \tag{H.91}$$
$$+\Big(16\frac{B}{\zeta_{\min}}\sqrt{\tfrac{\ln(d/\rho)}{b}}+\tfrac{1}{16}\Big)\big(2C(\theta P_{\mathcal{K}}+(1-\theta)P_{\mathcal{A}}+G)\big).$$

as desired.

$\square$

## H.9    PROOF OF PROPOSITION I.1

*Proof.* By the same token as proposition 4.2 in Zhang et al. (2024), follow the proof of theorem G.5.

$\square$

## H.10    PROOF OF PROPOSITION 5.1

*Proof.* Fix any sampled $\mathcal{D}$. Since $\mathcal{M}_\theta(\mathcal{D})$ is taken to be the global risk minimizer, we have that:

$$\theta\mathcal{L}_{\mathcal{K}}(\mathcal{M}_\theta(\mathcal{D}),\mathcal{D}_f)+(1-\theta)\mathcal{L}_{\mathcal{A}}(\mathcal{M}_\theta(\mathcal{D}),\mathcal{D}_r) \tag{H.92}$$
$$\le \theta\mathcal{L}_{\mathcal{K}}(\boldsymbol{w},\mathcal{D}_f)+(1-\theta)\mathcal{L}_{\mathcal{A}}(\boldsymbol{w},\mathcal{D}_r)\;\forall\boldsymbol{w}\in\mathcal{W} \tag{H.93}$$

subtracting $\frac{\lambda}{2}\|\boldsymbol{w}\|_2$ from both sides.

Let $\boldsymbol{w}_U$ be the parameter that results in a parameterized model $f_{\boldsymbol{w}_U}$ which outputs a uniform distribution; by Prop. 4.2, such a parameter exists. We then have that:

$$\mathcal{L}_{\mathcal{K}}(\boldsymbol{w}_U,\mathcal{D}_f) = \sum_{i=1}^{|\mathcal{D}_f|} D_{KL}(U[0,|\mathcal{Y}|])\|U[0,|\mathcal{Y}|] = 0 \tag{H.94}$$

and

$$\mathcal{L}_{\mathcal{A}}(\boldsymbol{w}_U, \mathcal{D}_r) = \sum_{i=1}^{|\mathcal{D}_r|} \mathbb{H}_{CE}(\boldsymbol{y}^{(i)}, U[0, |\mathcal{Y}|]) = -\sum_{i=1}^{|\mathcal{D}_r|}\sum_{j=1}^{|\mathcal{Y}|} \boldsymbol{y}_j^{(i)} \ln \frac{1}{|\mathcal{Y}|} = |\mathcal{D}_r| \ln |\mathcal{Y}| \quad \text{(H.95)}$$

where $\boldsymbol{y}$ is a one hot vector of length $\mathcal{Y}$ such that for $\boldsymbol{y}_j^{(i)}$, $j = 1, ..., |\mathcal{Y}|$,

$$\boldsymbol{y}_j^{(i)} = \begin{cases} 1 & \text{instance } i \text{ is labeled class } j \\ 0 & \text{instance } i \text{ is not labeled class } j \end{cases} \quad \text{(H.96)}$$

Incorporating the above into Eq. (H.93) yields:

$$\theta\mathcal{L}_{\mathcal{K}}(\mathcal{M}_\theta(\mathcal{D}), \mathcal{D}_f) + (1-\theta)\mathcal{L}_{\mathcal{A}}(\mathcal{M}_\theta(\mathcal{D}), \mathcal{D}_r) \leq \theta\mathcal{L}_{\mathcal{K}}(\boldsymbol{w}_U, \mathcal{D}_f) + (1-\theta)\mathcal{L}_{\mathcal{A}}(\boldsymbol{w}_U, \mathcal{D}_r) \quad \text{(H.97)}$$
$$\leq \theta(0) + (1-\theta)|\mathcal{D}_r| \ln |\mathcal{Y}| \quad \text{(H.98)}$$
$$= |\mathcal{D}_r|(1-\theta) \ln |\mathcal{Y}| \quad \text{(H.99)}$$

This then yields that:

$$\mathcal{L}_{\mathcal{K}}(\mathcal{M}_\theta(\mathcal{D}), \mathcal{D}_f) \leq \frac{1-\theta}{\theta}(|\mathcal{D}_r| \ln |\mathcal{Y}| - \mathcal{L}_{\mathcal{A}}(\mathcal{M}_\theta(\mathcal{D}), \mathcal{D}_r) \quad \text{(H.100)}$$
$$\leq \frac{1-\theta}{\theta}|\mathcal{D}_r| \ln |\mathcal{Y}| \quad \text{(H.101)}$$

since the cross entropy is nonnegative, yielding that $-\mathcal{L}_{\mathcal{A}}(\mathcal{M}_\theta(\mathcal{D}), \mathcal{D}_r) \leq 0$.

Then, we have:

$$||f_{\mathcal{M}_\theta(\mathcal{D})}(\mathcal{D}_f) - U[0, |\mathcal{Y}|]||_\infty \leq ||f_{\mathcal{M}_\theta(\mathcal{D})}(\mathcal{D}_f) - U[0, |\mathcal{Y}|]||_1 \quad \text{(H.102)}$$
$$\leq 2TV(f_{\mathcal{M}_\theta(\mathcal{D})}(\mathcal{D}_f) - U[0, |\mathcal{Y}|]) \quad \text{(H.103)}$$
$$\leq 2\sqrt{\frac{1}{2}D_{KL}(f_{\mathcal{M}_\theta(\mathcal{D})}, ||U[0, |\mathcal{Y}|])}, \text{ Pinsker's inequality (Pinsker, 1964)} \quad \text{(H.104)}$$
$$= \sqrt{2D_{KL}(f_{\mathcal{M}_\theta(\mathcal{D})}, ||U[0, |\mathcal{Y}|])} \quad \text{(H.105)}$$
$$= \sqrt{2\mathcal{L}_{\mathcal{K}}(\mathcal{M}_\theta(\mathcal{D}), \mathcal{D}_f)} \quad \text{(H.106)}$$
$$\leq \sqrt{2|\mathcal{D}_r|(\frac{1-\theta}{\theta}) \ln |\mathcal{Y}|} \quad \text{(H.107)}$$

by the above bound on $\mathcal{L}_{\mathcal{K}}$, as desired. $\qquad \square$

## H.11    PROOF OF COROLLARY 5.2

*Proof.* To have Eq. (7), by Prop. 5.1, it suffices to solve for $\theta$ in the bound obtained. This results in:

$$\sqrt{2(\frac{1-\theta}{\theta})|\mathcal{D}_r|\ln|\mathcal{Y}|} \leq \varepsilon \iff \frac{1-\theta}{\theta}|\mathcal{D}_r|\ln|\mathcal{Y} \leq \frac{\varepsilon^2}{2} \tag{H.108}$$

$$\iff \frac{|\mathcal{D}_r|\ln|\mathcal{Y}|}{\theta} - \frac{|\mathcal{D}_r|\ln|\mathcal{Y}|\theta}{\theta} \leq \frac{\varepsilon^2}{2} \tag{H.109}$$

$$\iff \frac{|\mathcal{D}_r|\ln|\mathcal{Y}|}{\theta} \leq \frac{\varepsilon^2}{2} + |\mathcal{D}_r|\ln|\mathcal{Y}| = \frac{\varepsilon^2 + 2|\mathcal{D}_r|\ln|\mathcal{Y}|}{2} \tag{H.110}$$

$$\iff \frac{\theta}{|\mathcal{D}_r|\ln|\mathcal{Y}|} \geq \frac{2}{\varepsilon^2 + 2|\mathcal{D}_r|\ln|\mathcal{Y}|} \tag{H.111}$$

$$\iff \theta \geq \frac{2|\mathcal{D}_r|\ln|\mathcal{Y}}{\varepsilon^2 + 2|\mathcal{D}_r|\ln|\mathcal{Y}|} \tag{H.112}$$

as desired.

$\square$

## H.12 PROOF OF THEOREM 5.5

First, before we prove theorem 5.5, we note that we can use Lemma H.2 and that $||\boldsymbol{w}|| \leq 2$ to obtain a simple bound. Let:

$$|\alpha^* - \alpha(\theta)| = |\mathcal{L}_\mathcal{A}(\mathcal{A}(\mathcal{D}_r), \mathcal{D}_r) - \mathcal{L}_\mathcal{A}(\mathcal{M}_\theta(\mathcal{D}), \mathcal{D}_r)| \tag{H.113}$$

$$\leq \frac{P_\mathcal{A}}{2}||\mathcal{M}_\theta(\mathcal{D}) - \mathcal{A}(\mathcal{D}_r)||_2^2 + ||\nabla_{\mathcal{A}(\mathcal{D}_r),\mathcal{A}}||_2||\mathcal{M}_\theta(\mathcal{D}) - \mathcal{A}(\mathcal{D}_r)||_2 \tag{H.114}$$

$$\leq \frac{C^2 P_\mathcal{A}}{2} + \lambda C^2 \tag{H.115}$$

after applying the triangle inequality and rearranging the first order condition on $\mathcal{A}(\mathcal{D}_r)$.

However, this bound is vacuous and not tight; it does not incorporate any information about $\theta$ or most of the constants that appear in Asm. 5.3 and Asm. 5.4. Given this, we seek to construct a tighter, non-vacuous bound. We first restate the proof without any asymptotic characterizations:

**Theorem H.11.** *Suppose Assumptions 5.3 and 5.4 hold, and let $P_\mathcal{K}, P_\mathcal{K}, F_\mathcal{K}, F_\mathcal{A}$ be as defined in Assumptions 5.3 and 5.4. Let $\alpha^* := \mathcal{L}_\mathcal{A}(\mathcal{A}(\mathcal{D}_r), \mathcal{D}_r)$ be the locally optimal (empirical) retain loss, achieved by $\mathcal{M}_\theta(\mathcal{D})$ when $\theta = 0$. Let $\alpha(\theta) := \mathcal{L}_\mathcal{A}(\mathcal{M}_\theta(\mathcal{D}), \mathcal{D}_r)$ be the locally optimal retain loss obtained by $\mathcal{M}_\theta(\mathcal{D})$ when $\theta \in (0, 1)$. Suppose all weights used throughout are bounded by $||\boldsymbol{w}||_2 \leq C$. Additionally, denote by $F := \theta M_\mathcal{K} + (1 - \theta)F_\mathcal{A}$ and $P := \theta P_\mathcal{K} + (1 - \theta)P_\mathcal{A}$. Consider regularization coefficient $\lambda \geq L + 2\theta CF + \sqrt{2\theta CF(P + 2\theta CF + 8P_\mathcal{K})}$. Then, we have the following bound:*

$$|\alpha^* - \alpha(\theta)| \leq \frac{P_\mathcal{K}}{2}(\frac{\lambda - P - \sqrt{(\lambda - P)^2 - 4\theta CF(2P_\mathcal{K} + \lambda)}}{2F})^2 + \tag{H.116}$$

$$\lambda C(\frac{\lambda - P - \sqrt{(\lambda - P)^2 - 4\theta CF(2P_\mathcal{K} + \lambda)}}{2F}). \tag{H.117}$$

*Proof.* First, when $\theta = 0$, we have that:

$$\boldsymbol{w}_{\alpha^*} := \arg\min_{\boldsymbol{w}\in\mathcal{W},||\boldsymbol{w}||_2\leq C} \mathcal{L}_\mathcal{A}(\boldsymbol{w}, \mathcal{D}_r) + \frac{\lambda}{2}||\boldsymbol{w}||_2^2 \tag{H.118}$$

which yields the first order condition:

$$\nabla_{\boldsymbol{w}_{\alpha^*},\mathcal{A}} + \lambda\boldsymbol{w}_{\alpha^*} = 0 \tag{H.119}$$

which, upon multiplying $1 - \theta$ on both sides, yields:

$$(1 - \theta)\nabla_{\boldsymbol{w}_{\alpha^*},\mathcal{A}} + (1 - \theta)\lambda\boldsymbol{w}_{\alpha^*} = 0 \tag{H.120}$$

Then, when $\theta \in (0, 1)$, we have:

$$\boldsymbol{w}_{\alpha(\theta)} := \arg\min_{\boldsymbol{w}\in\mathcal{W},||\boldsymbol{w}||_2\leq C} \theta\mathcal{L}_{\mathcal{K}}(\boldsymbol{w},\mathcal{D}_f) + (1 - \theta)\mathcal{L}_{\mathcal{A}}(\boldsymbol{w},\mathcal{D}_r) + \frac{\lambda}{2}||\boldsymbol{w}||_2^2 \tag{H.121}$$

which yields the first order condition:

$$\theta\nabla_{\boldsymbol{w}_{\alpha(\theta)},\mathcal{K}} + (1 - \theta)\nabla_{\boldsymbol{w}_{\alpha(\theta)},\mathcal{A}} + \lambda\boldsymbol{w}_{\alpha(\theta)} = 0 \tag{H.122}$$

Subtracting Eq. (H.120) from Eq. (H.122) yields:

$$\theta\nabla_{\boldsymbol{w}_{\alpha(\theta)},\mathcal{K}} + (1 - \theta)\nabla_{\boldsymbol{w}_{\alpha(\theta)},\mathcal{A}} + \lambda\boldsymbol{w}_{\alpha(\theta)} - (1 - \theta)\nabla_{\boldsymbol{w}_{\alpha^*},\mathcal{A}} - (1 - \theta)\lambda\boldsymbol{w}_{\alpha^*} = 0 \tag{H.123}$$

which simplifies to:

$$\theta\nabla_{\boldsymbol{w}_{\alpha(\theta)},\mathcal{K}} + (1 - \theta)(\nabla_{\boldsymbol{w}_{\alpha(\theta)},\mathcal{A}} - \nabla_{\boldsymbol{w}_{\alpha^*},\mathcal{A}}) + \lambda(\boldsymbol{w}_{\alpha(\theta)} - \boldsymbol{w}_{\alpha^*}) = -\theta\lambda\boldsymbol{w}_{\alpha^*} \tag{H.124}$$

The fundamental theorem of calculus then yields:

$$\int_0^1 \boldsymbol{H}_{\boldsymbol{w}_{\alpha^*}+t(\boldsymbol{w}_{\alpha(\theta)}+\boldsymbol{w}_{\alpha^*}),\mathcal{A}}(\boldsymbol{w}_{\alpha(\theta} - \boldsymbol{w}_{\alpha^*})dt = \nabla_{\boldsymbol{w}_{\alpha(\theta)},\mathcal{A}} - \nabla_{\boldsymbol{w}_{\alpha^*},\mathcal{A}} \tag{H.125}$$

and

$$\int_0^1 \boldsymbol{H}_{\boldsymbol{w}_{\alpha^*}+t(\boldsymbol{w}_{\alpha(\theta)}+\boldsymbol{w}_{\alpha^*}),\mathcal{K}}(\boldsymbol{w}_{\alpha(\theta)} - \boldsymbol{w}_{\alpha^*})dt = \nabla_{\boldsymbol{w}_{\alpha(\theta)},\mathcal{K}} - \nabla_{\boldsymbol{w}_{\alpha^*},\mathcal{K}} \tag{H.126}$$

We thus denote:

$$\bar{\boldsymbol{H}}_{\mathcal{K}} := \int_0^1 \boldsymbol{H}_{\boldsymbol{w}_{\alpha^*}+t(\boldsymbol{w}_{\alpha(\theta)}+\boldsymbol{w}_{\alpha^*}),\mathcal{K}}dt \tag{H.127}$$

$$\bar{\boldsymbol{H}}_{\mathcal{A}} := \int_0^1 \boldsymbol{H}_{\boldsymbol{w}_{\alpha^*}+t(\boldsymbol{w}_{\alpha(\theta)}+\boldsymbol{w}_{\alpha^*}),\mathcal{A}}dt \tag{H.128}$$

$$\Delta\boldsymbol{w} := \boldsymbol{w}_{\alpha(\theta)} - \boldsymbol{w}_{\alpha^*} \tag{H.129}$$

Incorporating Eq. (H.125) and Eq. (H.126) into Eq. (H.124) then yields:

$$\theta(\nabla_{\boldsymbol{w}_{\alpha^*},\mathcal{K}} + \bar{\boldsymbol{H}}_{\mathcal{K}}\Delta\boldsymbol{w}) + (1 - \theta)(\nabla_{\boldsymbol{w}_{\alpha^*},\mathcal{A}} + \bar{\boldsymbol{H}}_{\mathcal{A}}\Delta\boldsymbol{w} - \nabla_{\boldsymbol{w}_{\alpha^*},\mathcal{A}}) + \lambda\Delta\boldsymbol{w} = -\theta\lambda\boldsymbol{w}_{\alpha^*}$$

$$\iff \theta\nabla_{\boldsymbol{w}_{\alpha^*},\mathcal{K}} + \theta\bar{\boldsymbol{H}}_{\mathcal{K}}\Delta\boldsymbol{w} + (1 - \theta)\bar{\boldsymbol{H}}_{\mathcal{A}}\Delta\boldsymbol{w} + \lambda\Delta\boldsymbol{w} + \theta\lambda\boldsymbol{w}_{\alpha^*} = 0$$

$$\iff \big(\theta\bar{\boldsymbol{H}}_{\mathcal{K}} + (1 - \theta)\bar{\boldsymbol{H}}_{\mathcal{A}} + \lambda\boldsymbol{I}\big)\Delta\boldsymbol{w} = -\theta\big(\nabla_{\boldsymbol{w}_{\alpha^*},\mathcal{K}} + \lambda\boldsymbol{w}_{\alpha^*}\big)$$

$$\iff \big(\boldsymbol{H}_{\boldsymbol{w}_{\alpha^*},\mathcal{K},\mathcal{A}} + \lambda\boldsymbol{I} + \theta(\bar{\boldsymbol{H}}_{\mathcal{K}} - \boldsymbol{H}_{\boldsymbol{w}_{\alpha^*},\mathcal{K}}) + (1 - \theta)(\bar{\boldsymbol{H}}_{\mathcal{A}} - \boldsymbol{H}_{\boldsymbol{w}_{\alpha^*},\mathcal{A}})\big)\Delta\boldsymbol{w}$$
$$= -\theta\big(\nabla_{\boldsymbol{w}_{\alpha^*},\mathcal{K}} + \lambda\boldsymbol{w}_{\alpha^*}\big)$$

$$\iff \big(\boldsymbol{H}_{\boldsymbol{w}_{\alpha^*},\mathcal{K},\mathcal{A}} + \lambda\boldsymbol{I}\big)\Delta\boldsymbol{w} = -\big(\theta(\bar{\boldsymbol{H}}_{\mathcal{K}} - \boldsymbol{H}_{\boldsymbol{w}_{\alpha^*},\mathcal{K}}) + (1 - \theta)(\bar{\boldsymbol{H}}_{\mathcal{A}} - \boldsymbol{H}_{\boldsymbol{w}_{\alpha^*},\mathcal{A}})\big)\Delta\boldsymbol{w}$$
$$- \theta\big(\nabla_{\boldsymbol{w}_{\alpha^*},\mathcal{K}} + \lambda\boldsymbol{w}_{\alpha^*}\big). \tag{H.130}$$

Then, note that:

$$||\bar{\boldsymbol{H}}_{\mathcal{K}} - \boldsymbol{H}_{\boldsymbol{w}_{\alpha^*},\mathcal{K}}||_2 = ||\int_0^1 \boldsymbol{H}_{\boldsymbol{w}_{\alpha^*}+t\Delta\boldsymbol{w},\mathcal{K}}dt - \boldsymbol{H}_{\boldsymbol{w}_{\alpha^*},\mathcal{K}}||_2 \tag{H.131}$$

$$\leq \int_0^1 ||\boldsymbol{H}_{\boldsymbol{w}_{\alpha^*}+t\Delta\boldsymbol{w},\mathcal{K}} - \boldsymbol{H}_{\boldsymbol{w}_{\alpha^*},\mathcal{K}}||_2 \tag{H.132}$$

$$\leq \frac{F_{\mathcal{K}}}{2}||\Delta\boldsymbol{w}||_2 \tag{H.133}$$

by the same token as in Prop. G.3.

Similarly:

$$||\bar{\boldsymbol{H}}_{\mathcal{A}} - \boldsymbol{H}_{\boldsymbol{w}_{\alpha^*},\mathcal{A}}||_2 \leq \frac{F_{\mathcal{A}}}{2}||\Delta\boldsymbol{w}||_2 \tag{H.134}$$

Also:

$$||\nabla_{\boldsymbol{w}_{\alpha^*},\mathcal{K}}||_2 = ||\nabla_{\boldsymbol{w}_{\alpha^*},\mathcal{K}} - \nabla_{\mathcal{K}(\mathcal{D}_f),\mathcal{K}}||_2 \tag{H.135}$$
$$\leq P_{\mathcal{K}}||\boldsymbol{w}_{\alpha^*} - \boldsymbol{w}_{\mathcal{K}(\mathcal{D}_f)}||_2 \tag{H.136}$$
$$\leq 2P_{\mathcal{K}}C \tag{H.137}$$

by definition of the uniform learner $\mathcal{K}$, Lemma H.1, and the triangle inequality.

Additionally:

$$||\lambda\boldsymbol{w}_{\alpha^*}||_2 \leq \lambda C \tag{H.138}$$

By the triangle inequality, incorporating Eq. (H.133), Eq. (H.134), Eq. (H.137), and Eq. (H.138) into Eq. (H.130), we have that:

$$||(\boldsymbol{H}_{\boldsymbol{w}_{\alpha^*},\mathcal{K},\mathcal{A}} + \lambda\boldsymbol{I})\Delta\boldsymbol{w}||_2 \leq (\theta||\bar{\boldsymbol{H}}_{\mathcal{K}} - \boldsymbol{H}_{\boldsymbol{w}_{\alpha^*},\mathcal{K}}||_2 + (1-\theta)||\bar{\boldsymbol{H}}_{\mathcal{A}} - \tag{H.139}$$

$$\boldsymbol{H}_{\boldsymbol{w}_{\alpha^*},\mathcal{A}}||_2)||\Delta\boldsymbol{w}||_2 + \theta||\nabla_{\boldsymbol{w}_{\alpha^*},\mathcal{K}}||_2 + \theta||\lambda\boldsymbol{w}_{\alpha^*}||_2 \tag{H.140}$$

$$\leq (\theta F_{\mathcal{K}} + (1-\theta)F_{\mathcal{A}})||\Delta\boldsymbol{w}||_2^2 + \theta C(2P_{\mathcal{K}} + \lambda) \tag{H.141}$$

Note that we have, where $\sigma_{\min}(\cdot)$ denotes the minimum singular value:

$$||(\boldsymbol{H}_{\boldsymbol{w}_{\alpha^*},\mathcal{K},\mathcal{A}} + \lambda\boldsymbol{I})\Delta\boldsymbol{w}||_2 \geq \sigma_{\min}(\boldsymbol{H}_{\boldsymbol{w}_{\alpha^*},\mathcal{K},\mathcal{A}} + \lambda\boldsymbol{I})||\Delta\boldsymbol{w}||_2 \text{ by property of op. norm} \tag{H.142}$$

$$= \lambda_{\min}(\boldsymbol{H}_{\boldsymbol{w}_{\alpha^*},\mathcal{K},\mathcal{A}} + \lambda\boldsymbol{I})||\Delta\boldsymbol{w}||_2 \text{ by Lemma H.3} \tag{H.143}$$

Furthermore, by Lemma H.1, we have that $||\boldsymbol{H}_{\boldsymbol{w}_{\alpha^*},\mathcal{K}}||_2 \leq P_{\mathcal{K}}$ and $||\boldsymbol{H}_{\boldsymbol{w}_{\alpha^*},\mathcal{A}}||_2 \leq P_{\mathcal{A}}$, which yields:

$$||\boldsymbol{H}_{\boldsymbol{w}_{\alpha^*},\mathcal{K},\mathcal{A}}||_2 \leq \theta P_{\mathcal{K}} + (1-\theta)P_{\mathcal{A}} \tag{H.144}$$

which by Lemma H.3 yields:

$$\lambda_{\min}(\boldsymbol{H}_{\alpha^*,\mathcal{K},\mathcal{A}} + \lambda\boldsymbol{I}) \in [\lambda - \theta P_{\mathcal{K}} - (1-\theta)P_{\mathcal{A}}, \mu + \theta P_{\mathcal{K}} + (1-\theta)P_{\mathcal{A}}] \tag{H.145}$$

With Eq. (H.143), this yields that:

$$||(\boldsymbol{H}_{\boldsymbol{w}_{\alpha^*},\mathcal{K},\mathcal{A}} + \lambda\boldsymbol{I})\Delta\boldsymbol{w}||_2 \geq (\lambda - \theta P_{\mathcal{K}} - (1-\theta)P_{\mathcal{A}})||\Delta\boldsymbol{w}||_2 \tag{H.146}$$

Incorporating this into Eq. (H.141) yields:

$$(\lambda - \theta P_{\mathcal{K}} - (1-\theta)P_{\mathcal{A}})||\Delta \boldsymbol{w}||_2 \le (\theta F_{\mathcal{K}} + (1-\theta)F_{\mathcal{A}})||\Delta \boldsymbol{w}||_2^2 + \theta C(2P_{\mathcal{K}} + \lambda) \quad \text{(H.147)}$$

Simplifying yields the quadratic inequality:

$$(\theta F_{\mathcal{K}} + (1-\theta)F_{\mathcal{A}})||\Delta \boldsymbol{w}||_2^2 - (\lambda - \theta P_{\mathcal{K}} - (1-\theta)P_{\mathcal{A}})||\Delta \boldsymbol{w}||_2 + \theta C(2P_{\mathcal{K}} + \lambda) \ge 0 \quad \text{(H.148)}$$

This then yields that:

$$||\Delta \boldsymbol{w}||_2 \le \frac{\lambda - P - \sqrt{(\lambda - P)^2 - 4\theta CF(2P_{\mathcal{K}} + \lambda)}}{2F} \quad \text{(H.149)}$$

This is only valid when:

$$(\lambda - P)^2 - 4\theta CF(2P_{\mathcal{K}} + \lambda) \ge 0 \quad \text{(H.150)}$$
$$\iff \lambda^2 - 2L\lambda + P^2 - 4\theta C2P_{\mathcal{K}}F - 4\theta C\lambda F \ge 0 \quad \text{(H.151)}$$
$$\iff \lambda^2 - 2P\lambda - 4\theta CF\lambda + P^2 - 8\theta CP_{\mathcal{K}}F \ge 0 \quad \text{(H.152)}$$
$$\iff \lambda^2 - (2P + 4\theta CF)\lambda + (P^2 - 8\theta CP_{\mathcal{K}}F) \ge 0 \quad \text{(H.153)}$$
$$\impliedby \lambda \ge P + 2\theta CF + \sqrt{2\theta CF(P + 2\theta CF + 8P_{\mathcal{K}}} \quad \text{(H.154)}$$

which holds by assumption. Note that all components of $2\theta CF + \sqrt{2\theta CF(P + 2\theta CF + 8P_{\mathcal{K}}}$ are nonnegative, rendering this valid. Incorporating Eq. (H.149) into Lemma H.2 yields the final bound as desired. $\qquad\square$

Then, theorem 5.5 follows as a corollary of theorem H.11:

*Proof.* Note that we take care to ensure the bound holds for any choice of $\theta \in [0,1]$. Hence, fix $\theta \in [0,1]$.

Let

$$a := \lambda - P > 0, \qquad \varepsilon := 4\theta CF(2P_K + \lambda), \qquad \Delta := \frac{a - \sqrt{a^2 - \varepsilon}}{2F}. \quad \text{(H.155)}$$

Theorem H.11 gives the inequality:

$$|\alpha^* - \alpha(\theta)| \le \frac{P_K}{2}\Delta^2 + \lambda C\Delta. \quad \text{(H.156)}$$

By the condition on $\lambda$ in the theorem, the square root is real for every $\theta \in [0,1]$ (i.e. $a^2 - \varepsilon \ge 0$). This yields:

$$a - \sqrt{a^2 - \varepsilon} = \frac{\varepsilon}{a + \sqrt{a^2 - \varepsilon}}. \quad \text{(H.157)}$$

Then, since $a + \sqrt{a^2 - \varepsilon} \ge a > 0$, (H.157) implies:

$$a - \sqrt{a^2 - \varepsilon} \le \frac{\varepsilon}{a}. \quad \text{(H.158)}$$

Dividing (H.158) by $2F$ yields:

$$\Delta \le \frac{\varepsilon}{2aF}. \quad \text{(H.159)}$$

Then, substituting $\varepsilon = 4\theta CF(2P_K + \lambda)$ from (H.155) yields:

$$\Delta \le \frac{4\theta CF(2P_K + \lambda)}{2aF} = \frac{2\theta C(2P_K + \lambda)}{a}. \tag{H.160}$$

We now bound the two terms on the right-hand side of (H.156). Using (H.160), we have that:

$$\lambda C \, \Delta \le \lambda C \cdot \frac{2\theta C(2P_K + \lambda)}{a} = \frac{2\lambda(2P_K + \lambda)}{a} \, C^2\theta = \mathcal{O}(\lambda C^2\theta), \tag{H.161}$$

$$\frac{P_K}{2}\Delta^2 \le \frac{P_K}{2}\left(\frac{2\theta C(2P_K + \lambda)}{a}\right)^2 = \frac{2P_K(2P_K + \lambda)^2}{a^2} \, C^2\theta^2 = \mathcal{O}(C^2\theta^2). \tag{H.162}$$

Combining (H.156), (H.161) and (H.162) and absorbing constants (which are independent of $\theta \in [0, 1]$) yields:

$$|\alpha^* - \alpha(\theta)| = \mathcal{O}\big(\lambda C^2\theta + C^2\theta^2\big), \qquad \text{for any } \theta \in [0, 1]. \tag{H.163}$$

as desired. $\qquad\square$

# I  ONLINE ALGORITHM

We also consider the online setting, where users send requests in sequential order (Nguyen et al., 2022). Here, we denote $\mathcal{D}_{f_k}$ as the forget set after the $k$-th request and the associated retain set as $\mathcal{D}_{r_k} = \mathcal{D} \setminus \cup_{i=1}^k \mathcal{D}_{f_i}$. Letting $\tilde{\boldsymbol{w}}_0 = \mathcal{A}(\mathcal{D})$, we estimate $\tilde{\boldsymbol{w}}_k$ recursively as $\tilde{\boldsymbol{w}}_k = \tilde{\boldsymbol{w}}_{k-1} - \frac{\tilde{\boldsymbol{H}}_{n,\lambda,k-1}^{-1}}{H}\nabla_{\tilde{\boldsymbol{w}}_{k-1},\mathcal{K},\mathcal{A}}$, where $\tilde{\boldsymbol{H}}_{n,\lambda,k-1}^{-1}$ is an estimator for $(\boldsymbol{H}_{\tilde{\boldsymbol{w}}_{k-1},\mathcal{K},\mathcal{A}} + \lambda\boldsymbol{I})^{-1}$ with respect to $\mathcal{D}_{f_k}$ and $\mathcal{D}_{r_k}$. $\nabla_{\tilde{\boldsymbol{w}}_{k-1},\mathcal{K},\mathcal{A}}$ is also computed with respect to $\mathcal{D}_{f_k}$ and $\mathcal{D}_{r_k}$. Adding noise to $\tilde{\boldsymbol{w}}_k$ as stipulated in theorem G.1 yields a $\boldsymbol{w}_k^-$ satisfying Def. 4.6. Furthermore, we have that:

**Proposition I.1.** *Let $\lambda_{\min}$ be the smallest eigenvalue of $\boldsymbol{H}_{\tilde{\boldsymbol{w}}_{k-1},\mathcal{K},\mathcal{A}}$, $\lambda > ||\boldsymbol{H}_{\boldsymbol{w}^*,\mathcal{K},\mathcal{A}}||_2$, and $||\nabla_{\tilde{\boldsymbol{w}}_k,\mathcal{K},\mathcal{A}}||_2, ||\nabla_{\tilde{\boldsymbol{w}}_k,\mathcal{K},\mathcal{A}}||_2 \le G$ for all $k$, all evaluated with respect to $\mathcal{D}_{f_k}$ and $\mathcal{D}_{r_k}$. Then, the bound in theorem G.5 is identical in the online setting.*

*Proof:*  See Appx. H.9.

In the online setting, Prop. I.1 yields Alg. 4.

Note that, for simplicity, we set $b = 1$. However, they can be added similarly to Alg. 3 if the user desires.

# J  ELIMINATING HYPERPARAMETERS IN CERTIFIED ALGORITHMS

Here, we summarize how to eliminate hyperparameters in Alg. 2, Alg. 3, and Alg. 4.

- $\lambda_{\min}$ can be chosen as 0 by convex approximation, or it can be estimated using simple algorithms like Gershgorin's circle theorem or inverse power iteration.
- By Lemma H.1, $\lambda$ can be chosen as $\theta P_{\mathcal{K}} + (1 - \theta)P_{\mathcal{A}}$
- Similarly, by Lemma H.1, $H$ can be chosen as $2\lambda$
- In practice, since $\frac{B}{\lambda + \lambda_{\min}}$ offers a bound on $\hat{\kappa}_l$ by Lemma H.6, $\zeta_{\min}$ can be chosen as $\lambda + \lambda_{\min}$
- By Lemma H.7, $n$ can be chosen as $n = 2\frac{B}{\lambda + \lambda_{\min}} \ln(\frac{B}{\lambda + \lambda_{\min}}b)$
- $G$ can be approximated by computing $||\nabla_{\boldsymbol{w}^*,\mathcal{K},\mathcal{A}}||_2$.
- $\theta$ can be chosen with Cor. 5.2 to satisfy a particular closeness to uniformity.
- In practice, we find that $C$ can be chosen as 10, 20, or 100.
- $b$ can be chosen to satisfy a particular concentration on the estimator, so we let it be free. However, one can set $b = 1$.
- Common heuristics for $\varepsilon$ and $\delta$ are available in the differential privacy literature.
- In practice, following what is common in certified unlearning e.g. in (Zhang et al., 2024), the Lipchitz constants in Asm. 5.3 and Asm. 5.4 are treated as hyperparameters. However, in practice, they can all be set to 1.

---

**Algorithm 4** Online $(\varepsilon, \delta, \theta)$-certified uniformity with DP

---

**Require:** Dataset $\mathcal{D}$; forget sets $\{\mathcal{D}_{f_1}, ..., \mathcal{D}_{f_k}\}$; pretrained model $\boldsymbol{w}^* = \mathcal{A}(\mathcal{D})$; privacy budgets $\varepsilon$ and $\delta$; ; privacy-utility tradeoff coefficient $\theta$; sample size $n$; local convex coefficient $\lambda$; norm upper bound $C$; cumulative Hessian upper bound $H$; individual Hessian minimum eigenvalue upper bound $\zeta_{\min}$; bound looseness probability $\rho$.

$\quad \tilde{\boldsymbol{w}}_0 \leftarrow \boldsymbol{w}^*$
$\quad \mathcal{D}_{r_0} \leftarrow \mathcal{D}$
$\quad$**for** $i = 1, ..., k$ **do**
$\quad\quad \mathcal{D}_{r_i} \leftarrow \mathcal{D}_{r_{i-1}} \setminus \mathcal{D}_{f_i}$
$\quad\quad \boldsymbol{P}_{0,\lambda} \leftarrow \nabla_{\tilde{\boldsymbol{w}}_{i-1}, \mathcal{K}, \mathcal{A}}$
$\quad\quad$**for** $t = 1, ..., n$ **do**
$\quad\quad\quad$ Sample $X_{t_i}$ from $D_{f_i}$ with probability $\theta$ or sample $X_{t_i}$ from $D_r$ with probability $1 - \theta$
$\quad\quad\quad$**if** $X_{t_i} \sim D_f$ **then**
$\quad\quad\quad\quad \boldsymbol{H}_{t,\lambda,i} \leftarrow \nabla_{\boldsymbol{w}}^2 \mathcal{L}_{\mathcal{K}}(\boldsymbol{w}^*, X_{t_i}) + \frac{\lambda \boldsymbol{I}}{2\theta}$
$\quad\quad\quad$**else if** $X_i \sim D_r$ **then**
$\quad\quad\quad\quad \boldsymbol{H}_{t,\lambda} \leftarrow \nabla_{\boldsymbol{w}}^2 \mathcal{L}_{\mathcal{A}}(\boldsymbol{w}^*, X_{t_i}) + \frac{\lambda \boldsymbol{I}}{2(1-\theta)}$
$\quad\quad\quad$**end if**
$\quad\quad\quad \boldsymbol{P}_{t,\lambda,i} = \boldsymbol{P}_{0,\lambda,i} + (\boldsymbol{I} - \frac{\boldsymbol{H}_{t,\lambda,i}}{H}) \boldsymbol{P}_{t-1,\lambda,i}.$
$\quad\quad$**end for**
$\quad\quad \tilde{\boldsymbol{w}}_i \leftarrow \tilde{\boldsymbol{w}}_{i-1} - \frac{\boldsymbol{P}_{n,\lambda,i}}{H}$
$\quad$**end for**
$\quad$ Compute $\Delta$ as the bound in Eq. (G.23).
$\quad \sigma = \frac{\Delta}{k\varepsilon} \sqrt{2 \ln(1.25/\delta)}$
$\quad \boldsymbol{w}^- \leftarrow \tilde{\boldsymbol{w}}_k + Y$ where $Y \sim \mathcal{N}(\boldsymbol{0}, \sigma^2 \boldsymbol{I})$
$\quad$**return** $\boldsymbol{w}^-$

---

# K   EXPERIMENTAL DETAILS

## K.1   DATASET DETAILS

**MNIST:** The MNIST dataset contains 70k 28x28 greyscale images of hand-drawn digits in 10 classes (Deng, 2012). We conduct our experiments with 49k training images and 21k test images. The classes are mutually exclusive.

**Kuzushiji-MNIST:** The Kuzushiji-MNIST (KMNIST) dataset contains 70k 28x28 greyscale images of Japanese kanji in 10 classes (Clanuwat et al., 2018). We conduct our experiments with 49k training images and 21k test images. The classes are mutually exclusive.

**CIFAR10:** The CIFAR10 dataset consists of 60k 32x32 color images in 10 classes. The classes are mutually exclusive and include airplanes, automobiles, birds, cats, deer, dogs, frogs, horses, ships, and trucks (Krizhevsky et al., 2009).

**CIFAR-100:** The CIFAR-100 dataset is similar to CIFAR-10 but contains 100 classes, each with 600 images, making a total of 60k 32x32 color images. The 100 classes are grouped into 20 superclasses, and each image comes with a "fine" label (the class to which it belongs) and a "coarse" label (the superclass to which it belongs) (Krizhevsky et al., 2009).

**SVHN:** The Street View House Numbers (SVHN) dataset (Netzer et al., 2011) contains images of double-digit numbers on house walls as colored 32x32 images. We load SVHN with 100 classes, corresponding to 10 * 10 for each digit. There are 73k training images, 26k testing images.

**TinyImageNet**: The TinyImageNet dataset (Le and Yang, 2015) contains 100000 64x64 images.

## K.2   MODEL DETAILS

**LogReg:** A logistic regression model that has a single linear layer between inputs and outputs, followed by a softmax output function.

**MLP:** A two-layer ReLU feedforward neural network.

**ResNet8:** A [1,1,1,0] residual network, , with standard convolutional blocks, as described in (He et al., 2016).

**ResNet18:** A [2,2,2,2] residual network, with standard convolutional blocks, as described in (He et al., 2016).

**ResNet50:** A [3, 4, 6, 3] residual network, with bottleneck convolutional blocks, as described in (He et al., 2016).

**ViT_S_16:** A vision transformer with $\approx 20$ million parameters, as detailed in (Dosovitskiy et al., 2021).

**ViT_B_16:** A vision transformer with $\approx 80$ million parameters, as detailed in (Dosovitskiy et al., 2021).

### K.3 BASELINE DETAILS

We implement several baselines and provide the rationale for their use below:

**Pretrained:** This is simply the pretrained model corresponding to whichever model and benchmark is specified. The rationale for using this is to demonstrate that we alter uniformity significantly from before without tarnishing accuracy for either the retain or test sets.

**Retrained:** This is a model retrained over the retain set, performing exact unlearning. The rationale for using this is to demonstrate that our methods mimic unlearning in how we preserve accuracy, but induce uniformity in a way that unlearning does not.

**Synthetic:** This method proceeds as follows: for each instance in the forget set, sample $k$ instances from the $\varepsilon$-ball, with respect to the $\ell_2$ norm, around that instance. Then, assign these $k$ instances random labels from the label space, choosing labels uniformly at random. Do this for all forget set instances, yielding $|\mathcal{D}_f|k$ new instances. Then, append this to the retain set and retrain over this augmented dataset. This provides strong accuracy for simple baselines like MNIST while also inducing uniformity, providing an alternative, simple algorithm to compare our method against in terms of time elapsed.

**Label Differential Privacy:** Specifically, we use the multi-stage training method of Ghazi et al. (2021) to obtain a model which is differentially private with respect to the labels–that is, an adversary cannot be sure whether the label they obtain is the true label. This is related to our work, albeit addresses a different threat model, as described in Appx. C. Still, we believe it is important to demonstrate that our method achieves privacy while not sacrificing utility to the extend that label differential privacy does, since it is a well-known method in the privacy literature that addresses a similar problem. For our experiments, we use the official repository with the hyperparameters reported in the paper: `https://github.com/google-research/label-dp`.

### K.4 HYPERPARAMETER DETAILS

Please note that, throughout, we do not do extensive hyperparameter optimization, which may lead to improved performance.

We use a standard train-test split of 70-30 throughout. Pretraining and synthetic training have the same hyperparameters as pretraining, unless mentioned otherwise. We use ADAM, with standard PyTorch hyperparameters aside from learning rate and weight decay, throughout. A batch size of 128 is used for pretraining and also for the retain set in Alg. 1 throughout. To ensure that gradients were well-aligned between the two conflicting objectives in Alg. 1, we implemented and ran gradient surgery (Yu et al., 2020) before each gradient step. Roughly, gradient surgery projects the gradient of one objective onto the normal plane of the other objective's gradient only when the two gradients conflict, i.e. have negative dot product. The projected gradient is chosen uniformly at random between the two. This effectively removes the component of the gradient that would increase the other objective's loss. For all Alg. 1 experiments and Alg. 2 experiments, we use a forget set size of 100 with a batch size (when loading the forget set into the finetuning in Alg. 1) of 10. We perform Alg. 1 for 100 epochs and finetune Alg. 2 for 50 epochs before running the certified Newton step. We generally use the forward KL divergence between model softmax outputs and the uniform distribution for $\mathcal{L}_{\mathcal{K}}$ and the cross entropy between model predictions and ground truth labels for $\mathcal{L}_{\mathcal{A}}$. For LogReg,

we instead use the square loss between the uniform softmax probabilities and the model softmax outputs, since the forward KL is not necessarily convex in $w$ in this case, while the square loss is; this allows us to use Alg. 2 with small $\lambda$. For our synthetic baseline, we use $\varepsilon = 8/255$ throughout, where $\varepsilon$ is the size of the $\varepsilon$-ball where we sample instances to assign random labels for retraining. For the LabelDP baseline, we use the multi-stage training algorithm of Ghazi et al. (2021) throughout.

Early stopping is implemented by saving the model which first meets the early stopping conditions, and continuing to see if any model performs better in terms of confidence distance while still meeting the early stopping conditions specified below.

**Compute:** We use two RTX 6000 Ada Generation NVIDIA GPUs throughout. The most resource intensive experiments are the LabelDP experiments, which take up most of the memory on both GPUs. Besides those, the other experiments take up at most a fourth of the compute resources available on one GPU. No experiments ran required more compute than these two GPUs provide.

**MNIST, LogReg Pretraining:** Epochs: 25. Learning rate: 0.01.

**MNIST, MLP Pretraining:** Epochs: 5. Learning rate: 0.01.

**MNIST, ResNet18 Pretraining:** Epochs: 2. Learning rate: 0.001.

**MNIST, LogReg Alg. 1:** Learning rate: 0.01

**MNIST, MLP Alg. 1:** Learning rate: 0.01

**MNIST, SVM Alg. 1:** Epochs: 100. Learning rate: 0.1. Optimized with SGD.

**MNIST, ResNet18 Alg. 1:** Learning rate: 0.001. Early stopping criterion of a confidence distance $< 0.32$ and a retain accuracy of $> 90\%$.

**MNIST, LogReg Alg. 2:** $M = 1$. $C = 10$, pretrained with PGD with the same hyperparameters as the standard pretraining. Since the losses are convex in $w$, $\lambda_{\min} = 0$. $\lambda = 0.0001$. Following Zhang et al. (2024), we use the variance $\sigma^2$ as a hyperparameter, corresponding to a broad range of choices of $\varepsilon$ and $\delta$. We choose $\sigma = 0.001$. This results in large $\varepsilon$ when choosing small $\delta$, as typical in differential privacy (Dwork et al., 2014) and certified unlearning (Qiao et al., 2025). However, we still observe good induced uniformity. We also use a learning rate of $\alpha = 0.5$ in the Newton step, which, following Appx. G and bounding the gradient with $\alpha F$ instead of $F$, results in roughly twice as large of an $\varepsilon$.

**MNIST, LogReg Alg. 3:** $\lambda = 1.5, J = 4.0, n = 1000, b = 100$, and otherwise same hyperparameters as Alg. 2.

**MNIST, MLP Alg. 3:** $\lambda = 1.4, J = 2.5, n = 1000, b = 100$, and a learning rate of $\alpha = 0.35$.

**MNIST, SVM Alg. 3:** $\lambda = 0.5, J = 2.5, n = 5000, b = 200$, and a learning rate of $\alpha = 0.05$. We also used a twice-differentiable surrogate for the hinge loss to replace the standard hinge loss, which was used as the pretraining loss $\mathcal{L}_{\mathcal{A}}$.

**MNIST, LogReg Synthetic Baseline:** Sampled $k$ instances for each forget set instance: 5.

**MNIST, ResNet18 Synthetic Baseline:** Sampled $k$ instances for each forget set instance: 500.

**MNIST, LogReg LabelDP Baseline:** Epochs: 200. Batch size: 256. Random flip, random left-right flip, and random cutout (8). SGD with learning rate 0.4 with momentum 0.9. $\varepsilon = 2.0$. Mixup for stage 1: 16. Mixup for stage 2: 8. Data split evenly between the two stages. Piecewise constant learning rate scheduler. These hyperparameters are chosen to match those in the best results of Ghazi et al. (2021). See Ghazi et al. (2021) for more details on these hyperparameters.

**MNIST, ResNet18 LabelDP Baseline:** Same as the MNIST LogReg LabelDP hyperpameters, except with a weight decay of 0.0005 throughout.

**KMNIST, LogReg Pretraining:** Epochs: 100. Learning rate: 0.01.

**KMNIST, MLP Pretraining:** Epochs: 100. Learning rate: 0.001.

**KMNIST, ResNet18 Pretraining:** Epochs: 12 Learning rate: 0.002.

**KMNIST, LogReg Alg. 1:** Same as pretraining.

**KMNIST, MLP Alg. 1:** Learning rate: 0.01.

**KMNIST, ResNet18 Alg. 1:** Learning rate: 0.002. Early stopping criterion of a confidence distance $< 0.32$ and a retain accuracy of $> 99\%$.

**KMNIST, ResNet18 Synthetic Baseline:** Sampled $k$ instances for each forget set instance: 500.

**KMNIST, ResNet18 LabelDP Baseline:** Same as the MNIST ResNet18 LabelDP hyperparameters.

**SVHN, ResNet50 Pretraining:** Epochs: 150. Learning rate: 0.001. Weight decay: 0.00005.

**SVHN, ResNet50 Alg. 1:** Same as pretraining.

**SVHN, ResNet50 Synthetic Baseline:** Sampled $k$ instances for each forget set instance: 500.

**SVHN, ResNet50 LabelDP Baseline:** Same as MNIST ResNet18 LabelDP hyperparameters.

**CIFAR10, ResNet18 Pretraining:** Epochs: 200 with SGD with a momentum of 0.9. Learning rate: 0.1. Weight decay: 0.0005.

**CIFAR10, ResNet18 Alg. 1:** Same as pretraining. Early stopping criterion of a confidence distance $< 0.42$ and a retain accuracy of $> 87\%$.

**CIFAR10, ResNet50 Pretraining:** Same as CIFAR10 ResNet18.

**CIFAR10, ResNet50 Alg. 1:** Same as pretraining. Early stopping criterion of a confidence distance $< 0.42$ and a retain accuracy of $> 87\%$.

**CIFAR10, ResNet8 Pretraining:** Same as CIFAR10 ResNet18.

**CIFAR10, ResNet8 Alg. 1:** Same as CIFAR10 ResNet18.

**CIFAR10, ResNet18 Synthetic Baseline:** Sampled $k$ instances for each forget set instance: 5000.

**CIFAR10, ResNet50 Synthetic Baseline:** Sampled $k$ instances for each forget set instance: 5000.

**CIFAR10, ResNet18 LabelDP Baseline:** Same as MNIST ResNet18 LabelDP hyperparameters, except with a batch size of 512.

**CIFAR10, ResNet50 LabelDP Baseline:** Same as CIFAR10 ResNet18 LabelDP.

**CIFAR10, ViT_S_16 Finetuning:** 8 epochs. Learning rate 0.0001 with AdamW.

**CIFAR10, ViT_B_16 Finetuning:** 10 epochs. Learning rate 0.0001 with AdamW.

**CIFAR100, ResNet50 Pretraining:** Same as CIFAR10 ResNet18.

**CIFAR100, ResNet50 Alg. 1:** Same as pretraining. Early stopping criterion of a confidence distance $< 0.42$ and a retain accuracy of $> 87\%$.

**CIFAR100, ResNet8 Pretraining:** Same as CIFAR100 ResNet50.

**CIFAR100, ResNet8 Alg. 1:** Same as CIFAR100 ResNet50.

**CIFAR100, ResNet50 Synthetic Baseline:** Sampled instances: 5000.

**CIFAR100, ResNet50 LabelDP Baseline:** Same as CIFAR10 ResNet18 LabelDP. Please note that our results differ from the results reported in the original paper of Ghazi et al. (2021); however, we verified our results through several runs and used the official paper repository at `https://github.com/google-research/label-dp` with the hyperparameters reported in the paper.

**CIFAR100, ViT_S_16 Finetuning:** 30 epochs. Learning rate 0.002 with SGD with momentum 0.9. 500 warmup steps with cosine scheduler.

**CIFAR100, ViT_S_16 Alg. 1:** 100 epochs. Learning rate 0.001 with SGD.

**CIFAR100, ViT_B_16 Finetuning:** 45 epochs. Learning rate 0.002 with SGD with momentum 0.9. 500 warmup steps with cosine scheduler.

**CIFAR100, ViT_B_16 Alg. 1:** Same as CIFAR100 ViT_S_16.

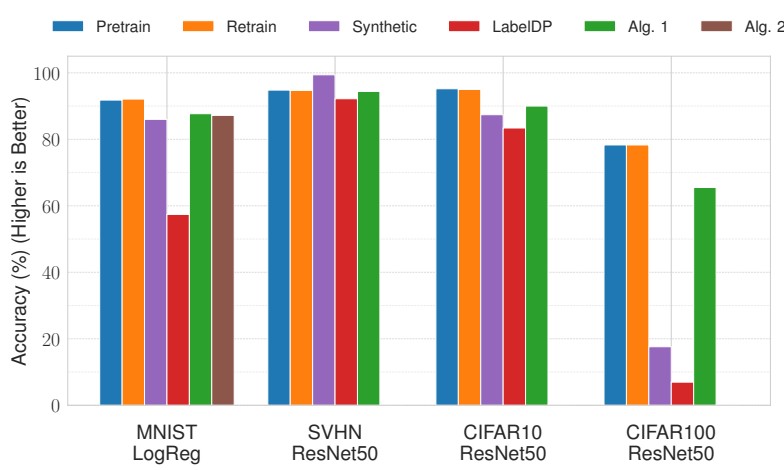

Figure L.5: Accuracy on test set for baselines as well as Alg. 1 and Alg. 2 with $\theta = 0.75$.

**TinyImageNet, ViT_S_16 Finetuning:**  30 epochs. Learning rate 0.0001, momentum 0.9, and weight decay 0.01 with SGD.

**TinyImageNet, ViT_S_16 Alg. 1:**  Same as CIFAR100 ViT_S_16.

**TinyImageNet, ViT_B_16 Finetuning:**  50 epochs. Learning rate 0.0001, momentum 0.9, and weight decay 0.01 with SGD.

**TinyImageNet, ViT_B_16 Alg. 1:**  Same as CIFAR100 ViT_S_16.

**Attacks**: $\alpha = 0.0001$. PGD learning rate: 0.001. PGD steps: 50.

**Test-Set Finetuning**: Finetune pretrained model for 20 more epochs with the same hyperparameters as pretraining, then run Alg. 1 with the same hyperparameters as the original run of Alg. 1. For CIFAR10/CIFAR100, finetune for 100 epochs.

## L  ADDITIONAL EXPERIMENTS

### L.1  TEST SET ACCURACIES FOR MAIN PAPER EXPERIMENTS

We provide a figure, similar to Fig. 2a, for the test set in Fig. L.5. We observe similar results as one does on the retain set, with test accuracies preserved by Alg. 1 and Alg. 2.

### L.2  TABLES FOR MAIN PAPER EXPERIMENTS

The tables for Fig. 2 are in Tab. L.1 and Tab. L.2. We include results for MNIST and KMNIST ResNet18 as well. We see that Alg. 1 induces uniformity without great damage to utility, while all other baselines–including the synthetic baseline–fail to do so without critically harming utility. Furthermore, we observe that ResNet50 performs better than ResNet18, providing more credibility to the claim in Sec. 6 that larger models tend to perform better when used in Alg. 1. Next, we observe that Alg. 1 can actually provide better retain and test accuracy than the pretrained model, as observed for ResNet50 over CIFAR100; this is because we also minimize the retain accuracy during finetuning. We similarly have to use early stopping for Alg. 1, as discussed in Sec. 6, since we use large models. Finally, we observe that LabelDP can induce uniformity, albeit at the cost of retain and test accuracy, but does so not only on the forget set but also the retain set; for a comparison of the confidence distances across the retain, test, and forget sets for LabelDP and our method, please see Tab. L.4. Additionally, for larger, more complex datasets like CIFAR100, LabelDP fails entirely. Please note that we do not perform extensive hyperparameter optimization during pretraining or retraining.

We observe similar results for Alg. 2 in Tab. L.2.

Table L.1: Results for Alg. 1, used in Fig. 2. We find that we are able to induce uniformity while only slightly decreasing retain and test accuracy. $\theta = 0.75$ throughout.

| Dataset | Model | Method | Retain Acc. | Test Acc. | Conf. Dist. (Lower Better) |
|---|---|---|---|---|---|
| MNIST | ResNet18 | Pretrain | 98.0% | 98.1% | 0.877 |
| | | Retrain | 97.3% | 97.1% | 0.876 |
| | | Synthetic | 100.0% | 99.1% | 0.010 |
| | | LabelDP | 98.8% | 98.8% | 0.593 |
| | | Alg. 1 | 99.6% | 99.1% | 0.070 |
| KMNIST | ResNet18 | Pretrain | 98.2% | 92.1% | 0.880 |
| | | Retrain | 98.4% | 92.4% | 0.884 |
| | | Synthetic | 99.9% | 96.7% | 0.019 |
| | | LabelDP | 98.9% | 96.1% | 0.530 |
| | | Alg. 1 | 99.1% | 94.7% | 0.257 |
| SVHN | ResNet50 | Pretrain | 99.6% | 94.8% | 0.980 |
| | | Retrain | 99.3% | 94.7% | 0.964 |
| | | Synthetic | 99.9% | 99.4% | 0.013 |
| | | LabelDP | 92.0% | 92.2% | 0.282 |
| | | Alg. 1 | 99.5% | 94.4% | 0.280 |
| CIFAR10 | ResNet18 | Pretrain | 100.0% | 95.3% | 0.898 |
| | | Retrain | 100.0% | 95.3% | 0.891 |
| | | Synthetic | 94.0% | 89.7% | 0.844 |
| | | LabelDP | 85.8% | 83.6% | 0.359 |
| | | Alg. 1 | 89.6% | 83.1% | 0.377 |
| | ResNet50 | Pretrain | 100.0% | 95.2% | 0.900 |
| | | Retrain | 100.0% | 95.0% | 0.891 |
| | | Synthetic | 91.4% | 87.4% | 0.818 |
| | | LabelDP | 85.5% | 83.4% | 0.334 |
| | | Alg. 1 | 94.7% | 90.0% | 0.270 |
| CIFAR100 | ResNet50 | Pretrain | 100.0% | 78.3% | 0.902 |
| | | Retrain | 100.0% | 78.3% | 0.765 |
| | | Synthetic | 17.7% | 17.6% | 0.189 |
| | | LabelDP | 8.41% | 6.95% | 0.203 |
| | | Alg. 1 | 91.4% | 65.5% | 0.298 |

Table L.2: Results for Alg. 2 for logistic regression trained over MNIST, used in Fig. 2. $\theta = 0.75$ throughout.

| Method | Retain Acc. | Test Acc. | Conf. Dist. (Lower Better) |
|---|---|---|---|
| Pretrain | 92.1% | 91.8% | 0.807 |
| Retrain | 92.0% | 92.1% | 0.807 |
| Synthetic | 86.4% | 86.0% | 0.313 |
| LabelDP | 57.1% | 57.4% | 0.125 |
| Alg. 1 | 87.8% | 87.7% | 0.180 |
| Alg. 2 | 87.1% | 87.2% | 0.280 |

### L.3 ADDITIONAL EXPERIMENTS ON TINYIMAGENET & VIT

We provide experimental results for Alg. 1 for ViT trained on CIFAR100 and TinyImageNet in Tab. L.3, observing similar behavior–in fact significantly lower confidence distance with little retain or test accuracy reduction–when compared to in Tab. L.1.

Table L.3: Results for Alg. 1 for ViT trained on CIFAR100 and TinyImageNet.

| Dataset | Model | Method | Retain Acc. | Test Acc. | Conf. Dist. (Lower Better) |
|---|---|---|---|---|---|
| CIFAR100 | ViT_S_16 | Pretrain | 95.2% | 90.1% | 0.942 |
| | | Retrain | 93.4% | 89.1% | 0.883 |
| | | Synthetic | 86.4% | 84.8% | 0.682 |
| | | Alg. 1 | 91.6% | 85.1% | 0.036 |
| | ViT_B_16 | Pretrain | 94.2% | 91.2% | 0.972 |
| | | Retrain | 95.2% | 91.0% | 0.952 |
| | | Synthetic | 17.7% | 17.6% | 0.189 |
| | | Alg. 1 | 91.6% | 88.6% | 0.074 |
| TinyImageNet | ViT_S_16 | Pretrain | 86.7% | 84.4% | 0.698 |
| | | Retrain | 87.2% | 84.4% | 0.742 |
| | | Synthetic | 87.9% | 86.3% | 0.833 |
| | | Alg. 1 | 84.2% | 81.4% | 0.057 |
| | ViT_B_16 | Pretrain | 95.0% | 91.7% | 0.822 |
| | | Retrain | 96.8% | 90.6% | 0.826 |
| | | Synthetic | 98.0% | 84.4% | 0.830 |
| | | Alg. 1 | 91.8% | 88.3% | 0.037 |
| | ResNet50 | Pretrain | 91.2% | 83.5% | 0.812 |
| | | Retrain | 91.6% | 80.9% | 0.924 |
| | | Synthetic | 92.1% | 80.6% | 0.569 |
| | | Alg. 1 | 92.1% | 81.6% | 0.197 |

Table L.4: A comparison of the confidence distances on the retain, test, and forget sets between Alg. 1 and LabelDP. In general, we induce uniformity on only the forget set, while maintaining confidently correct predictions on the retain and test sets, while LabelDP falls short. Note that this is only for one experiment run.

| Dataset | Model | Method | Retain Conf. Dist. (Higher Better) | Test Conf. Dist. (Higher Better) | Forget Conf. Dist. (Lower Better) |
|---|---|---|---|---|---|
| MNIST | ResNet18 | LabelDP | 0.579 | 0.577 | 0.593 |
| | | Alg. 1 | 0.503 | 0.509 | 0.070 |
| KMNIST | ResNet18 | LabelDP | 0.495 | 0.466 | 0.530 |
| | | Alg. 1 | 0.870 | 0.828 | 0.257 |
| SVHN | ResNet50 | LabelDP | 0.288 | 0.285 | 0.282 |
| | | Alg. 1 | 0.888 | 0.855 | 0.280 |
| CIFAR10 | ResNet18 | LabelDP | 0.371 | 0.365 | 0.359 |
| | | Alg. 1 | 0.724 | 0.690 | 0.377 |
| | ResNet50 | LabelDP | 0.366 | 0.361 | 0.334 |
| | | Alg. 1 | 0.725 | 0.701 | 0.270 |
| CIFAR100 | ResNet50 | LabelDP | 0.182 | 0.156 | 0.203 |
| | | Alg. 1 | 0.576 | 0.470 | 0.298 |

## L.4 LABELDP AND ALG. 1 CONFIDENCE DISTANCES FOR RETAIN, TEST, AND FORGET SETS

Here, we present Tab. L.4, which details the confidence distances for the retain, test, and forget sets of our method vs. LabelDP. Not only do we achieve better retain and test accuracy, but also we induce uniformity on *only* the forget set, while LabelDP induces uniformity on the forget, retain, and test sets altogether, functionally the same as adjusting the temperature. This does not suffice for our threat model, since we want to preserve confident predictions on the retain and test sets.

## L.5 PARETO FRONTIER MAIN PAPER TABLE

The results which correspond to Fig. 3b, Fig. 3a, and Fig. L.12 are included in Tab. L.5 and Tab. L.6.

Table L.5: Results for Alg. 1 and Alg. 2 as we explore the Pareto frontier over MNIST, single run for Fig. 3b, Fig. 3a, and Fig. L.12.

| Model | $\theta$ | Retain Acc. | Test Acc. | Conf. Dist. (Lower Better) |
|---|---|---|---|---|
| LogReg | 0.000 | 93.7% | 92.9% | 0.817 |
| | 0.125 | 91.9% | 91.2% | 0.583 |
| | 0.250 | 92.9% | 92.4% | 0.178 |
| | 0.375 | 92.0% | 92.3% | 0.208 |
| | 0.500 | 92.4% | 92.1% | 0.342 |
| | 0.625 | 91.3% | 91.1% | 0.374 |
| | 0.750 | 91.6% | 91.2% | 0.263 |
| | 0.850 | 88.4% | 87.5% | 0.204 |
| | 0.950 | 89.2% | 89.1% | 0.169 |
| Cert. LogReg | 0.000 | 92.5% | 92.2% | 0.738 |
| | 0.125 | 92.2% | 91.3% | 0.573 |
| | 0.250 | 91.6% | 91.5% | 0.324 |
| | 0.375 | 91.4% | 90.5% | 0.327 |
| | 0.500 | 90.5% | 90.6% | 0.300 |
| | 0.625 | 90.6% | 89.7% | 0.451 |
| | 0.750 | 87.1% | 87.2% | 0.280 |
| | 0.850 | 89.3% | 88.4% | 0.206 |
| | 0.950 | 85.0% | 85.7% | 0.092 |
| MLP | 0.000 | 100.0% | 98.1% | 0.893 |
| | 0.125 | 98.3% | 97.6% | 0.399 |
| | 0.250 | 99.8% | 97.4% | 0.068 |
| | 0.375 | 97.4% | 96.6% | 0.247 |
| | 0.500 | 99.5% | 97.1% | 0.037 |
| | 0.625 | 96.4% | 95.8% | 0.166 |
| | 0.750 | 97.5% | 95.7% | 0.037 |
| | 0.850 | 92.5% | 92.8% | 0.096 |
| | 0.950 | 91.3% | 90.0% | 0.029 |
| ResNet18 | 0.000 | 99.6% | 99.4% | 0.896 |
| | 0.125 | 99.3% | 99.1% | 0.892 |
| | 0.250 | 97.2% | 97.5% | 0.427 |
| | 0.375 | 99.6% | 99.3% | 0.586 |
| | 0.500 | 97.0% | 97.0% | 0.357 |
| | 0.625 | 97.0% | 97.0% | 0.357 |
| | 0.750 | 97.0% | 97.0% | 0.151 |
| | 0.850 | 95.7% | 95.4% | 0.234 |
| | 0.950 | 93.6% | 94.0% | 0.288 |

## L.6 OPTIMIZATION DYNAMICS

### L.6.1 EMPIRICAL RESULTS

Upon using Alg. 1, we observe that logistic regression fails to induce uniformity for more complex benchmarks than MNIST, e.g. KMNIST. Logistic regression has poor test accuracy; we thus conclude that a model must be large enough to generalize well in order to have uniformity induced over it without a large cost to retain accuracy. We discuss mathematical intuition for this in Appx. L.6.2.

However, when using Alg. 1 on larger models, we observe that we need early stopping. After achieving a good uniformity-utility tradeoff, large models e.g. ResNet50 on CIFAR10 will powerfully increase accuracy at the cost of uniformity. This is undesired behavior when compared to, for example, a ResNet8 trained on CIFAR10, where we initially increase uniformity at the cost of accuracy but slowly regain accuracy without critically damaging uniformity. We illustrate this in Fig. L.6. Altogether, our method works best for large models with early stopping during finetuning. We characterize this mathematically in Appx. L.6.2 as well.

Table L.6: Results for Alg. 1 as we explore the Pareto frontier over CIFAR10 and CIFAR100 for ResNet50, single run for Fig. 3b, Fig. 3a, and Fig. L.12.

| Dataset | $\theta$ | Retain Acc. | Test Acc. | Conf. Dist. (Lower Better) |
|---|---|---|---|---|
| CIFAR10 | 0.000 | 100.0% | 92.9% | 0.817 |
| | 0.125 | 99.9% | 94.1% | 0.616 |
| | 0.250 | 99.9% | 93.8% | 0.586 |
| | 0.375 | 99.9% | 94.6% | 0.501 |
| | 0.500 | 91.9% | 85.7% | 0.395 |
| | 0.625 | 90.9% | 86.9% | 0.343 |
| | 0.750 | 94.7% | 90.0% | 0.270 |
| | 0.850 | 56.8% | 56.0% | 0.108 |
| | 0.950 | 10.7% | 10.4% | 0.168 |
| CIFAR100 | 0.000 | 92.5% | 92.2% | 0.738 |
| | 0.125 | 99.9% | 77.0% | 0.353 |
| | 0.250 | 99.9% | 77.5% | 0.252 |
| | 0.375 | 99.9% | 77.0% | 0.296 |
| | 0.500 | 99.9% | 77.1% | 0.301 |
| | 0.625 | 90.9% | 70.2% | 0.353 |
| | 0.750 | 91.4% | 65.5% | 0.298 |
| | 0.85 | 21.2% | 20.6% | 0.138 |
| | 0.950 | 40.8% | 30.3% | 0.019 |

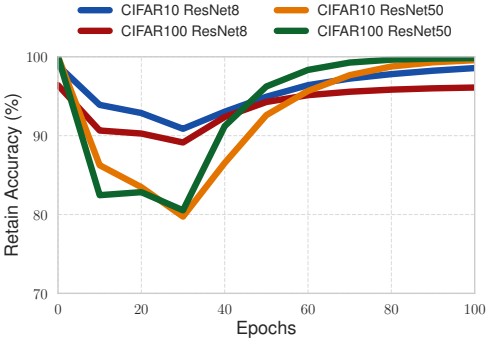

(a) Retain Accuracy vs. Epochs, $\theta = 0.75$

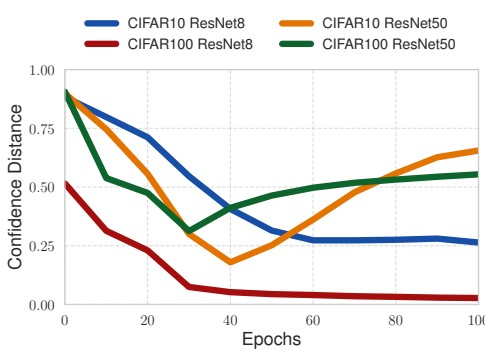

(b) Confidence Distance vs. Epochs, $\theta = 0.75$

Figure L.6: For CIFAR10 and CIFAR100 ResNet50, we observe a sharp drop in confidence distance followed by a sharp increase in Fig. L.6b, in line with the drops and increases for retain accuracy in Fig. L.6a. Test accuracy is similar. This highlights the need for early stopping when using Alg. 1 for large models, since otherwise one escapes from a good privacy-utility tradeoff. For smaller models, e.g. MNIST MLP, this issue does not persist–we obtain good uniformity after an initial drop in accuracy, but then increase accuracy and decrease confidence distance simultaneously.

We provide a plot characterizing how test accuracy for Alg. 1 and Alg. 2 applied on CIFAR10 and CIFAR100 for $\theta = 0.75$ changes over 100 epochs in Fig. L.7 for ResNet8 and ResNet50. We observe similar behavior to retain accuracy.

Results as used in Fig. L.6a, Fig. L.6b, and Fig. L.7 are included in Tab. L.7.

### L.6.2 INTUITION FOR EARLY STOPPING

In what follows, we give mathematical justification for the behavior observed in Fig. L.6 and Tab. L.7.

Firstly, recall that random vectors are nearly orthogonal in high dimensions (Vershynin, 2018). In particular, for larger models, the gradients will conflict i.e. point in opposite directions more strongly,

Table L.7: Studying the optimization dynamics of Alg. 1. Used in Fig. L.6.

| Dataset | Model | Epoch | Retain Acc. | Test Acc. | Conf. Dist. (Lower Better) |
|---|---|---|---|---|---|
| CIFAR10 | ResNet8 | 0 | 98.9% | 90.8% | 0.883 |
| | | 10 | 88.9% | 80.4% | 0.713 |
| | | 20 | 90.8% | 83.1% | 0.539 |
| | | 30 | 92.9% | 85.3% | 0.386 |
| | | 40 | 95.4% | 87.3% | 0.295 |
| | | 50 | 96.5% | 88.9% | 0.263 |
| | | 60 | 97.3% | 89.9% | 0.261 |
| | | 70 | 97.9% | 90.0% | 0.295 |
| | | 80 | 98.2% | 90.4% | 0.270 |
| | | 90 | 98.6% | 90.5% | 0.276 |
| | | 100 | 98.9% | 90.9% | 0.245 |
| CIFAR10 | ResNet50 | 0 | 100.0% | 95.2% | 0.899 |
| | | 10 | 72.5% | 66.4% | 0.593 |
| | | 20 | 77.9% | 75.0% | 0.176 |
| | | 30 | 88.8% | 85.3% | 0.131 |
| | | 40 | 92.9% | 88.7% | 0.231 |
| | | 50 | 96.1% | 91.1% | 0.394 |
| | | 60 | 98.0% | 92.5% | 0.459 |
| | | 70 | 98.9% | 93.2% | 0.579 |
| | | 80 | 99.4% | 93.7% | 0.636 |
| | | 90 | 99.6% | 94.0% | 0.665 |
| | | 100 | 99.7% | 94.0% | 0.665 |
| CIFAR100 | ResNet8 | 0 | 96.4% | 67.8% | 0.515 |
| | | 10 | 84.9% | 59.2% | 0.112 |
| | | 20 | 89.5% | 63.4% | 0.063 |
| | | 30 | 93.0% | 67.3% | 0.048 |
| | | 40 | 94.6% | 68.5% | 0.046 |
| | | 50 | 95.2% | 69.0% | 0.038 |
| | | 60 | 95.6% | 69.4% | 0.036 |
| | | 70 | 95.9% | 69.5% | 0.031 |
| | | 80 | 96.0% | 69.8% | 0.030 |
| | | 90 | 96.1% | 70.0% | 0.026 |
| | | 100 | 96.2% | 70.2% | 0.026 |
| CIFAR100 | ResNet50 | 0 | 100.0% | 78.3% | 0.906 |
| | | 10 | 64.9% | 50.8% | 0.170 |
| | | 20 | 83.6% | 65.6% | 0.348 |
| | | 30 | 93.1% | 71.7% | 0.419 |
| | | 40 | 96.9% | 74.4% | 0.467 |
| | | 50 | 98.7% | 75.5% | 0.505 |
| | | 60 | 99.4% | 76.4% | 0.519 |
| | | 70 | 99.7% | 76.7% | 0.530 |
| | | 80 | 99.8% | 76.5% | 0.546 |
| | | 90 | 99.8% | 76.8% | 0.555 |
| | | 100 | 99.9% | 77.3% | 0.561 |

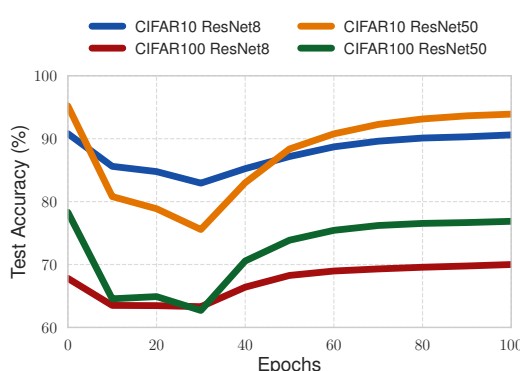

Figure L.7: Test Accuracy vs. Epochs, $\theta = 0.75$, MNIST. This has similar behavior to Fig. L.6a.

since their parameter space is very large. Second, every gradient step of our Alg. 1 is given by $\theta\nabla_{\mathbf{w}}\mathcal{L}_{\mathcal{K}}(\mathbf{w}, \mathcal{D}_f) + (1 - \theta)\nabla_{\mathbf{w}}\mathcal{L}_{\mathcal{A}}(\mathbf{w}, \mathcal{D}_r)$.

In what follows, consider a large model e.g. CIFAR10 ResNet50.

When we begin finetuning the pretrained model with Alg. 1, $\theta$ is large, $\mathcal{L}_{\mathcal{A}}$ is small, and $\mathcal{L}_{\mathcal{K}}$ is large. Thus, $\nabla_{\mathbf{w}}\mathcal{L}_{\mathcal{K}}$ significantly dominates $\nabla_{\mathbf{w}}\mathcal{L}_{\mathcal{A}}$. For large models, since the gradients are nearly orthogonal, we will move very fast in the direction of the forget gradient.

As finetuning continues, we reach a point where $\theta$ is large, $\mathcal{L}_{\mathcal{K}}$ is small, and $\mathcal{L}_{\mathcal{A}}$ is large. At this point, the $\nabla_{\mathbf{w}}\mathcal{L}_{\mathcal{A}}$ begins to dominate, albeit less significantly since $\theta$ is large. For large models, since the gradients are nearly orthogonal, we will move fast in the direction of the retain gradient. However, due to our choice of large $\theta$, the retain gradient will be reduced in magnitude. Thus, we will move more slowly at this stage.

We must stop shortly after this, otherwise the forget loss will climb back up to a point where the model is no longer reasonably uniform.

Importantly, since smaller models (e.g. CIFAR10 ResNet8) have smaller parameter spaces, the gradients do not conflict as much. Thus, when we begin finetuning the model, while we do increase in retain loss initially, after the uniform loss is minimized we can minimize the retain loss freely. As such, we can move in a direction that minimizes both the forget and retain gradients, and do not need to stop early.

However, as noted in the main paper, the model needs to be sufficiently large to achieve strong test accuracy, e.g. logistic regression trained on KMNIST does not work well. We hypothesize that this is because a model with more parameters has many more subspaces where two task losses can coexist in a way that provides a good tradeoff. We leave studying this and the above more formally to future work.

### L.7 EVALUATING TEST-TIME PRIVACY ATTACKS

In what follows, we assume a test-time privacy (TTP) adversary with open-weight model access. We describe our attacks, specifically in Appx. A.

The results for the attacks with Alg. 5, Alg. 6, and Alg. 7 are in Tab. L.8 and Tab. L.9. We see that our method effectively defends against Alg. 5, Alg. 6, and Alg. 7 for various choices of $\gamma$ in most scenarios. In our choices of $\gamma$, we follow the adversarial robustness literature, choosing $\gamma$ sufficiently small such that the perturbation is invisible to the naked eye, but sufficiently large such that our attack is effective. However, in some cases, the attacks succeed despite the use of our method. Still, as demonstrated in Tab. L.10, we find that our algorithm renders the forget accuracy very low in these cases. As such, the adversary cannot be confidently correct–rather, in most cases, they can at best be confidently wrong. In particular, we have significantly better protection from attacks than in the pretrain, retrain, synthetic, or LabelDP cases. We provide visual intuition for our attacks in Fig. L.8.

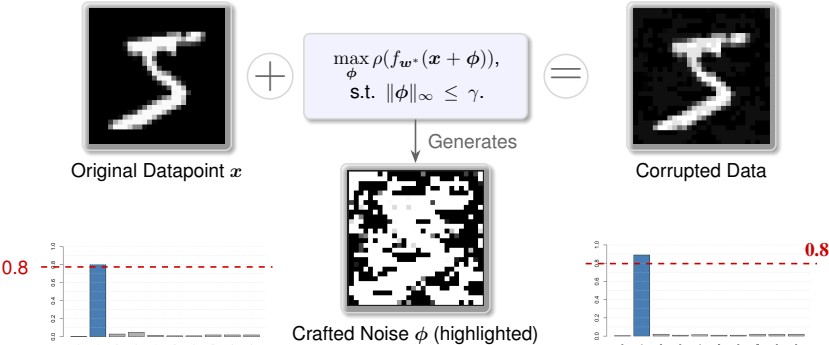

Figure L.8: We corrupt an instance to increase the confidence of the final prediction. Noise is highlighted throughout for clarity.

---

**Algorithm 5** Gaussian Noise Open-Weight Test Time Privacy Attack

**Require:** Forget set $\mathcal{D}_f$; pretrained model $\boldsymbol{w}^* = \mathcal{A}(\mathcal{D})$; adversarial $\gamma$

$\quad \mathcal{D}_f^{adv} = []$
$\quad$ **for** $i = 1, ..., |\mathcal{D}_f|$ **do**
$\quad\quad \boldsymbol{x}_{adv} = \mathcal{D}_f^{(i)} + \beta, \beta \sim \mathcal{N}(0, \gamma \boldsymbol{I})$
$\quad\quad \mathcal{D}_f^{adv,(i)} = \boldsymbol{x}_{\text{adv}}$
$\quad$ **end for**
$\quad\quad$ **return** $\mathcal{D}_f^{\text{adv}}$

---

## L.8  ROBUSTNESS OF ALG. 1 CLASSIFIER ON NEIGHBORING TEST INSTANCES

To illustrate what happens for the finetuned classifier on neighboring test instances, we run an experiment evaluating accuracy and average confidence distance on test instances which are nearest neighbors to forget instances. Specifically, for each forget instance $\mathbf{x} \in \mathcal{D}_f$, we found the nearest neighbor of $\mathbf{x}$ in the test set. We evaluated this nearest neighbor in pixel space with respect to the $\ell_2$ distance.

Results are in Tab. L.11. We notice that the classifier works as intended. That is, we obtain high accuracy as well as high confidence distance on the test set, including for nearby instances. While we do observe a drop for CIFAR100 ResNet50, we also observe that the confidence distance is still much higher than the 0.298 confidence distance on the forget set.

## L.9  ENSURING TEST-TIME PRIVACY FOR TEST INSTANCES

In our paper, we focus on training data examples because this is the basis for scenarios addressed by the GDPR, HIPAA, etc. Providing the same guarantee for non-training (test) data is an equally

---

**Algorithm 6** FGSM-Style Open-Weight Test Time Privacy Attack

**Require:** Forget set $\mathcal{D}_f$; pretrained model $\boldsymbol{w}^* = \mathcal{A}(\mathcal{D})$; adversarial $\gamma$; symmetry breaking $\alpha$

$\quad \mathcal{D}_f^{adv} = []$
$\quad$ **for** $i = 1, ..., |\mathcal{D}_f|$ **do**
$\quad\quad \boldsymbol{x}_0 = \mathcal{D}_f^{(i)} + \beta, \beta \sim U([-\alpha\gamma, \alpha\gamma])$
$\quad\quad \boldsymbol{x}_{\text{adv}} = \mathcal{D}_f^{(i)} + \gamma \text{sign}(\nabla_{\boldsymbol{x}} \rho(f_{\boldsymbol{w}^*}(\boldsymbol{x}))\big|_{\boldsymbol{x}=\boldsymbol{x}_0})$
$\quad\quad \mathcal{D}_f^{adv,(i)} = \boldsymbol{x}_{\text{adv}}$
$\quad$ **end for**
$\quad\quad$ **return** $\mathcal{D}_f^{\text{adv}}$

---

---

**Algorithm 7** PGD-Style Open-Weight Test Time Privacy Attack

---

**Require:** Forget set $\mathcal{D}_f$; pretrained model $\boldsymbol{w}^* = \mathcal{A}(\mathcal{D})$; adversarial $\gamma$; symmetry breaking $\alpha$; step count $N$

$\quad \mathcal{D}_f^{adv} = []$

$\quad$ **for** $i = 1, ..., |\mathcal{D}_f|$ **do**

$\quad\quad \boldsymbol{x}_{adv}^0 = \mathcal{D}_f^{(i)} + \beta, \beta \sim U([-\alpha\gamma, \alpha\gamma])$

$\quad\quad$ **for** $j = 1, ..., N$ **do**

$\quad\quad\quad \boldsymbol{x}_{adv}^j = \boldsymbol{x}_{adv}^{j-1} + \beta\mathrm{sign}(\nabla_{\boldsymbol{x}}\rho(f_{\boldsymbol{w}^*}(\boldsymbol{x}))\big|_{\boldsymbol{x}=\boldsymbol{x}_0})$

$\quad\quad\quad \boldsymbol{x}_{adv}^j = \prod_{\mathcal{B}_\gamma(\mathcal{D}_f^{(i)})}(\boldsymbol{x}_{adv})$

$\quad\quad$ **end for**

$\quad\quad \mathcal{D}_f^{adv,(i)} = \boldsymbol{x}_{adv}^N$

$\quad$ **end for**

$\quad\quad$ **return** $\mathcal{D}_f^{adv}$

---

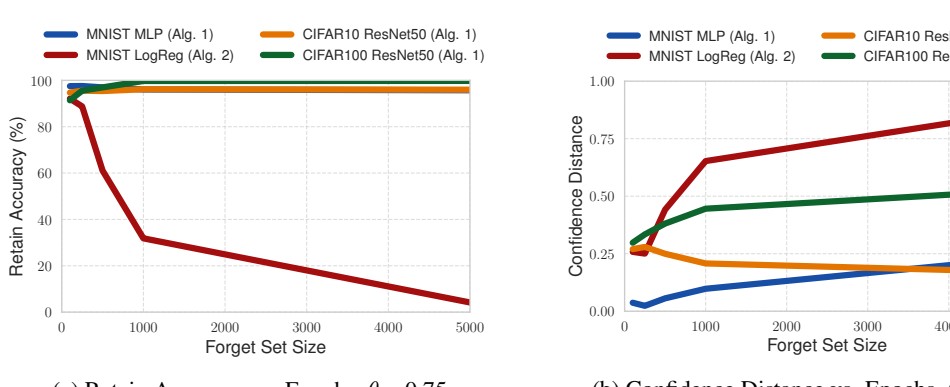

(a) Retain Accuracy vs. Epochs, $\theta = 0.75$      (b) Confidence Distance vs. Epochs, $\theta = 0.75$

Figure L.9: We observe that retain accuracy stays fairly stable as the forget size increases, in Fig. L.9a, except for Alg. 2 where it causes catastrophic failure due to the magnitude of the Newton step. Furthermore, we find that confidence distance slowly increases as the forget set size increases in Fig. L.9b.

important problem. However, the proposed method can be extended to cover this new case without loss of generality. One can just finetune on test instances highlighted to be corrupted and then run Alg. 1. In Tab. L.12, we find that finetuning with test instances yields similar performance to using our algorithm over just the training instances.

### L.10 ABLATION STUDY ON FORGET SET SIZE

Figures are provided in Fig. L.9 and Fig. L.10. In Tab. L.13, we provide experiments for ResNet50 trained on CIFAR10 and CIFAR100 for Alg. 1. In Tab. L.14, we provide experiments on MLP trained over MNIST for Alg. 1 and logistic regression trained over MNIST for Alg. 2.

Throughout our experiments, we use a forget set size of 100. We do so because for our use case, it is likely that a data controller would want to induce uniformity only for a small number of instances. We observe that as one increases the forget set size, it becomes harder to induce uniformity with the same hyperparameters. Still, for Alg. 1, we are able to obtain strong uniformity with good retain and test accuracy for significantly larger forget set sizes. Furthermore, we observe that Alg. 2 fails for sufficiently large forget set size; this is likely because Hessian matrix is significantly larger (in norm) for a larger forget set, resulting in a catastrophically large Newton step. Mitigating this phenomenon is left to future work, where Hessian-free techniques like those of Qiao et al. (2025) may be advantageous.

Table L.8: Confidence distances over the forget set after Alg. 5, Alg. 6, and Alg. 7 are applied to pretrained models and models finetuned with Alg. 1 for MNIST and KMNIST. Lower is better.

| Dataset | Model | $\gamma$ | Method | Attack | Prior Conf. Dist. (Lower Better) | Attack Conf. Dist. (Lower Better) |
|---|---|---|---|---|---|---|
| MNIST | MLP | $\frac{2}{255}$ | | Pretrain | Alg. 6 | 0.879 | 0.884 |
| | | | | Alg. 5 | 0.037 | 0.043 |
| | | | Alg. 1 | Alg. 6 | 0.037 | 0.054 |
| | | | | Alg. 7 | 0.037 | 0.185 |
| | | $\frac{5}{255}$ | | Pretrain | Alg. 6 | 0.879 | 0.888 |
| | | | | Alg. 5 | 0.037 | 0.064 |
| | | | Alg. 1 | Alg. 6 | 0.037 | 0.089 |
| | | | | Alg. 7 | 0.037 | 0.488 |
| | | $\frac{8}{255}$ | | Pretrain | Alg. 6 | 0.879 | 0.891 |
| | | | | Alg. 5 | 0.037 | 0.091 |
| | | | Alg. 1 | Alg. 6 | 0.037 | 0.125 |
| | | | | Alg. 7 | 0.037 | 0.632 |
| | ResNet18 | $\frac{2}{255}$ | | Pretrain | Alg. 6 | 0.895 | 0.896 |
| | | | | Alg. 5 | 0.070 | 0.075 |
| | | | Alg. 1 | Alg. 6 | 0.070 | 0.133 |
| | | | | Alg. 7 | 0.070 | 0.133 |
| | | $\frac{5}{255}$ | | Pretrain | Alg. 6 | 0.895 | 0.897 |
| | | | | Alg. 5 | 0.070 | 0.088 |
| | | | Alg. 1 | Alg. 6 | 0.070 | 0.164 |
| | | | | Alg. 7 | 0.070 | 0.350 |
| | | $\frac{8}{255}$ | | Pretrain | Alg. 6 | 0.895 | 0.898 |
| | | | | Alg. 5 | 0.070 | 0.116 |
| | | | Alg. 1 | Alg. 6 | 0.070 | 0.248 |
| | | | | Alg. 7 | 0.070 | 0.638 |
| KMNIST | ResNet18 | $\frac{2}{255}$ | | Pretrain | Alg. 6 | 0.858 | 0.865 |
| | | | | Alg. 5 | 0.257 | 0.258 |
| | | | Alg. 1 | Alg. 6 | 0.257 | 0.302 |
| | | | | Alg. 7 | 0.257 | 0.317 |
| | | $\frac{5}{255}$ | | Pretrain | Alg. 6 | 0.858 | 0.872 |
| | | | | Alg. 5 | 0.257 | 0.259 |
| | | | Alg. 1 | Alg. 6 | 0.257 | 0.370 |
| | | | | Alg. 7 | 0.257 | 0.420 |
| | | $\frac{8}{255}$ | | Pretrain | Alg. 6 | 0.858 | 0.878 |
| | | | | Alg. 5 | 0.257 | 0.259 |
| | | | Alg. 1 | Alg. 6 | 0.257 | 0.433 |
| | | | | Alg. 7 | 0.257 | 0.520 |

The results in figures Fig. L.9 and Fig. L.10, as well as Tab. L.14 and Tab. L.13, we keep the batch size used for the forget set in Alg. 1 to 10 instances. However, upon scaling the batch size proportionately with the forget set size (specifically dividing the forget set size by ten), we observe much better performance. These results are in Tab. L.15. Specifically, for a MLP trained on MNIST and a forget set size of 5000, we obtain much better uniformity than when we did not scale the batch size.

## L.11 Evaluating Confidence Distance as a TTP Metric

The $\ell_2$ metric $||f(\mathbf{x}) - \frac{\vec{1}}{K}||_2$ has similar utility to our presented metric. However, it is slightly less interpretable and may accidentally overpenalize uncertain outputs. For example, if one class has no probability but the other 9 classes are uniform. Still, as demonstrated in Tab. L.16, we find that we minimize this metric as well for the same models and datasets. This holds similarly for other potential metrics e.g. the $\ell_1$ metric.

Table L.9: Confidence distances over the forget set after Alg. 5, Alg. 6, and Alg. 7 are applied to pretrained models and models finetuned with Alg. 1 for SVHN, CIFAR10, and CIFAR100. Lower is better.

| Dataset | Model | $\gamma$ | Method | Attack | Prior Conf. Dist. (Lower Better) | Attack Conf. Dist. (Lower Better) |
|---|---|---|---|---|---|---|
| SVHN | ResNet50 | $\frac{1}{255}$ | | Pretrain | Alg. 6 | 0.972 | 0.886 |
| | | | | Alg. 5 | 0.289 | 0.519 |
| | | | Alg. 1 | Alg. 6 | 0.289 | 0.571 |
| | | | | Alg. 7 | 0.289 | 0.582 |
| | | $\frac{2}{255}$ | | Pretrain | Alg. 6 | 0.972 | 0.904 |
| | | | | Alg. 5 | 0.289 | 0.519 |
| | | | Alg. 1 | Alg. 6 | 0.289 | 0.613 |
| | | | | Alg. 7 | 0.289 | 0.640 |
| CIFAR10 | ResNet18 | $\frac{1}{255}$ | | Pretrain | Alg. 6 | 0.898 | 0.898 |
| | | | | Alg. 5 | 0.377 | 0.300 |
| | | | Alg. 1 | Alg. 6 | 0.377 | 0.353 |
| | | | | Alg. 7 | 0.377 | 0.356 |
| | | $\frac{2}{255}$ | | Pretrain | Alg. 6 | 0.898 | 0.822 |
| | | | | Alg. 5 | 0.377 | 0.301 |
| | | | Alg. 1 | Alg. 6 | 0.377 | 0.397 |
| | | | | Alg. 7 | 0.377 | 0.408 |
| | ResNet50 | $\frac{1}{255}$ | | Pretrain | Alg. 6 | 0.900 | 0.806 |
| | | | | Alg. 5 | 0.270 | 0.336 |
| | | | Alg. 1 | Alg. 6 | 0.270 | 0.374 |
| | | | | Alg. 7 | 0.270 | 0.378 |
| | | $\frac{2}{255}$ | | Pretrain | Alg. 6 | 0.900 | 0.827 |
| | | | | Alg. 5 | 0.270 | 0.336 |
| | | | Alg. 1 | Alg. 6 | 0.270 | 0.407 |
| | | | | Alg. 7 | 0.270 | 0.425 |
| CIFAR100 | ResNet50 | $\frac{1}{255}$ | | Pretrain | Alg. 6 | 0.906 | 0.587 |
| | | | | Alg. 5 | 0.298 | 0.275 |
| | | | Alg. 1 | Alg. 6 | 0.298 | 0.360 |
| | | | | Alg. 7 | 0.298 | 0.373 |
| | | $\frac{2}{255}$ | | Pretrain | Alg. 6 | 0.906 | 0.652 |
| | | | | Alg. 5 | 0.298 | 0.276 |
| | | | Alg. 1 | Alg. 6 | 0.298 | 0.421 |
| | | | | Alg. 7 | 0.298 | 0.468 |

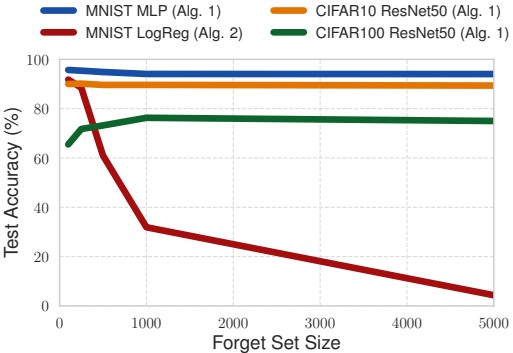

Figure L.10: Test Accuracy vs. Forget Set Size, $\theta = 0.75$. This has similar behavior to Fig. L.9a.

Table L.10: Accuracies over the forget set after Alg. 6 and Alg. 7 are applied to pretrained models and models finetuned with Alg. 1 for SVHN, CIFAR10, and CIFAR100. We see that Alg. 1 significantly lowers forget set accuracy.

| Dataset | Model | $\gamma$ | Method | Attack | Prior. Forget Acc. (Lower Better) | Atk. Forget Acc. (Lower Better) |
|---|---|---|---|---|---|---|
| MNIST | MLP | $\frac{5}{255}$ | Pretrain | Alg. 7 | 97.0% | 96.0% |
| | | | Alg. 1 | Alg. 7 | 71.0% | 43.0% |
| | | $\frac{8}{255}$ | Pretrain | Alg. 7 | 98.3% | 96.0% |
| | | | Alg. 1 | Alg. 7 | 71.0% | 41.0% |
| | ResNet18 | $\frac{8}{255}$ | Pretrain | Alg. 7 | 98.9% | 100.0% |
| | | | Alg. 1 | Alg. 7 | 28.0% | 54.0% |
| KMNIST | ResNet18 | $\frac{8}{255}$ | Pretrain | Alg. 7 | 96.0% | 96.0% |
| | | | Alg. 1 | Alg. 7 | 50.0% | 55.0% |
| SVHN | ResNet50 | $\frac{1}{255}$ | Pretrain | Alg. 6 | 97.0% | 66.0% |
| | | | | Alg. 5 | 40.0% | 53.0% |
| | | | Alg. 1 | Alg. 6 | 40.0% | 52.0% |
| | | | | Alg. 7 | 40.0% | 52.0% |
| | | $\frac{2}{255}$ | Pretrain | Alg. 6 | 97.0% | 66.0% |
| | | | | Alg. 5 | 40.0% | 52.0% |
| | | | Alg. 1 | Alg. 6 | 40.0% | 53.0% |
| | | | | Alg. 7 | 40.0% | 53.0% |
| CIFAR10 | ResNet18 | $\frac{1}{255}$ | Pretrain | Alg. 6 | 100.0% | 61.0% |
| | | | | Alg. 5 | 60.0% | 35.0% |
| | | | Alg. 1 | Alg. 6 | 60.0% | 34.0% |
| | | | | Alg. 7 | 60.0% | 34.0% |
| | | $\frac{2}{255}$ | Pretrain | Alg. 6 | 100.0% | 61.0% |
| | | | | Alg. 5 | 60.0% | 35.0% |
| | | | Alg. 1 | Alg. 6 | 60.0% | 33.0% |
| | | | | Alg. 7 | 60.0% | 32.0% |
| | ResNet50 | $\frac{1}{255}$ | Pretrain | Alg. 6 | 100.0% | 56.0% |
| | | | | Alg. 5 | 55.0% | 33.0% |
| | | | Alg. 1 | Alg. 6 | 55.0% | 30.0% |
| | | | | Alg. 7 | 55.0% | 30.0% |
| | | $\frac{2}{255}$ | Pretrain | Alg. 6 | 100.0% | 56.0% |
| | | | | Alg. 5 | 55.0% | 30.0% |
| | | | Alg. 1 | Alg. 6 | 55.0% | 30.0% |
| | | | | Alg. 7 | 55.0% | 32.0% |
| CIFAR100 | ResNet50 | $\frac{1}{255}$ | Pretrain | Alg. 6 | 100.0% | 32.0% |
| | | | | Alg. 5 | 48.0% | 11.0% |
| | | | Alg. 1 | Alg. 6 | 48.0% | 10.0% |
| | | | | Alg. 7 | 48.0% | 11.0% |
| | | $\frac{2}{255}$ | Pretrain | Alg. 6 | 100.0% | 32.0% |
| | | | | Alg. 5 | 48.0% | 11.0% |
| | | | Alg. 1 | Alg. 6 | 48.0% | 10.0% |
| | | | | Alg. 7 | 48.0% | 11.0% |

## L.12 An Additional Baseline with Randomly Sampled Labels: GaussianUniform

In what follows, we present the *GaussianUniform* baseline, an alternative idea to our approach based on the notable work of Zhang et al. (2017), which demonstrates that a neural network can fully minimize its loss over a training dataset where samples have labels sampled uniformly at random. The approach of GaussianUniform is as follows:

1. Begin with a training dataset $\mathcal{D} = \mathcal{D}_f \cup \mathcal{D}_r$.

2. Perturb all samples in $\mathcal{D}$ to yield $\mathcal{D}' = \mathcal{D}'_f \cup \mathcal{D}'_r$. We use mean zero Gaussian noise with 0.1 variance, which adds a small amount of noise.

Table L.11: Accuracies and confidence distances for test instances which are nearest neighbors (with respect to $\ell_2$ distance) of forget set instances. We observe that models continue to confidently and correctly classify nearby test instances after finetuning with Alg. 1.

| Dataset | Model | Method | Acc. | Conf. Dist. (Higher Better) |
|---------|-------|--------|------|------------------------------|
| MNIST | MLP | Pretrain | 100.0% | 0.894 |
| | | Alg. 1 | 93.0% | 0.754 |
| | ResNet18 | Pretrain | 100.0% | 0.896 |
| | | Alg. 1 | 100.0% | 0.875 |
| KMNIST | ResNet18 | Pretrain | 99.0% | 0.871 |
| | | Alg. 1 | 99.0% | 0.850 |
| SVHN | ResNet50 | Pretrain | 95.0% | 0.982 |
| | | Alg. 1 | 95.0% | 0.939 |
| CIFAR10 | ResNet18 | Pretrain | 98.0% | 0.876 |
| | | Alg. 1 | 85.0% | 0.538 |
| | ResNet50 | Pretrain | 96.0% | 0.884 |
| | | Alg. 1 | 90.0% | 0.700 |
| CIFAR100 | ResNet50 | Pretrain | 78.0% | 0.778 |
| | | Alg. 1 | 66.0% | 0.484 |

3. Sample all labels in $\mathcal{D}$ uniformly at random to yield $\tilde{\mathcal{D}} = \tilde{\mathcal{D}}_f \cup \tilde{\mathcal{D}}_r$.

4. Train $\mathcal{A}(\mathcal{D}' \cup \tilde{\mathcal{D}})$.

In this scenario, inducing uncertainty may not be necessary, since the forget set would have very strong uniformity, with strong accuracy on the retain set available by slightly perturbing with Gaussian noise. We use the same hyperparameters as pretraining for the respective model and dataset tested, as reported in Appx. K. We find that this is not the case for ResNet50 trained on SVHN, CIFAR10, and CIFAR100. However, it achieves very poor test accuracy compared to Alg. 1 on both normal $\mathcal{D}_{\text{test}}$ and perturbed test $\mathcal{D}'_{\text{test}}$ datasets; thus, we prefer Alg. 1 to this approach, since it generalizes well and also does not require retraining. Results are in Tab. L.17.

### L.13 TIGHTNESS OF BOUND IN THEOREM 5.5

To evaluate how tight our bound is, we run an experiment for MNIST logistic regression. We use the notation of theorem 5.5 in Tab. L.18. We find that our constant bound is fairly tight as $\theta \to 1$; we leave using more advanced techniques to ensure better tightness to future work.

### L.14 CONFIDENCE INTERVALS FOR MAIN PAPER EXPERIMENTS

In what follows, we report confidence intervals for only ResNet50 trained on SVHN due to compute constraints. We find that variance is low in Tab. L.19.

### L.15 PROPORTIONS OF TIME ELAPSED IN ALG. 1

Results are reported in Tab. L.20.

### L.16 WARMUP VALUES FOR MNIST LOGREG

Results are contained in Tab. L.21. We find that after applying the certified Newton step in Alg. 2, we obtain better retain and test accuracy, at small cost to uniformity. Thus, warming up is not the only component of achieving good results in Alg. 2.

Table L.12: Finetuning on test instances and running Alg. 1. Here, "pretrain" denotes the initially pretrained model (no additional test instances), while other rows correspond to finetuning on the specified number of test instances.

| Dataset | Model | % Forget Size | Method | Retain Acc. | Test Acc. | Conf. Dist. (Lower Better) |
|---|---|---|---|---|---|---|
| MNIST | MLP | 0 | Pretrain | 98.3% | 97.0% | 0.879 |
| | | | Alg. 1 | 97.5% | 95.7% | 0.037 |
| | | 1/5 | Pretrain | 99.1% | 97.3% | 0.892 |
| | | | Alg. 1 | 96.4% | 93.9% | 0.147 |
| | | 1/2 | Pretrain | 99.0% | 97.1% | 0.896 |
| | | | Alg. 1 | 96.2% | 93.4% | 0.157 |
| | ResNet18 | 0 | Pretrain | 99.2% | 98.9% | 0.879 |
| | | | Alg. 1 | 99.6% | 99.1% | 0.070 |
| | | 1/5 | Pretrain | 98.9% | 98.5% | 0.884 |
| | | | Alg. 1 | 99.6% | 98.9% | 0.181 |
| | | 1/2 | Pretrain | 99.4% | 100.0% | 0.897 |
| | | | Alg. 1 | 99.5% | 98.7% | 0.243 |
| KMNIST | ResNet18 | 0 | Pretrain | 98.2% | 92.1% | 0.880 |
| | | | Alg. 1 | 99.1% | 94.7% | 0.257 |
| | | 1/5 | Pretrain | 100.0% | 97.5% | 0.900 |
| | | | Alg. 1 | 99.7% | 96.5% | 0.301 |
| | | 1/2 | Pretrain | 100.0% | 97.7% | 0.895 |
| | | | Alg. 1 | 99.5% | 96.9% | 0.233 |
| SVHN | ResNet50 | 0 | Pretrain | 99.5% | 94.4% | 0.971 |
| | | | Alg. 1 | 99.0% | 94.4% | 0.280 |
| | | 1/5 | Pretrain | 99.7% | 94.6% | 0.974 |
| | | | Alg. 1 | 99.2% | 94.4% | 0.184 |
| | | 1/2 | Pretrain | 99.5% | 94.5% | 0.986 |
| | | | Alg. 1 | 99.8% | 95.1% | 0.391 |
| CIFAR10 | ResNet18 | 0 | Pretrain | 100.0% | 95.3% | 0.898 |
| | | | Alg. 1 | 89.6% | 83.1% | 0.377 |
| | | 1/5 | Pretrain | 100.0% | 98.4% | 0.894 |
| | | | Alg. 1 | 89.4% | 85.6% | 0.358 |
| | | 1/2 | Pretrain | 100.0% | 93.5% | 0.887 |
| | | | Alg. 1 | 87.9% | 83.7% | 0.494 |
| | ResNet50 | 0 | Pretrain | 100.0% | 95.2% | 0.900 |
| | | | Alg. 1 | 94.7% | 90.0% | 0.270 |
| | | 1/5 | Pretrain | 99.9% | 93.4% | 0.896 |
| | | | Alg. 1 | 91.6% | 87.1% | 0.392 |
| | | 1/2 | Pretrain | 99.8% | 93.6% | 0.894 |
| | | | Alg. 1 | 89.1% | 86.7% | 0.390 |
| CIFAR100 | ResNet50 | 0 | Pretrain | 100.0% | 78.3% | 0.902 |
| | | | Alg. 1 | 91.4% | 65.5% | 0.298 |
| | | 1/5 | Pretrain | 100.0% | 79.0% | 0.880 |
| | | | Alg. 1 | 93.5% | 69.1% | 0.021 |
| | | 1/2 | Pretrain | 99.9% | 78.2% | 0.860 |
| | | | Alg. 1 | 97.9% | 72.8% | 0.221 |

## L.17 VISUALIZATION OF SOFTMAX OUTPUTS

We provide a comparison of pretrained $f$ and Alg. 1 softmax probabilities across five different CIFAR10 forget set samples, demonstrating visually the effectiveness of Alg. 1 at inducing uniformity (and the relevance of our confidence distance metric) in Fig. L.11.

Table L.13: Results on applying Alg. 1 on ResNet50 for various forget set sizes over CIFAR10 and CIFAR100. Please see Appx. L.10 for a discussion on Alg. 2. $\theta = 0.75$ throughout.

| Dataset | Model | Forget Size | Retain Acc. | Test Acc. | Conf. Dist. (Lower Better) |
|---|---|---|---|---|---|
| CIFAR10 | ResNet50 | 100 | 94.7% | 90.0% | 0.270 |
| | | 250 | 96.4% | 90.3% | 0.289 |
| | | 500 | 95.1% | 88.6% | 0.190 |
| | | 1000 | 97.0% | 90.0% | 0.144 |
| | | 5000 | 95.8% | 89.5% | 0.176 |
| CIFAR100 | ResNet50 | 100 | 91.4% | 65.5% | 0.298 |
| | | 250 | 99.7% | 78.0% | 0.370 |
| | | 500 | 99.8% | 76.1% | 0.475 |
| | | 1000 | 99.7% | 74.8% | 0.492 |
| | | 5000 | 99.8% | 74.1% | 0.613 |

Table L.14: Results on applying Alg. 1 and Alg. 2 on various forget set sizes over MNIST. We observe that, while Alg. 1 still works well, confidence distance increases as forget set size does; please see Appx. L.10 for a discussion on Alg. 2. $\theta = 0.75$ throughout.

| Method | Model | Forget Size | Retain Acc. | Test Acc. | Conf. Dist. (Lower Better) |
|---|---|---|---|---|---|
| Alg. 1 | MLP | 100 | 97.5% | 95.7% | 0.037 |
| | | 250 | 97.6% | 95.1% | 0.010 |
| | | 500 | 95.8% | 93.8% | 0.121 |
| | | 1000 | 94.8% | 93.3% | 0.162 |
| | | 5000 | 96.6% | 95.0% | 0.420 |
| Alg. 2 | LogReg | 100 | 92.1% | 91.8% | 0.258 |
| | | 250 | 85.4% | 85.2% | 0.243 |
| | | 500 | 5.7% | 5.9% | 0.824 |
| | | 1000 | 4.4% | 4.6% | 0.891 |
| | | 5000 | 2.4% | 2.5% | 0.899 |

(a) Sample 1    (b) Sample 2    (c) Sample 3    (d) Sample 4    (e) Sample 5

Figure L.11: Comparison of Pretrain (red) and Alg. 1 (blue) softmax probabilities across five different CIFAR10 forget set samples.

Table L.15: Results on applying Alg. 1 on various forget set sizes over MNIST while scaling the batches. We observe much better performance for a forget set size of 5000 than in Tab. L.14, and comparable performance on other forget set sizes.

| Method | Model | Forget Size | Retain Acc. | Test Acc. | Conf. Dist. (Lower Better) |
|---|---|---|---|---|---|
| Alg. 1 | MLP | 100 | 97.5% | 95.7% | 0.037 |
| | | 250 | 95.6% | 96.6% | 0.131 |
| | | 500 | 93.8% | 93.4% | 0.187 |
| | | 1000 | 92.3% | 92.2% | 0.192 |
| | | 5000 | 95.3% | 94.9% | 0.227 |

Table L.16: $\ell_2$ confidence distances for models finetuned with Alg. 1.

| Dataset | Model | Conf. Dist. Type | Pretrained (Lower Better) | Alg. 1 (Lower Better) |
|---|---|---|---|---|
| MNIST | MLP | Paper | 0.879 | 0.037 |
| | | $\ell_2$ | 0.930 | 0.053 |
| | ResNet18 | Paper | 0.895 | 0.070 |
| | | $\ell_2$ | 0.944 | 0.070 |
| KMNIST | ResNet18 | Paper | 0.880 | 0.257 |
| | | $\ell_2$ | 0.911 | 0.302 |
| SVHN | ResNet50 | Paper | 0.972 | 0.289 |
| | | $\ell_2$ | 0.979 | 0.298 |
| CIFAR10 | ResNet18 | Paper | 0.898 | 0.377 |
| | | $\ell_2$ | 0.947 | 0.435 |
| | ResNet50 | Paper | 0.900 | 0.270 |
| | | $\ell_2$ | 0.948 | 0.323 |
| CIFAR100 | ResNet50 | Paper | 0.902 | 0.298 |
| | | $\ell_2$ | 0.911 | 0.311 |

Table L.17: Results for the *GaussianUniform* baseline described in Appx. L.12. This method results in significantly degraded accuracy on the test set compared to a model finetuned with Alg. 1, despite achieving high accuracy on the perturbed retain set ($D_r'$). Note that performance on $\tilde{\mathcal{D}}_r$ is similar to $\mathcal{D}_r'$.

| Dataset | Model | Train Set Acc. (%) | | Test Set Acc. (%) | | | Conf. Dist. |
|---|---|---|---|---|---|---|---|
| | | $\mathcal{D}_r'$ | $\mathcal{D}_f'$ | $\mathcal{D}_{\text{test}}$ | $\mathcal{D}_{\text{test}}'$ | Alg. 1 | $\mathcal{D}_f$ |
| SVHN | ResNet50 | 82.4% | 2.0% | 6.2% | 81.1% | 94.4% | 0.009 |
| CIFAR10 | ResNet50 | 100.0% | 12.0% | 11.1% | 73.3% | 90.0% | 0.573 |
| CIFAR100 | ResNet50 | 99.9% | 4.0% | 3.4% | 40.4% | 65.5% | 0.057 |

## L.18 RESULTS FOR KMNIST LOGREG AND MLP

Results are contained in Tab. L.22. As mentioned in Sec. 6, one can see that a small model for a more complex benchmark, logistic regression on KMNIST, fails to induce uniformity, since the pretrained model is too small to generalize. However, on a bigger model i.e. an MLP trained over KMNIST, since it is large enough to generalize, one can induce uniformity over it. Thus, larger models which generalize well are preferred for our method, in line with the goals of ML.

## L.19 ABLATION STUDY ON SYNTHETIC BASELINE SAMPLE SIZE

Below, we study what we happen if we increase the number of samples sampled in the $\varepsilon$-ball in the synthetic baseline. For each forget set instance, we sample $k$ instances from the $\varepsilon$-ball around the forget set, and then assign random labels to these instances, yielding an additional $|\mathcal{D}_f|k$ instances in the training data. We then retrain the model over the retain set along with these new $|\mathcal{D}_f|k$ instances. For a MLP trained over MNIST, we observe better performance as we increase sample size. However, for a ResNet18 trained over CIFAR10, even if we have a very large sample size. This is presented in Tab. L.23. Thus, since Alg. 1 can induce uniformity as shown in Tab. L.1, without great cost to retain or test accuracy, it is better than the synthetic baseline.

Table L.18: Comparison of bounds

| $\alpha^* - \alpha(1)$ | O(1) bound size |
|---|---|
| 1.678 | 3.374 |

Table L.19: Results for Alg. 1 for ResNet50 trained on SVHN over three runs. We find that we are able to induce uniformity while only slightly decreasing retain and test accuracy, $\theta = 0.75$ throughout, with minimal variance for all metrics.

| Dataset | Model | Method | Retain Acc. | Test Acc. | Conf. Dist. (Lower Better) |
|---|---|---|---|---|---|
| SVHN | ResNet50 | Pretrain | $99.9\% \pm 0.1\%$ | $95.3\% \pm 0.5\%$ | $0.987 \pm 0.001$ |
| | | Alg. 1 | $99.5\% \pm 0.2\%$ | $94.8\% \pm 0.3\%$ | $0.276 \pm 0.004$ |

Table L.20: Time proportions of each step in Alg. 1.

| Dataset | Model | Retain Grad. | Forget Grad. | Surgery | Reg. Grad. | Step |
|---|---|---|---|---|---|---|
| MNIST | MLP | 0.967 | 0.033 | 0.000 | 0.000 | 0.000 |
| | ResNet18 | 0.984 | 0.016 | 0.0010 | 0.000 | 0.000 |
| KMNIST | ResNet18 | 0.989 | 0.011 | 0.000 | 0.000 | 0.000 |
| SVHN | ResNet50 | 0.980 | 0.020 | 0.000 | 0.000 | 0.000 |
| CIFAR10 | ResNet18 | 0.989 | 0.011 | 0.000 | 0.000 | 0.000 |
| | ResNet50 | 0.992 | 0.008 | 0.000 | 0.000 | 0.000 |
| CIFAR100 | ResNet50 | 0.993 | 0.006 | 0.000 | 0.000 | 0.000 |

## L.20 TEST ACCURACY PLOT FOR PARETO FRONTIER EXPERIMENTS

We provide a plot characterizing test accuracy for Alg. 1 and Alg. 2 applied on MNIST for various choices of $\theta$ in Fig. L.12.

## L.21 RESULTS FOR ALG. 3

Due to the issues presented in Appx. F, we focus on Alg. 1 and Alg. 2 in the main paper and present results for Alg. 3 here. In Tab. L.24, we show that Alg. 1 can induce uniformity while maintaining accuracy for logistic regression, albeit with a stronger tradeoff than Alg. 2

## L.22 HYPERPARAMETER SENSITIVITY FOR ALG. 3

In what follows, we demonstrate the sensitivity of Alg. 3 to the choice of hyperparameters for logistic regression trained on MNIST (using the forward KL for uniformity). Here $b = 1$ and $n = 1000$. Below, the $x$-axis is the estimator recursive timesteps $t = 1, ..., n$ and the $y$ axis is the difference between the exact Newton step and the estimated Newton step in Frobenius norm (before any Gaussian noise is added). Specifically, we find that if $\lambda \approx |\lambda_{\min}|$ and $J \approx |\lambda + \lambda_{\min}|$ does not hold, the estimator diverges very quickly. Specifically, as shown in Fig. L.13, if $\lambda = 1.0, J = 2.5$

Table L.21: Warmup values for Alg. 2 for logistic regression trained over MNIST, contrasted with the values after Alg. 2 is applied.

| $\theta$ | Method | Retain Acc. | Test Acc. | Conf. Dist. (Lower Better) |
|---|---|---|---|---|
| 0.0 | Warmup | 91.4% | 91.6% | 0.765 |
| 0.0 | Alg. 2 | 92.5% | 92.2% | 0.738 |
| 0.25 | Warmup | 90.1% | 90.4% | 0.283 |
| 0.25 | Alg. 2 | 91.6% | 91.5% | 0.324 |
| 0.50 | Warmup | 89.3% | 89.7% | 0.215 |
| 0.50 | Alg. 2 | 90.5% | 90.6% | 0.300 |
| 0.75 | Warmup | 88.4% | 88.6% | 0.154 |
| 0.75 | Alg. 2 | 87.1% | 87.2% | 0.280 |
| 0.95 | Warmup | 85.4% | 86.0% | 0.097 |
| 0.95 | Alg. 2 | 85.0% | 85.7% | 0.092 |

Table L.22: Results for Alg. 1 applied to logistic regression and MLP trained over KMNIST, $\theta = 0.75$ for Alg. 1.

| Model | Method | Retain Acc. | Test Acc. | Conf. Dist. (Lower Better) |
|---|---|---|---|---|
| LogReg | Pretrain | 81.4% | 66.4% | 0.775 |
| | Retrain | 80.9% | 65.3% | 0.770 |
| | Alg. 1 | 77.4% | 63.4% | 0.770 |
| MLP | Pretrain | 100% | 88.4% | 0.900 |
| | Retrain | 100% | 88.5% | 0.887 |
| | Alg. 1 | 92.8% | 80.3% | 0.039 |

Table L.23: Results for the synthetic baseline applied with various sampled $k$ on a MLP over MNIST and a ResNet18 over CIFAR10. We observe that increasing $k$ yields better performance, but nevertheless even very large $k$ (an additional 50k instances, with a forget set size of 100) fails to induce uniformity for CIFAR10.

| Dataset | Model | Sampled $k$ | Retain Acc. | Test Acc. | Conf. Dist. (Lower Better) |
|---|---|---|---|---|---|
| MNIST | MLP | 5 | 99.4% | 97.1% | 0.541 |
| | | 25 | 99.0% | 96.4% | 0.183 |
| | | 125 | 99.4% | 96.8% | 0.105 |
| | | 250 | 98.6% | 96.0% | 0.066 |
| | | 500 | 99.6% | 96.3% | 0.003 |
| CIFAR10 | ResNet18 | 5 | 99.0% | 91.0% | 0.683 |
| | | 25 | 99.2% | 91.3% | 0.865 |
| | | 125 | 98.8% | 90.9% | 0.856 |
| | | 250 | 98.4% | 90.7% | 0.869 |
| | | 500 | 98.3% | 91.1% | 0.852 |
| | | 5000 | 94.0% | 89.7% | 0.844 |

(here, $\lambda$ is off by 0.5) or $\lambda = 0.5, J = 2.0$ (here, $J$ is off by 0.5), the estimator diverges before timestep 150, resulting in integer overflow. However, when $\lambda = 0.5, J = 2.5$, the estimator stays stable across samples and does not diverge. Still, the estimator does not converge but rather oscillates, demonstrating the poor sample complexity of the estimator as well–although it is asymptotically unbiased, it may take many samples to converge.

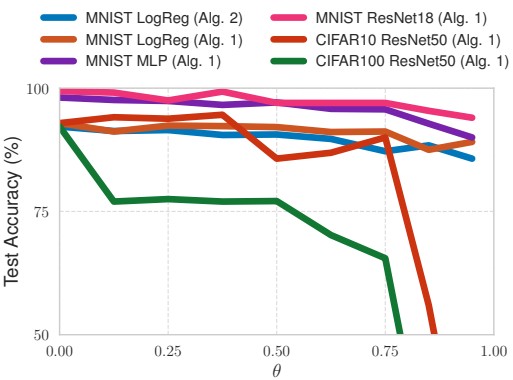

Figure L.12: Test Accuracy vs. $\theta$, MNIST. This has similar behavior to Fig. 3a.

Table L.24: Results for Alg. 3 on MNIST across different models. We compare against Pretraining, Alg. 1, and Alg. 2 (where available). We observe that Alg. 3 achieves competitive uniformity (Conf. Dist.) on MLP, though with a trade-off in retain accuracy.

| Dataset | Model | Method | Retain Acc. | Test Acc. | Conf. Dist. (Lower Better) |
|---|---|---|---|---|---|
| MNIST | LogReg | Pretrain | 92.1% | 91.8% | 0.807 |
| | | Alg. 1 | 87.8% | 87.7% | 0.180 |
| | | Alg. 2 | 87.1% | 87.2% | 0.280 |
| | | Alg. 3 | 77.9% | 77.3% | 0.267 |
| | SVM | Pretrain | 92.3% | 92.1% | 0.543 |
| | | Alg. 1 | 90.3% | 90.1% | 0.241 |
| | | Alg. 3 | 83.9% | 83.9% | 0.215 |
| | MLP | Pretrain | 98.3% | 97.0% | 0.886 |
| | | Alg. 1 | 97.5% | 95.7% | 0.037 |
| | | Alg. 3 | 88.8% | 87.5% | 0.057 |

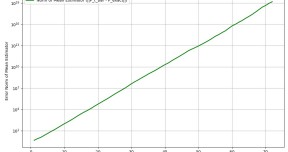 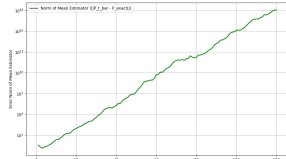 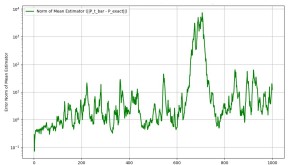

(a) $\lambda = 1.0$ $J = 2.5$. Here, $J$ is off by 0.5, and the estimator diverges. (b) $\lambda = 0.5$, $J = 2.0$. Here, $\lambda$ is off by 0.5, and the estimator diverges. (c) $\lambda = 0.5$, $J = 2.5$. Here, the estimator stays roughly stable.

Figure L.13: We observe that small changes in hyperparameters result in divergence of the inverse Hessian estimator in Alg. 3.

## M   BROADER IMPACTS

Potential positive impacts are motivated by the threat model as discussed in Sec. 1 and Appx. A ; per our example provided in the introduction, violations of test-time privacy constitute a real threat for ML safety. Hence, providing the defense that we do constitutes a positive societal impact. However, we acknowledge the potential danger in providing a new threat model–it is possible that potential adversaries had not thought of this before. Still, we provide a way to address this threat.

## N  SYMBOL TABLE

| **Symbols** | |
|---|---|
| $f$ | A pretrained classifier. |
| $f_u$ | A pretrained classifier after unlearning has been conducted over $\boldsymbol{x}_p$. |
| $\boldsymbol{x}_p$ | A data instance corresponding to person $p$. |
| $\mathcal{X}$ | A sample space, subset of $\mathbb{R}^d$. |
| $\mathcal{Y}$ | A label space, subset of $\mathbb{R}^o$. |
| $\mathcal{Z}$ | The Cartesian product of a sample space and a label space. This is the space where a dataset is drawn from. |
| $\mathcal{D}$ | A dataset, subset of $\mathcal{Z}^n$, which is the n-fold Cartesian product of $\mathcal{Z}$. This represents a set of $n$ data instances. |
| $\mathcal{D}_f$ | A forget set, a subset of a dataset $\mathcal{D}$. |
| $\mathcal{D}_r$ | A retain set, the complement of the forget set in $\mathcal{D}$. |
| $\mathcal{W}$ | A space of parameters, subset of $\mathbb{R}^p$. |
| $\mathcal{A}$ | A function that maps datasets to parameters; this represents the learning algorithm that a ML model provider uses throughout our paper. |
| $\mathcal{H}_{\mathcal{W}}$ | A set of functions which map samples in $\mathcal{X}$ to the probability simplex $\Delta_{|\mathcal{Y}|}$, parameterized by a $\boldsymbol{w} \in \mathcal{W}$. |
| $\|\boldsymbol{w}\|_2$ | The $\ell_2$ norm of a vector $\boldsymbol{w}$. |
| $\|A\|_2$ | The 2-operator norm of a matrix $\boldsymbol{A}$. |
| $\lambda$ | An $\ell_2$ regularization coefficient used in Alg. 1 for regularization and the certified algorithms e.g. Alg. 2 for local convex approximation. |
| $\lambda_{\min}(\boldsymbol{A})$ | The minimum eigenvalue of a matrix $\boldsymbol{A}$. |
| $\mathcal{K}$ | A uniform learner, which maps samples to parameters which, when one parametrizes a function by any such parameter, a uniform distribution over all possible labels, $U[0, |\mathcal{Y}|]$ is outputted. |
| $f_{\boldsymbol{w}}$ | A classifier parameterized by a parameter $\boldsymbol{w} \in \mathcal{W}$. |
| $\mathcal{L}_{\mathcal{A}}$ | A loss function to yield accurate predictions e.g. the cross entropy loss between model predictions and labels. |
| $\mathcal{L}_{\mathcal{K}}$ | A loss function to yield accurate uniformity e.g. the Kullback-Liebler divergence between softmax outputs and uniform distribution. |
| $\theta$ | A trade off parameter in $(0, 1)$ between utility and uniformity. |

| | |
|---|---|
| | **Symbols** |
| $\mathcal{M}_\theta$ | A map between datasets and parameters that is the minimizer of a Pareto objective between $\mathcal{L}_\mathcal{A}$ and $\mathcal{L}_\mathcal{K}$, where $\theta$ spans the (convex) Pareto frontier. |
| $J$ | The bound on the norm of the Hessian at $\boldsymbol{w}^*$. |
| $\mathcal{D}_f^{(i)}$ | The $i$th instance of the forget set. |
| $\mathcal{D}_r^{(j)}$ | The $j$th instance of the retain set. |
| $\mathcal{D}_r^{(j,\mathcal{X})}$ | The feature of the $j$th instance of the retain set. |
| $\mathcal{D}_r^{(j,\mathcal{Y})}$ | The label of the $j$th instance of the retain set. |
| $\approx_{\varepsilon,\delta,\mathcal{T}}$ | Used to denote when two algorithms are $(\varepsilon,\delta)$ indistinguishable across all subsets $\mathcal{T} \subset \mathcal{W}$, i.e. $\mathcal{M}(\mathcal{D}) \approx_{\varepsilon,\delta,\mathcal{T}} \mathcal{M}'(\mathcal{D}')$ means that $\Pr[\mathcal{M}(\mathcal{D}) \in \mathcal{T}] \le e^\varepsilon \Pr[\mathcal{M}'(\mathcal{D}') \in \mathcal{T}] + \delta$ and $\Pr[\mathcal{M}'(\mathcal{D}') \in \mathcal{T}] \le e^\varepsilon \Pr[\mathcal{M}(\mathcal{D}) \in \mathcal{T}] + \delta$. |
| $C$ | A bound on the model weights. |
| $P_\mathcal{K}$ | The Lipschitz constant for the gradients of $\ell_\mathcal{K}$, the component loss functions of $\mathcal{L}_\mathcal{K}$, from Asm. 5.3. |
| $P_\mathcal{A}$ | The Lipschitz constant for the gradients of $\ell_\mathcal{A}$, the component loss functions of $\mathcal{L}_\mathcal{A}$, from Asm. 5.3. |
| $F_\mathcal{K}$ | The Lipschitz constant for the Hessians of $\ell_\mathcal{K}$, the component loss functions of $\mathcal{L}_\mathcal{K}$, from Asm. 5.4. |
| $F_\mathcal{A}$ | The Lipschitz constant for the Hessians of $\ell_\mathcal{A}$, the component loss functions of $\mathcal{L}$, from Asm. 5.4. |
| $P$ | A convex combination of $P_\mathcal{K}$ and $P_\mathcal{A}$ with respect to $\theta$. |
| $F$ | A convex combination of $F_\mathcal{K}$ and $F_\mathcal{A}$ with respect to $\theta$. |
| $\mathcal{N}(0, \sigma^2 \boldsymbol{I})$ | The standard normal distribution with an isotropic covariance matrix. |
| $\nabla_{\boldsymbol{w}^*,\mathcal{K},\mathcal{A}}$ | The gradient of the Pareto objective evaluated at $\boldsymbol{w}^*$, used in the main paper with regularization. |
| $\boldsymbol{H}_{\boldsymbol{w}^*,\mathcal{K},\mathcal{A}}$ | The Hessian of the Pareto objective evaluated at $\boldsymbol{w}^*$, used in the main paper without regularization. |

