# OpenReview forum: "Inducing Uncertainty on Open-Weight Models for Test-Time Privacy in Image Recognition"
_ICLR.cc/2026/Conference — Submitted to ICLR 2026_

### Official Review · Reviewer_TfiQ · 2025-10-31

**Soundness:** 2
**Presentation:** 3
**Contribution:** 2
**Rating:** 4
**Confidence:** 3

**Summary:**

This paper studies a novel test-time privacy threat model in which correct predictions on certain sensitive queries may leak private information. To provide “test-time privacy,” the goal is to output a model that predicts uniformly on the sensitive queries while remaining accurate on the rest. The authors define a target reference model $\mathcal{M}_{\theta}$, trained with an objective that is a regularized, weighted sum of (i) minimizing loss on the retain set and (ii) encouraging predictions close to uniform on the forget set, with $\theta$ controlling the tradeoff.

They then define an $(\epsilon, \delta, \theta)$-certified Pareto learner as an algorithm that modifies the ERM solution on the entire query set so that its output is indistinguishable from the reference model $\mathcal{M}_{\theta}$, where $\epsilon, \delta$ measure indistinguishability similar to certified unlearning or differential privacy. The proposed certified Pareto learner takes one Newton step toward the reference model and then adds calibrated Gaussian noise.

Theoretical analysis characterizes the privacy–utility tradeoff by upper-bounding the accuracy gap between the “unlearned” model and $\mathcal{M}_{\theta}$. Experiments further show that retain-set utility remains high, even for relatively large $\theta$.

**Strengths:**

The paper is clearly written and introduces a novel, relevant threat model that is both timely and worth investigating.

**Weaknesses:**

- The definition of a certified-Pareto learner requires indistinguishability between the “unlearned’’ model and the reference model $\mathcal{M}_{\theta}$. This introduces an extra parameter $\theta$, which makes the guarantees harder to interpret: both the privacy and utility statements are only relative to the chosen reference model.

- Theorem 4.5 bounds the accuracy drop only relative to $M_\theta$. However, a large $\theta$ can already degrade $M_\theta$’s accuracy and still may not ensure strong test-time privacy. In particular, selecting $\theta$ via Equation (7) and Corollary 4.2 implies that achieving $\theta > 0.75 $ requires $\epsilon = \sqrt{|D_r| \ln |Y|} $, which exceeds 1 even for moderate $|D_r|$. At the same time, $\theta = 0.75$ appears to induce substantial error for $\mathcal{M}_{\theta}$ in the experiments.

Minor notation suggestion: consider renaming the $\epsilon$ in Equation (7), since it differs from the $\epsilon$ used in the certified-Pareto learner’s indistinguishability guarantee.

**Questions:**

- As I understand it, the goal is to produce a model that outputs (approximately) uniform predictions on a protected set of sensitive queries while remaining accurate on the rest. Why is uniformity the right target for privacy—rather than, for example, defining test-time privacy as an upper bound on accuracy over the sensitive set (e.g., “no better than chance” or some specified cap)?

- Pushing predictions toward uniformity on the sensitive set appears to necessarily reduce overall performance, with the magnitude depending on both the sensitive set and the retained set. Can you provide lower bounds on the accuracy drop of the reference model?

---

> ### Author Response · Authors · 2025-11-18
> **Rebuttal by the Authors**
>
> We thank the reviewer TfiQ for their thorough feedback. We appreciate their compliments on the clarity of our writing and the timeliness of our proposed threat model. Below, we address the reviewer's questions.
>
> ### > **Q1**: Is it a weakness of the TTP definition that it is evaluated only against the chosen reference model $\mathcal{M}_{\theta}$?
>
> **A1**: Excellent question. The TTP definition that we provide is motivated by and mirrors previous definitions provided in differential privacy and certified unlearning, which also compare to a reference leave-one-out or retrained-from-scratch model, respectively. The implicit assumption with all of these definitions is that the reference model performs well for the task. As such, we do not believe this to be problematic. If the reviewer wants further clarification on this, we are happy to provide related references.
>
> ### > **Q2**: Does it hurt the interpretability of the TTP definition to include the extra $\theta$ parameter?
>
> **A2**: We thank the reviewer for raising the issue of interpretability. This is a strong characteristic of our approach. In fact, we believe the extra $\theta$ parameter provides additional interpretability. In our definition, $\varepsilon$ characterizes the indistinguishability, $\delta$ the catastrophic failure probability, and $\theta$ *the strength of the TTP induced on the pretrained model*. For example, the certification provided by an $(\varepsilon, \delta, 0.25)$ model is clearly weaker than that provided by an $(\varepsilon, \delta, 0.75)$ model.
>
> ### > **Q3**: Do large $\theta$, like $\theta = 0.75$, ensure TTP while resulting in high retain/test accuracies?
>
> **A3**: We thank the reviewer for raising this thoughtful question. In practice, we observe that $\theta = 0.75$ provides strong uniformity without sacrificing retain/test accuracies for large-scale models and benchmarks. Below, we attach results from our existing Table L.3. For a vision transformer pretrained on CIFAR100, we observe a <3% drop in retain accuracy with nearly exact uniformity, as demonstrated below. Furthermore, as demonstrated in Fig. 3, for a ResNet50 trained on CIFAR10 or CIFAR100, $\theta > 0.75$ results in poor retain accuracy while $\theta < 0.75$ results in poor confidence distance.
>
> | **Dataset** | **Model** | **Method** | **Retain Acc.** | **Test Acc.** | **Conf. Dist. (Lower Better)** |
> | :--- | :--- | :--- | :--- | :--- | :--- |
> | CIFAR100 | ViT_B_16 | Pretrain | 94.2% | 91.2% | 0.972 |
> | | | **Alg. 1** | **91.6%** | **88.6%** | **0.074** |
>
> Connecting this with our mathematical results, while Corollary 5.2 generally results in a large $\theta$ for small $\varepsilon$ and realistic $|\mathcal{D}_r|$ and $|\mathcal{Y}|$, it only provides a sufficient, *but not necessary*, condition for Eq. 7 to hold. That is, it is perfectly consistent that we can choose $\theta = 0.75$ and still obtain strong uniformity without sacrificing retain/test accuracies. We have rephrased Corollary 5.2 in the paper for clarity.

---

> > ### Author Response · Authors · 2025-11-18
> > **Rebuttal by the Authors (cont.)**
> >
> > ### > **Q4**: Why is uniformity optimal for TTP, instead of upper bounding accuracy on $\mathcal{D}_f$?
> >
> > **A4**: We greatly appreciate the reviewer raising this clever question. In our setting, the TTP adversary aims to obtain a harmful prediction $m$ on instance $\mathbf{x}_p$ to use as a warrant to cause harm to a person $p$. As such, we would ideally like to obfuscate $m$ completely. However, as mentioned in the threat model in Appx. A (now summarized in Sec. 3), we *do not assume* that the model provider knows which of the classes the harmful prediction $m$ belongs to. For example, model providers like Google which release e.g. the Gemini series of LLMs, do not know who is going to use their model in advance.
> >
> > Thus, upper bounding accuracy (i.e. inducing misclassification) does not protect against TTP attacks. To make this concrete, following the medical insurance example in our introduction, suppose the model provider misclassified $\mathbf{x}_p$ as “Benign” instead of “Melanoma” to protect against the insurance provider. But, as it turns out, the model user is a medical facility instead of the insurance provider. Thus, doctors may miss out on a critical medical diagnosis for a person $p$, again resulting in harm. We adjusted our existing example relating to this in  Appx. C to make this more clear.
> >
> > Furthermore, in this case, inducing uniformity is *optimal*. Since the model provider does not know which class $m$ to protect beforehand, they must ensure that the TTP adversary is uncertain across *all* classes. We have adjusted Appx. A to better reflect this. Please let us know if you have another suggestion for ensuring TTP in mind.
> >
> > ### > **Q5**: Does pushing predictions towards uniformity necessarily reduce overall utility, and is there a lower bound for the utility decrease?
> >
> > **A5**: We appreciate the reviewer’s insight in asking this question. Yes, there is a trade-off, which is present in all privacy-preserving machine learning methods. However, finetuning with Alg. 1 might actually *increase* retain accuracy. In particular, we provide results from our paper’s Table L.3 below, in which we observe that Alg. 1 applied to a ResNet50 pretrained on TinyImageNet results in an *increased* retain accuracy from 91.2% to 92.1%, while still inducing strong uniformity on the forget set.
> >
> > | **Dataset** | **Model** | **Method** | **Retain Acc.** | **Test Acc.** | **Conf. Dist. (Lower Better)** |
> > | :--- | :--- | :--- | :--- | :--- | :--- |
> > | TinyImageNet | ResNet50 | Pretrain | 91.2% | 83.5% | 0.812 |
> > | | | **Alg. 1** | **92.1%** | **81.6%** | **0.197** |
> >
> > While we do not provide a lower bound for the utility decrease, we do provide an upper bound in Theorem 5.5, demonstrating that the size of the parameters, regularization, and choice of $\theta$ critically affect the change in accuracy after applying $\mathcal{M}_{\theta}$. We additionally believe that the size of the forget set and the number of labels will be involved in the lower bound.
> >
> > However, providing an explicit lower bound is challenging, and well beyond the scope of this work. In particular, to our knowledge, for the more frequently studied and related setting of machine unlearning, while upper bounds similar to our Theorem 5.5 are given, approaches to proving lower bounds on the generalization error (or the sample complexity necessary for a certain level of generalization error) cannot be adapted to our TTP setting due to critical fundamental differences. For example, [1] relies on supposing the pretrained model is $(\varepsilon, \delta)$-DP and then bounding population risk under DP, which is tractable, but then using the fact that $(\varepsilon, \delta)$-DP algorithms imply (\varepsilon, \delta)-certified unlearning, which is not the case in our setting. Similarly, [2] relies on doing exact unlearning, i.e. unlearning by retraining from scratch, whereas all our approaches are based on finetuning. A more recent ICLR’25 paper [3] proves the tightest lower bound on deletion capacity to date, but this holds only for strongly convex losses and relies heavily on the fact that $(\varepsilon, \delta)$-unlearning aims to mimic $\mathcal{A}(\mathcal{D}\_r)$, whereas  $(\varepsilon, \delta)$-TTP aims to mimic $\mathcal{M}_{\theta}(\mathcal{D})$. Finally, all of the aforementioned deletion capacity lower bounds rely on highly technical proofs. As such, we leave a comprehensive study of the equivalent of “deletion capacity” for TTP, as well as computing a  lower bound on the retain error or generalization error after TTP is applied, to future work.
> >
> > Still, we believe that providing such a lower bound is important to understand the general theoretical qualities of TTP, and we thus leave it to future work. We have updated our future directions in Appx. F to reflect this. We thank the reviewer for the recommendation.

---

> > > ### Author Response · Authors · 2025-11-18
> > > **Rebuttal by the Authors (cont.)**
> > >
> > > ### > **Q6**: How does the size of the forget set affect the overall performance after uniformity is induced?
> > >
> > > **A6**: While a full characterization of how the size of the forget set affects algorithms meant to ensure TTP is left for future work, we provide preliminary results in Appx. L.10. In particular, we demonstrate that as the forget set size increases, retain accuracy stays stable while forget set uniformity decreases slowly.  For example, after applying Alg. 1 to a MLP trained on MNIST, as the forget set size increases from 100 to 5000, retain accuracy goes from 97.5% to 96.6% while the induced uniformity on the forget set goes from 0.037 to 0.420. This large increase in uniformity is likely due to the fact that the forget set is ~10% of the training dataset. We attach these results, pulled from existing table L.14, below:
> > >
> > > | **Dataset** | **Model** | **Forget Size** | **Retain Acc.** | **Test Acc.** | **Conf. Dist. (Lower Better)** |
> > > | :--- | :--- | :--- | :--- | :--- | :--- |
> > > | Alg. 1 | MLP | 100 | 97.5% | 95.7% | 0.037 |
> > > | | | 250 | 97.6% | 95.1% | 0.010 |
> > > | | | 500 | 95.8% | 93.8% | 0.121 |
> > > | | | 1000 | 94.8% | 93.3% | 0.162 |
> > > | | | 5000 | 96.6% | 95.0% | 0.420 |
> > >
> > > ---
> > > We thank the reviewer for their several excellent questions about:
> > > 1. Our TTP definition and its interpretability;
> > > 2. Lower bounding the retain accuracy; and
> > > 3. The effect of the size of the forget set in ensuring TTP.
> > >
> > > We provided answers detailing:
> > > 1. The positives of our TTP definition in terms of standard assumptions and interpretability;
> > > 2. Experimental results demonstrating that retain accuracy may not necessarily drop, also detailing the challenges in deriving an analytical lower bound; and
> > > 3. Preliminary results on how the forget set size alters the assurances of our TTP algorithms.
> > >
> > > Each of these directions requires great attention in their own right, and are left to future work. In this paper, we defined the TTP problem and derived many of its formal underpinnings. We also obtained efficient algorithms that work in practice across many model and dataset pairs. We hope the community will be as excited by this new problem setting as we are, addressing several outstanding challenges like those pointed out by the reviewer.
> > >
> > > We again thank the reviewer and are happy to clarify further if any questions remain.
> > >
> > > ---
> > > ### References:
> > > [1] Huang et al. Tight Bounds for Machine Unlearning via Differential Privacy, Journal of Privacy and Confidentiality’23.
> > >
> > > [2] Ullah et al. ​​Machine Unlearning via Algorithmic Stability, COLT’21.
> > >
> > > [3] Allouah et al. The Utility and Complexity of In- and Out-of-Distribution Machine Unlearning, ICLR’25.

---

> > > > ### Author Response · Authors · 2025-11-27
> > > > **Official Comment by the Authors**
> > > >
> > > > Dear Reviewer TfIQ,
> > > >
> > > > We are thankful for the reviewer’s efforts so far. Given that the discussion period is finishing soon, we would like to ask if you have any more questions. During the rebuttal, we have clarified that:
> > > >
> > > > + The certified TTP definition mimics previous definitions in certified unlearning and differential privacy, which assume a reference model that performs well for the given task.
> > > > + The $\theta$ in the certified definition adds to the interpretability of the guarantee.
> > > > + Large $\theta$, like 0.75, can ensure high retain and test accuracy as well as strong uniformity on the forget set.
> > > > + Uniformity is optimal for TTP since the model provider does not know which adversary they are preventing the attack from.
> > > > + Pushing predictions towards uniformity does not necessarily decrease retain and test accuracy, and providing an analytical lower bound is nontrivial and outside the scope of our work.
> > > >
> > > > We also provided empirical results on how Alg. 1 scales as the forget set size increases. In the previous results, we used a fixed forget set batch size of 10 instances. **Below, we provide new empirical results for forget set size scaling with dynamic batch sizes**. In particular, for a MLP trained on MNIST, we scale the batch size proportionally with the forget set. Specifically, we divide the forget set size by 10. In doing so, we observe much better uniformity for a forget set size of 5000. We also see similar performance for forget set sizes of 1000 or less. We have updated the manuscript with these results in Table L.15.
> > > >
> > > >
> > > > | **Method** | **Model** | **Forget Size** | **Retain Acc.** | **Test Acc.** | **Conf. Dist. (Lower Better)** |
> > > > | :---: | :---: | :---: | :---: | :---: | :---: |
> > > > | Alg. 1 | MLP | 100 | 97.5% | 95.7% | 0.037 |
> > > > | | | 250 | 95.6% | 96.6% | 0.131 |
> > > > | | | 500 | 93.8% | 93.4% | 0.187 |
> > > > | | | 1000 | 92.3% | 92.2% | 0.192 |
> > > > | | | 5000 | 95.3% | 94.9% | 0.227 |
> > > >
> > > > We again thank the reviewer for their effort and time and would appreciate it if they could provide feedback on our responses. We are happy to clarify any further questions.
> > > >
> > > > Best,
> > > >
> > > > The Authors

---

### Official Review · Reviewer_8s6z · 2025-11-01

**Soundness:** 3
**Presentation:** 3
**Contribution:** 4
**Rating:** 8
**Confidence:** 2

**Summary:**

The work defines a new threat model called test-time-privacy, where open weight model is used by the adversary to inflict harm on person x based on the prediction about x from corrupted public data. This paper addresses the following RQ: Can we ensure test-time privacy against adversaries with access to an open-weight model? The work proposes algorithm to induce maximal uncertainty on the protected set while preserving utility on the rest.

**Strengths:**

- formulation of a new threat model: test-time-privacy (TTP) which addresses a key safety concern often understudied in machine learning literature,
- proposes a method to address the TTP threat,
- method balances uncertainty and utility,

**Weaknesses:**

- the method only applies to classification problems, what about generative models and others?

**Questions:**

How challenging is applying this method to generative models as current settings apply only to classification models?

---

> ### Author Response · Authors · 2025-11-18
> **Rebuttal by the Authors**
>
> We thank the reviewer 8s6z for their feedback. We appreciate their compliments on the novelty of our threat model and its relevance for the ML literature and highlighting that our proposed method explicitly balances uncertainty and utility. Below, we address the reviewer’s questions.
>
> ### > **Q1**: How can we motivate and protect TTP in the generative setting?
>
> **A1**: Excellent question. To motivate TTP in the generative setting, we included an example of how a text-to-image (T2I) diffusion model could be used to violate TTP in Appx. E. We repeat it here for clarity.
>
> Consider a text-to-image generative model trained on a large dataset, including web data, which has web articles and associated images. A popular news site publishes an article about a businessperson, but mistakenly uses a picture of an unrelated individual $p$, $\mathbf{x}_p$, as the header image. This creates a strong, albeit false, association between this person’s likeness and the (perhaps negative) content of the article. When prompted with a string similar to the headline of the news article, the model generates an image (or a similar image) of person $p$, algorithmically cementing a false narrative about person $p$. A TTP adversary could leverage such a model to, for example, deny a person $p$’s job application.
>
> To solve this issue, we could leverage our proposed *finetuning Pareto learner* framework. Specifically, in our paper’s Sec. 4.1, we define the *finetuning Pareto learner* as providing an optimal tradeoff between the utility on the retain instances and softmax uniformity on the forget set. This yielded Alg. 1, which has pseudocode on lines 324-333 in the most recent manuscript. In the generative setting, we could define the Pareto learner as providing high-fidelity generations for the retain set, but low-fidelity generations for the forget set. This way, a TTP adversary would be unsure of the information related to person $p$.
>
> Nonetheless, providing a generative analogue of Alg. 1 using this framework is nontrivial. Firstly, defining a *clear and differentiable* objective for the forget set is complex. In our current (discriminative) setting, we use the simple objective of maximizing softmax uniformity with the KL. In the generative setting, this no longer works, since there are infinitely many “classes”. Furthermore, the objective of “low-fidelity” given above is ambiguous: it could mean generating a blurry/noisy image, generating a semantically different image, or refusing to generate an image at all. Each of these approaches would require a different and more complex loss function than that used in Alg. 1.
>
> Next, the “forget set” in the generative setting is often a *concept* or *association*, not a discrete set of instances. Alg. 1 finetunes on a specific set of inputs $\mathcal{D}_f$. To ensure TTP in the T2I example given above, we must remove the association between a class of prompts and the likeness of a person $p$. This association may be encoded diffusely across the model's weights. Thus, ensuring TTP with finetuning is ill-posed in this setting.
>
> As such, ensuring TTP in the generative setting is an exciting challenge. We have updated our paper’s Appx. F to highlight this extension as a key direction for future work.
>
> ---
> We thank the reviewer for their question about extending TTP to the generative setting. Above, we detailed the gaps in ensuring TTP for generative models. This direction is important and must be systematically analyzed in future work, building off our formalization; efficient algorithms; and experimental design for the discriminative setting.
>
> We again thank the reviewer and are happy to clarify further if any questions remain concerning the threat model, the formalization, the algorithms, or the experiments of this paper.

---

### Official Review · Reviewer_TVYj · 2025-11-03

**Soundness:** 3
**Presentation:** 3
**Contribution:** 2
**Rating:** 4
**Confidence:** 5

**Summary:**

The paper considers a privacy game where a set of examples D_f are considered "incorrect." That is, D_f is assumed to contain false, or corrupted data individuals. An attacker wants to claim something nefarious about the individual associated with x in D_f by demonstrating that a model f labels x in some way. In other words, by demonstrating f(x) = y_bad, the attacker can achieve some benefit. For example, the attacker might be an insurer looking to (falsely) deny coverage to the user associated with x.

A model curator, who also knows the set D_f, wants to defend against any such attacker. To do so, the curator wishes to output a model that labels any x in D_f randomly. The setting assumes that the curator will be publishing the weights of f, and so certain trivial solutions are out of scope. For example, the curator cannot route examples x in D_f to a random labeler and all other examples to some true model.

The paper sets up the Exact Pareto Learner as a solution for the problem, which is the minimizer of a loss function that rewards uniformity on D_f and correctness on D \ D_f. These two components are balanced by a parameter theta, which takes a convex combination of these two losses.

The paper then provides an algorithm that is epsilon, delta - indistinguishable (in the unlearning sense) from the Exact Pareto Learner.  This result holds under Lipschitzness assumptions on the loss functions and boundedness of the parameter space. The paper provides empirical results that demonstrate that finding a local minimum for the Exact Pareto Learner finds a good balance between obscuring labels on D_f and attaining good accuracy outside of D_f.

**Strengths:**

Strength #1: I think the societal problem of relying on model-based predictions that are false is an interesting one, and one that could use more technical solutions. This work is an interesting first step in that direction. I commend the originality of the setting in this regard.

**Weaknesses:**

Weakness #1: One obvious solution to this problem is retain a model mixing D_f into the training distribution with random labels. Behind the formalism of the paper, this is essentially what the Exact Pareto Leaner is doing. While I do not think ideas should be penalized for being simple, this seems especially straightforward.

Weakness #2: The entire point of designing an epsilon, delta indistinguishable algorithm is practicality. If the curator wants to protect some set D_f, they would prefer to modify a pretrained model rather than retrain something like an LLM from scratch. Thus, it is somewhat disappointing that the indistinguishable algorithm is in fact the impractical one. Theoretically, it requires inverting the Hessian of a potentially large-parameter model. The authors offer an algorithm that circumvents this in the Appendix. However, no empirical results are provided, and the obvious solution essentially wins the day.

Weakness #3. A third weakness is non-technical and concerns the motivation for the setting. The set D_f is assumed to be known as corrupted by the adversary, the curator, and presumably also the individuals whose data constitute D_f and recognize their features as false. I find it hard to motivate that there is essentially public agreement that these are incorrect features, and yet the adversary would be able to execute the hypothetical fraud where they demonstrate a model’s output as a pretext for causing harm to the individual (e.g. by denying insurance coverage). To put it another way. the paper implicitly imagines a scenario where an individual with corrupted feature vector x is saved by demonstrating to some third-party that f(x) is random. However, it seems to me that, realistically, they are equally saved by demonstrating to the third-party that x is corrupted.

Weakness #4: I think the presentation would be improved by moving the security game in appendix A to the main body. It is difficult to exactly understand the threat model without it.

**Questions:**

Can the authors provide an explanation for the limitations of Algorithm 3? Why was it not evaluated empirically? Is it because the estimation error of the Hessian is too high to achieve utility? Some other reason?

---

> ### Author Response · Authors · 2025-11-18
> **Rebuttal by the Authors**
>
> We thank the reviewer TVYj for their insightful feedback. We appreciate their compliments on the novelty of our work in taking a first step towards solving the societal problems which arise due to incorrect model predictions. Below, we address the reviewer’s questions.
>
> ### > **Q1**: Is the exact Pareto learner retraining a model from scratch?
>
> **A1**: Critically, **the exact Pareto learner is not retraining from scratch, but rather finetuning a pretrained model**, as detailed in the Alg. 1 pseudocode (lines 324-333). Thus, our algorithm is indeed practical for large-scale models and benchmarks.
>
> To make this clearer, we edited our paper and renamed the “exact Pareto learner” to the “finetuning Pareto learner” systematically.
>
> ### > **Q2**: Can we simply incorporate $\mathcal{D}_f$ into the training distribution with random labels and then retrain?
>
> **A2**: We cannot, since it *does not scale well to harder models and datasets*. Following the reviewer’s comment, we conduct the following experiment: We split the training data into retain and forget sets, where the forget set consists of 100 instances. We then randomly and uniformly relabeled the forget set before training the model on the combined retain and relabeled forget sets.
>
> The results below, for a ResNet18 trained on MNIST and ResNet50 trained on CIFAR10, indicate that training the model with randomly assigned labels on the forget set (“Relabel”) cannot induce uniformity, especially when compared to Alg. 1. This is not surprising as the model learns high confidence on these instances, which is far from the goal of achieving uniform distribution over the forget set. In contrast, our Alg. 1 *explicitly* specifies this goal in its objective.
>
> | **Dataset** | **Model** | **Method** | **Retain Acc.** | **Test Acc.** | **Conf. Dist. (Lower Better)** |
> | :--- | :--- | :--- | :--- | :--- | :--- |
> | MNIST | ResNet18 | Pretrain | 98.0% | 98.1% | 0.877 |
> | | | Relabel | 100.00% | 99.4% | 0.893 |
> | | | **Alg. 1** | **99.6%** | **99.1%** | **0.070** |
> | CIFAR10 | ResNet50 | Pretrain | 100.0% | 95.2% | 0.900 |
> | | | Relabel | 100.0% | 95.3% | 0.885 |
> | | | **Alg. 1** | **94.7%** | **90.0%** | **0.270** |
>
> Our original submission also tried incorporating randomly labelled instances from the ball around $\mathcal{D}_f$ into the training distribution, and retraining from scratch. However, *this approach also does not scale*. This is the “synthetic” baseline detailed in our experiments throughout our paper.  Notice that the exact Pareto learner (Alg. 1) *does* scale to larger models and datasets. For clarity, we provide results for a ResNet50 pretrained on CIFAR10, in which the Synthetic baseline fails to induce uniformity on the forget set, while Alg. 1 achieves >94% accuracy and strong uniformity. These results are in Table L.1 in the Appendix.
>
> | **Dataset** | **Model** | **Method** | **Retain Acc.** | **Test Acc.** | **Conf. Dist. (Lower Better)** |
> | :--- | :--- | :--- | :--- | :--- | :--- |
> | CIFAR10 | ResNet50 | Pretrain | 100.0% | 95.2% | 0.900 |
> | | | Synthetic | 91.4% | 87.4% | 0.818 |
> | | | **Alg. 1** | **94.7%** | **90.0%** | **0.270** |
>
> ### > **Q3**: Although it should not be penalized for its simplicity, isn’t the exact (finetuning) Pareto learner very conceptually straightforward?
>
> **A3**: We agree with the reviewer that our finetuning algorithm should not be penalized for its simplicity. The core contribution of our work is the identification and formalization of the novel TTP threat model, which the reviewer praised as a “strength.” On top of Alg. 1’s formulation, there was significant engineering work done in implementing it correctly. For example, we implemented gradient surgery [1] in order to correctly align the gradients over the retain and forget sets at every iteration. We also did extensive hyperparameter tuning and exploration, as demonstrated in Appx. L.5 with the exploration of the Pareto frontier.
>
> Furthermore, we are also particularly happy that our formal results have led to the development of a conceptually straightforward approach which (based on our comprehensive empirical analyses) provides a strong solution across various model and dataset complexities. The simplicity of this approach is advantageous for deployment in real-world settings, as well as its reproducibility and inclusion as a key baseline in future solutions to the significant problem of TTP that we have introduced. So, with respect, we regard this as a point of strength of our paper, not a weakness.
>
> We hope that the research community can leverage these results to further extend Alg. 1. For clarity, we have updated Appx. K.4 with an explanation of our gradient surgery implementation.

---

> ### Author Response · Authors · 2025-11-18
> **Rebuttal by the Authors (cont.)**
>
> ### > **Q4**: Is the adversary aware that $\mathcal{D}_f$ is corrupted, and does this matter for TTP?
>
> **A4**: The adversary is *not necessarily aware that $\mathcal{D}_f$ is corrupted*, even though they have access to the instances in $\mathcal{D}_f$ itself. In practice, an adversary will likely not inspect each instance in their dataset.
>
> However, we believe that whether the adversary is aware of the corruption in $\mathcal{D}_f$ or not is immaterial to the adversary’s goals. In particular, the adversary aims to leverage the predictions of model $\hat{f}$ as an *automated warrant* to e.g. deny insurance coverage. That is, we assume throughout that the TTP adversary is relying on the model to make their decisions. Thus, whether the adversary is aware of the corruption or not, they still need to obtain the adversarial prediction $m$ in order to harm an individual. Furthermore, we *assume a worst-case adversary* which, even if they know the data is corrupted, still tries to obtain the harmful prediction $m$. This can occur if the adversary is malicious or negligent. Thus, through inducing uncertainty, we ensure that the worst-case TTP adversary does not have this warrant.
>
> ### > **Q5**: To ensure TTP, can the person $p$ just demonstrate to a third-party that their instance $\mathbf{x}_p$ is corrupted instead of inducing uniformity?
>
> **A5**: The reviewer raises an excellent point. Firstly, the burden of proving to a third-party that a data point is corrupted can be highly nontrivial for a non-technical person. Secondly, our work is motivated by *prevention* rather than *correction*. The adversary’s system is automated. Our goal is to prevent this automation from causing harm *before* the harm occurs, eliminating the need for a costly, time consuming, and technically challenging appeals process. We have updated the paper to reflect this insight, clarifying this after detailing the full threat model in Appx. A.
>
> ### > **Q6**: Can the security game be moved to the main paper?
>
> **A6**: We thank the reviewer for highlighting the importance of the security game to the paper’s readability. We have updated the paper and added a concise version of the security game to Sec. 3 of the main paper. Please let us know if you believe additional information is required there.
>
> ### > **Q7**: Why are there no empirical results for Alg. 3, which circumvents inverting the Hessian in Alg. 2?
>
> **A7**: We prove in Appx. G that Alg. 3 achieves certified TTP without assuming convexity or needing to invert the Hessian. However, effectively implementing Alg. 3 is difficult in practice. In particular, Alg. 3 is *much more sensitive* to hyperparameters than Alg. 1 or Alg. 2. Furthermore, although the inverse Hessian estimator is asymptotically unbiased, it is very *sample inefficient*. Finally, Alg. 3 relies on local convex approximation, which results in a certified Newton step in a direction *off the Pareto frontier*. Due to these limitations, we decided to defer Alg. 3 to the Appendix and focus primarily on Alg. 1 and Alg. 2. We have updated our limitations section in Appx. F as well as provided empirical evidence for these limitations in Sec. L.22 to better reflect this. Additionally, in Appx. F, we detail why adopting other Hessian-free approaches e.g. [2] is nontrivial.
>
> Below (and in Sec. L.21), we have run Alg. 3 on logistic regression, with hyperparameters in Appx. K. We find that Alg. 3 still can achieve uniformity, but with a higher drop in retain accuracy due to estimator error. We are currently performing a hyperparameter grid search on Alg. 3 for SVM and a 2-layer MLP and will update the reviewer with these results in the next few days.
>
> | **Dataset** | **Model** | **Method** | **Retain Acc.** | **Test Acc.** | **Conf. Dist. (Lower Better)** |
> | :--- | :--- | :--- | :--- | :--- | :--- |
> | MNIST | LogReg | Pretrain | 92.1% | 91.8% | 0.807 |
> | | | Alg. 1 | 87.8% | 87.7% | 0.180 |
> | | | Alg. 2 | 87.1% | 87.2% | 0.280 |
> | | | Alg. 3 | 77.9% | 77.3% | 0.267 |
>
> ---
> We thank the reviewer for their several comprehensive questions about:
> 1. Our finetuning Pareto learner and Alg. 1;
> 2. The assumptions of our threat model; and
> 3. Our proposed Alg. 3.
>
> We provided answers detailing:
> 1. Our Alg. 1 compared to which simply relabel the forget set and retrain, as well as the nontrivial engineering challenges which arose while implementing Alg. 1;
> 2. That the adversary uses an automated approach to decision-making, and we aim to ensure TTP by addressing this beforehand rather than after the fact, whether the adversary knows about the dataset corruption or not; and
> 3. Preliminary results for Alg. 3, along with details concerning its limitations in practice compared to Alg. 1.
>
> We again thank the reviewer and are happy to clarify further if any questions remain.
>
> ---
> ### References:
>
> [1] Yu et al. Gradient Surgery for Multi-Task Learning, NeurIPS’20.
>
> [2] Koloskova et al. Certified Unlearning for Deep Neural Networks, ICML’25.

---

> ### Author Response · Authors · 2025-11-27
> **Official Comment by the Authors**
>
> Dear Reviewer TVYj,
>
> We are thankful for your effort and time. Since the rebuttal period is closing soon, we would like to ask if you have any additional questions. During the rebuttal, we have clarified that:
>
> + Our Alg. 1 is finetuning a pretrained model, not retraining from scratch.
> + Alg. 1 is not incorporating random labels into $\mathcal{D}_f$ and then retraining, since this does not work at scale while Alg. 1 does.
> + Whether the adversary is aware that $\mathcal{D}_f$ is corrupted or not is immaterial for the adversary. The adversary seeks to obtain an automated warrant to e.g. deny insurance coverage.
> + Our method focuses on prevention rather than correction; instead of appealing to a third-party after harm has occurred, we aim to prevent harm in the first place.
>
> We also provided empirical results for Alg. 3 applied to logistic regression trained on MNIST and noted its sensitivity to hyperparameters. **Below, we provide new empirical results for Alg. 3** applied to SVM trained on MNIST and a MLP trained on MNIST. Hyperparameter details are reported in Appx. K. We find that, with appropriately chosen $\lambda$ and $J$, we can induce uniformity. As discussed in our rebuttal, the estimation error incurred by the Hessian estimator results in lower retain and test accuracy than those of Alg. 1. We have updated the manuscript with these results in Table L.24.
>
> | **Dataset** | **Model** | **Method** | **Retain Acc.** | **Test Acc.** | **Conf. Dist. (Lower Better)** |
> | :--- | :--- | :--- | :---: | :---: | :---: |
> | MNIST | SVM | Pretrain | 92.3% | 92.1% | 0.543 |
> | | | Alg. 1 | 90.3% | 90.1% | 0.241 |
> | | | Alg. 3 | 83.9% | 83.9% | 0.215 |
> | | MLP | Pretrain | 98.3% | 97.0% | 0.886 |
> | | | Alg. 1 | 97.5% | 95.7% | 0.037 |
> | | | Alg. 3 | 88.8% | 87.5% | 0.057 |
>
> We again thank you for your time and would appreciate it if you could confirm that the original questions have been addressed. We are happy to clarify any further questions.
>
> Best,
>
> The Authors

---

### Meta-Review · Area_Chair_ZqvS · 2026-01-06

**Summary:**

Reviewer TVYj's initial score was 4, with the highest confidence 5. The reviewer asked how their algorithm compares with retraining a model that mixes $D_f$ into the training distribution with random labels, as stated in Weakness 1. The authors provided new experimental results on CIFAR-10 and MNIST, showing that their Algorithm 1 can induce uniformity, whereas the reviewer's suggestion cannot. If I were the reviewer, I would further request an investigation using datasets with more than 10 classes.

Somewhat related, Reviewer 8s6z asked whether their approach could extend to generative models (more precisely T2I cases). I feel this is a valid and quite relevant question, while the authors' answers are relatively dry and unenthusiastic, writing "this is out of scope".

Reviewer TfiQ questioned the method’s optimality for TTP and the TTP guarantees. The authors diligently replied with new experiments, together with detailed explanations of the certified TTP definition. This reviewer might have increased the score based on the quality of the authors' answers.

**Reviewer Concerns:**

See above.

**Reviewer Scores:**

It feels like it needs more discussion between Reviewer TVYj and the authors to reach a sound judgment of this paper, as currently Reviewer TVYj's questions are there, and also the authors' responses are there, but it is impossible to know whether the authors' answers would be satisfying to Reviewer TVYj. As mentioned above, my personal feeling is that the reviewer would have more follow-up questions regarding the uniformity, and the current replies by the authors are not convincing enough for me to fully convince myself that this issue is resolved.

Reviewer 8s6z is the most positive reviewer among the three, but based on the quality of the review and a low confidence, score 8 does not seem clearly justified. For this reason, I would downweigh this reviewer's score but instead trust the other two reviewers' judgment more.

---

### Decision · Program_Chairs · 2026-01-26

Reject